# Training tactile sensors to learn force sensing from each other

Zhuo Chen [1] ✉, Ni Ou [1], Xuyang Zhang [1], Zhiyuan Wu[1], Yongqiang Zhao [1], Yupeng Wang [1], Emmanouil Spyrakos Papastavridis [1], Nathan Lepora [2], Lorenzo Jamone[3], Jiankang Deng [4] ✉ & Shan Luo [1] ✉

Humans achieve stable and dexterous object manipulation by coordinating grasp forces across multiple fingers and palms, facilitated by a unified tactile memory system in the somatosensory cortex. This system encodes and stores tactile experiences across skin regions, enabling the flexible reuse and transfer of touch information. Inspired by this biological capability, we present Gen-Force, the first framework that enables transferable force sensing across diverse tactile sensors in robotic hands. GenForce unifies tactile signals into shared marker representations, analogous to cortical sensory encoding, allowing force prediction models trained on one sensor to be transferred to others without the need for exhaustive force data collection. We demonstrate that GenForce generalizes across both homogeneous sensors with varying configurations and heterogeneous sensors with distinct sensing modalities and material properties. This transferable force sensing capability is also demonstrated in robot manipulation tasks including daily-object grasping, slip detection and compensation with multi-sensor force coordination. Our results highlight a scalable paradigm for cross-sensor robotic tactile sensing, offering new pathways toward adaptable and tactile memory-driven robot manipulation in unstructured environments.

Robots capable of perceiving and interacting with the environment are revolutionizing the healthcare, service, and manufacturing industries[1,2]. While these robots rely on various sensing modalities, tactile sensing is indispensable for achieving dexterous manipulation and ensuring safe human-robot interaction[3,4]. However, replicating the human sense of touch in robotic systems remains a fundamental challenge, particularly for tasks requiring precise in-hand force feedback.

Human skin is equipped with diverse sensory receptors that detect mechanical stimuli and provide rich contact information[5]. Inspired by this, tactile robotics aims to mirror the function of human mechanoreceptors to enhance dexterity and intelligence[6,7] by developing various bio-inspired tactile sensing systems (Fig. 1A–i). These tactile sensors are based on various sensing principles, such as piezoresistive[8], capacitive[9], magnetic[10,11], and optical[12,13] transductions, offering unique advantages in detecting static or dynamic contact information. For example (Fig. 1A–ii), GelSight[13] provides high-resolution tactile images by using a camera to capture detailed textures and geometries, making it ideal for tasks requiring fine-grained tactile feedback, like object recognition or 3D reconstruction[14]. uSkin[15] uses Hall-effect sensors to measure displacements of embedded magnets, offering high sensitivity to multi-axis forces and compactness for precise manipulation tasks. TacTip[12,16] features the tracking of biomimetic pin movement with a camera, offering robust force, shape, and slip detection abilities in dynamic environments.

However, diversity in sensing principles, structural designs, and material properties creates significant domain gaps among tactile sensors, hindering the transfer of learned tactile experience and

[1]King's College London, London, UK. [2]University of Bristol, Bristol, UK. [3]University College London, London, UK. [4]Imperial College London, London, UK.
✉e-mail: zhuo.7.chen@kcl.ac.uk; j.deng16@imperial.ac.uk; shan.luo@kcl.ac.uk

necessitating repetitive data collection. For example, training force prediction models[17] for robotic hands with multiple tactile sensors (Fig. 1B) requires extensive paired tactile signal-force label datasets for each sensor, introducing practical barriers due to costly force/torque sensors (such as ATI Nano17) and time-consuming data collection. This challenge is exacerbated by the inevitable degradation of soft elastomers from wear and tear, requiring frequent replacement and recalibration. Thus, current practice demands repeated collection of paired tactile signal-force labels and retraining of force prediction models for each new sensor deployment.

The ability to unify and transfer sensory information across skin regions is a critical aspect of human control over grasp forces, ensuring both stability and dexterity during object manipulation[18,19]. This capability is equally essential for robots equipped with tactile sensors to perform in-hand object manipulation tasks in unstructured environments[20]. In humans, the tactile memory system (Fig. 1C) enables the storage and retrieval of experienced tactile information, such as haptic stimuli, across skin regions on hands[21,22]. Mechanoreceptors in the skin detect deformation, which is translated into a unified sensory encoding, and transmitted to the somatosensory cortex via peripheral nerves for storage and processing[5]. This human ability to adapt, unify, and transfer tactile sensation offers valuable inspiration for developing transferable tactile sensing in robots. Mimicking this unified representation and transferable tactile sensing could enable tactile sensors to learn from each other, reuse collected tactile experience and transfer tactile sensing across robotic hands, enhancing their dexterity and adaptability.

Recent advances in transfer learning[23] and representation learning[24–28] have made strides toward unifying latent tactile representations and enabling the transfer of tactile experience. However, existing approaches face significant limitations compared to the human capability of tactile memory. For example, traditional transfer learning techniques still require the collection of labeled data from new sensors for finetuning[29]. Recent reported representation learning methods[24–28] focus only on vision-based tactile sensors and align sensor representations in feature space, necessitating large-scale datasets from different sensors and often depend on task-specific decoders trained with labeled data to enhance accuracy. Some efforts to directly transform tactile signals across sensors[30–33] similarly fall short on one-to-one translation, as they overlook the importance of a unified tactile representation among sensors and lack generalizability to diverse tactile sensors with force-prediction capability. Critically, the above approaches fail to account for the fundamental material differences among sensors, which are essential for tasks demanding high accuracy such as force prediction.

We present GenForce (Fig. 1D), the first general framework to enable transferable force sensing across diverse tactile sensors. GenForce unifies tactile signals from skin deformation into shared marker representations, analogous to sensory encoding, and enables deformation transfer across sensors regardless of sensing principles and physical configurations through marker-to-marker translation (Fig. 2C). Force prediction capability (Fig. 2D) can therefore be learned with generated images and force labels from other sensors. GenForce further improves force accuracy by compensating for material differences. Extensively validated for generalizability, accuracy, and robustness, GenForce is applicable to diverse tactile sensors, including flat-surface vision-based tactile sensors (GelSight[13]), flat-surface electronic sensor arrays (uSkin[15]) with either three-axis sensing or z-axis-only sensing, and curved-surface vision-based tactile sensors (TacTip[16]), spanning varying configurations (Fig. 2A–B). This approach significantly redefines the paradigm of training force prediction models: instead of relying on repetitive and time-consuming data collection for each sensor, models can be trained by learning across sensors, greatly facilitating large-scale tactile sensing deployment. Beyond transferable force sensing, GenForce demonstrates practical

applicability in robot manipulation tasks with multi-sensor force coordination across heterogeneous tactile sensors, including daily-object grasping, slip detection and compensation (see Supplementary Videos 4-7). Our study lays a foundation for cross-sensor tactile sensing in tactile robotics[34], enabling the transfer of multimodal sensory skills and paving the way for the development of human-like embodied AI.

## Results

### Arbitrary marker-to-marker translation

To test the generalizability of the marker-to-marker translation (M2M) for diverse tactile sensors, we first propose a simple simulation pipeline to acquire extensive deformed marker images (Fig. 3A). We design 12 distinct reference marker patterns, i.e., Array (A), Circle (C) and Diamond (D) shapes with varying sizes and densities referring to GelSight[13], uSkin[15], TacTip[16] and GelTip[35] sensors. Eighteen 3D-printed indenters with diverse geometrical properties (vertices, edges, and curvatures) are used for indentation[33]. Each marker pattern can serve as both source sensor and target sensor, resulting in a total of $12 \times 11 = 132$ sensor combinations. We employ two quantitative metrics[36], including Fréchet Inception Distance (FID) and Kernel Inception Distance (KID), to assess image similarity. Lower values indicate greater visual similarity. The visualization of feature spaces uses t-distributed stochastic neighbor embedding (t-SNE)[37]. Note that, we abbreviate each source-to-target combination as source_target. For example, we write sensor combination D2-to-C1 as D2_C1 for simplification.

As demonstrated in Fig. 3B, C, approximately 100-fold decreases are found in FID and KID across all combinations after using M2M, indicating the generated images successfully represent the image styles of the targe images while aligning the deformations with the source images. Specifically, before using M2M (Fig. 3B–i and Fig. 3C–i), as each type of marker pattern is visually distinctive (Fig. 3A), the difference is prominent with an average FID larger than 400 and an average KID larger than 0.75. The source images and target images are separated distinctly in the feature space (Fig. 3D–i). After using M2M, the average FID (Fig. 3B–ii) drops to 4 and the average KID (see Fig. 3C–ii) drops to 0.01. The generated images and target images are aligned closely in feature space (Fig. 3D–ii) and visually indistinguishable (see Supplementary Fig. 1D and Supplementary Video 1). This suggests the effectiveness of the marker-to-marker translation across various marker patterns, providing a pretrained model for the marker-to-marker translation in real sensors.

### Learning across homogeneous tactile sensors

Large-scale deployment of homogeneous tactile sensors on robotic hands has garnered significant attention[38,39]. A common approach is to directly apply trained force prediction models from existing sensors to other sensors[39], i.e., the source-only method. However, sensor variations often exist due to inconsistencies in fabrication and will lead to large force errors by using source-only method. We validate our GenForce model across homogeneous sensors, particularly for GelSight sensors due to widespread use in robotic hands[14,38,39]. We detail the study of material effects in the next section and focus here on the variations in illuminations and marker patterns.

We build the data collection setup (Fig. 4A) and fabricate five GelSight sensors (Supplementary Fig. 4A), i.e., Array-I (A-I), Array-II (A-II), Circle-I (C-I), Circle-II (C-II), and Diamond-I (D-I). We divide all indenters into two groups (Supplementary Fig. 1A): a seen group of 12 indenters (used for training) and an unseen group of 6 indenters (excluded from training). Approximately 180,000 force-image pairs are collected per sensor. For M2M model training, we only use the last four images during the movement to a target position as location-paired images (Fig. 4B), resulting in a total of 17,280 images per sensor. The distribution of the data across different force ranges is shown in

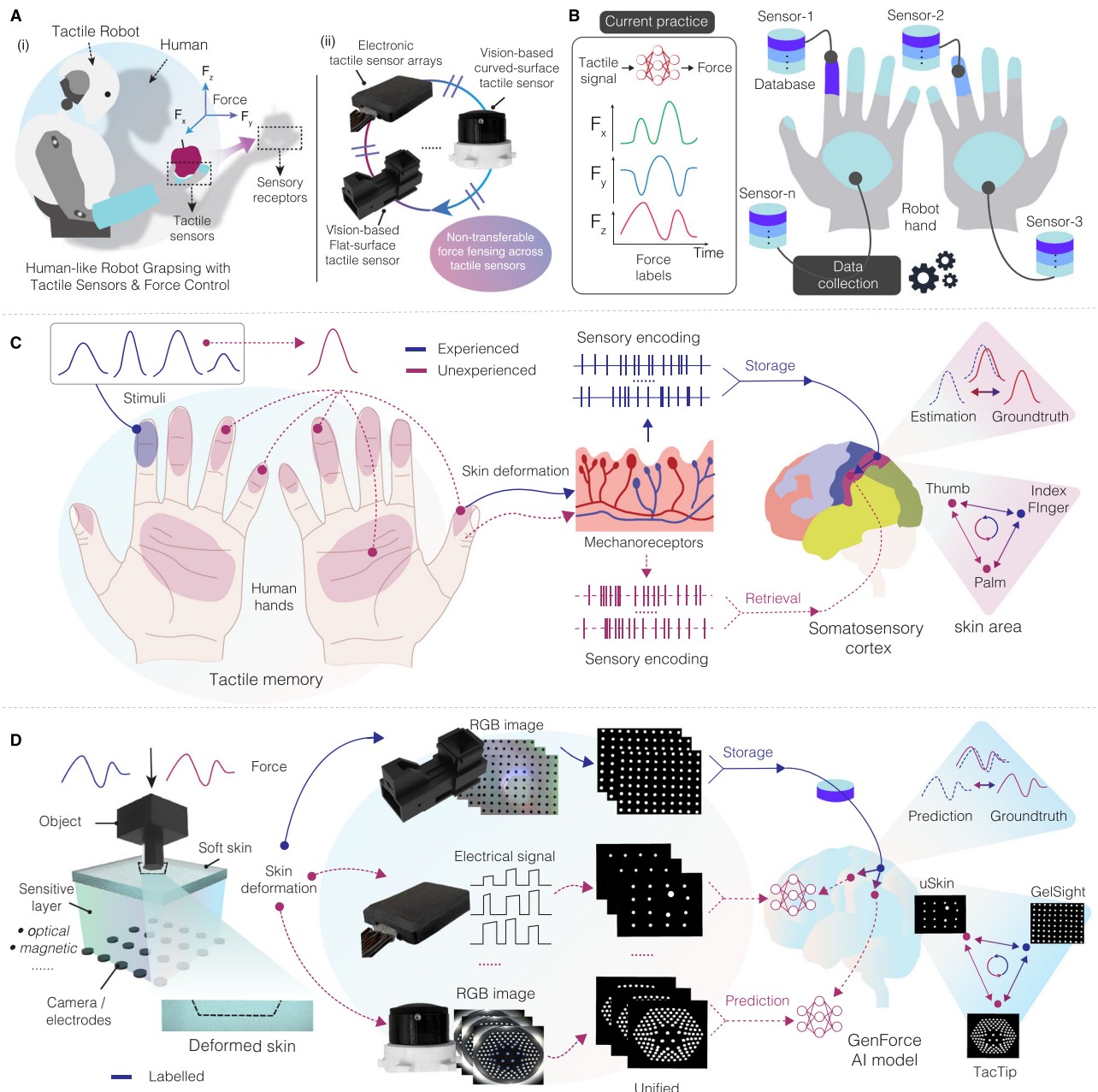

**Fig. 1 | Transferable force sensing. A** Robot grasping objects with tactile sensors and force control mimics human actions with sensory receptors. These bio-inspired tactile sensors cannot transfer force data with each other due to differences in sensing principles, structural designs and material properties. **B** Current practice to train force prediction models uses repetitive and costly data collection process for force labels. **C** Humans use a tactile memory system to estimate stimuli on unexperienced skin regions by retrieving tactile memories stored in the somatosensory cortex. **D** Overview of the GenForce model. Tactile sensors produce diverse tactile signals under the same deformation due to differences in sensing principles, structural designs and material properties. GenForce unifies tactile signals into marker representation, enables marker-to-marker translation across various sensors, and achieves high-accuracy force prediction on uncalibrated sensors using data transferred from calibrated sensors.

Supplementary Fig. 6A. It covers normal forces from −16 N to 0 N and shear forces from −6 N to 6 N. Collected tactile images are showcased in Supplementary Fig. 3. We utilize a marker segmentation method (Fig. 4C) to obtain marker-based tactile images from RGB tactile images (more details in Supplementary Fig. 4B and Methods). The pre-trained M2M model from the simulation is then fine-tuned with location-paired marker images from the seen group.

Similar to the simulation results, image similarity (see Supplementary Fig. 5B–D) improves significantly and the generated images and the target images across all combinations are well-aligned. The FID and KID values both drop by more than 98%. The results from the unseen group (Supplementary Fig. 5E–G) demonstrate comparable performance to that of the seen group. Furthermore, the generated images showcased in Supplementary Fig. 5A and Supplementary Video 1 validate the generalizability of the M2M model to real-world homogeneous tactile sensors.

Regarding force prediction performance, the source-only method exhibits large errors (Fig. 4D, E and Supplementary Video 2). The maximum error in normal force exceeds 4.8 N. Shear force errors average above 0.28 N. In addition, most sensor combinations have negative $R^2$ values and high variance (Fig. 4E) and demonstrate poor performance in real-time force prediction (Supplementary Fig. 6B).

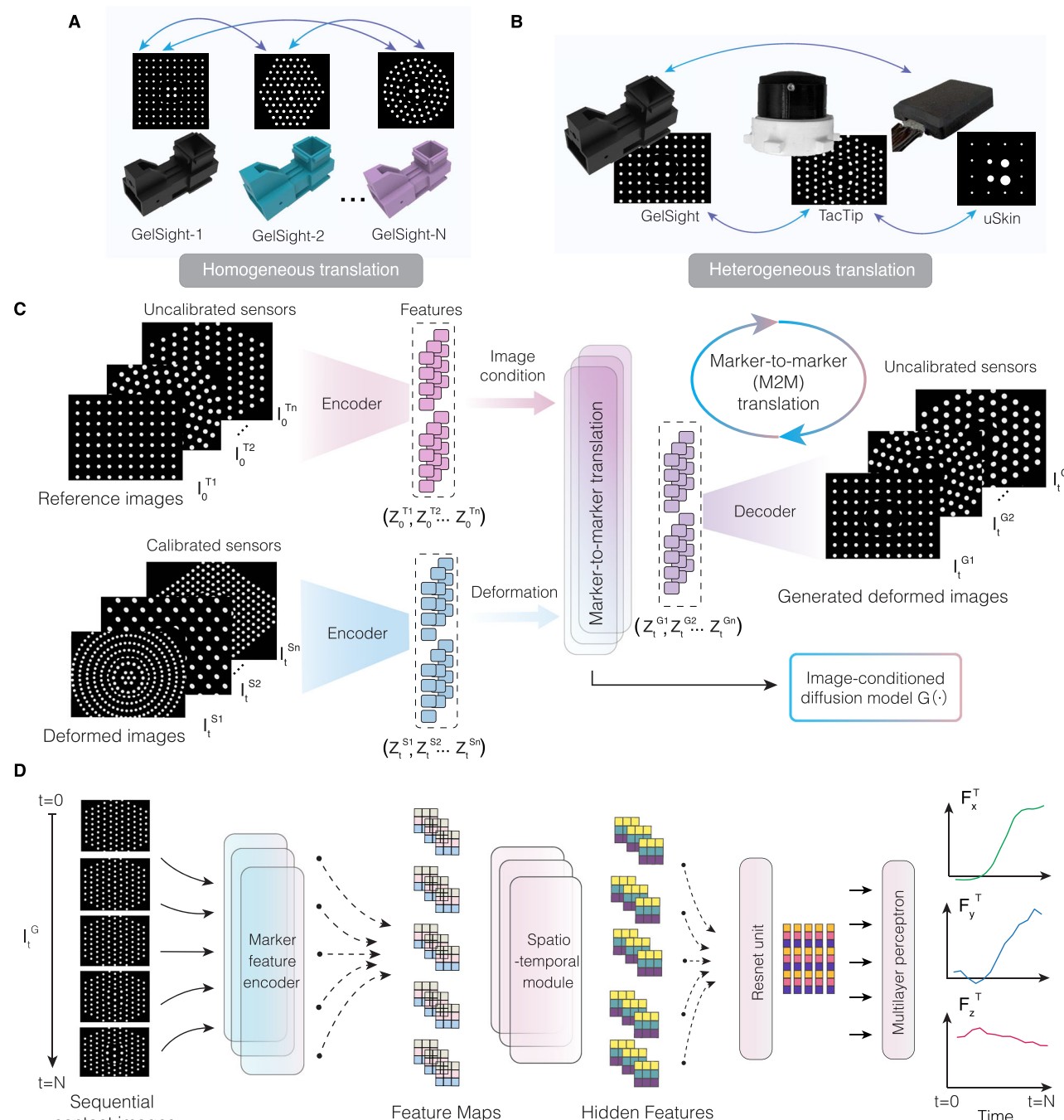

**Fig. 2 | Marker-to-marker translation and spatiotemporal force prediction.**
**A** Homogeneous translation demonstrated with GelSight sensors featuring varying marker patterns. **B** Heterogeneous translation demonstrated with GelSight, Tac-Tip, and uSkin sensors. **C** Marker-to-marker translation (M2M) model. The model takes deformed images from calibrated sensors as input, conditioned on reference images from uncalibrated sensors, to synthesize deformed images that mimic the response of the uncalibrated sensors. **D** Spatiotemporal force prediction model. The network processes sequential contact images to predict three-axis forces, leveraging a spatiotemporal module to enhance accuracy.

After using GenForce model, all force errors are significantly reduced (Fig. 4D), and $R^2$ values (Fig. 4E) improve across all combinations. For normal force, the maximum error is below 1 N, while the minimum error decreases to less than 0.7 N. Notably, the maximum force error in C-II_D-I improves from 4.8 N to 0.96 N (80% reduction). For shear forces, the minimum errors decrease to less than 0.06 N. The $R^2$ values show consistent improvement in both normal and shear direction, averaging above 0.8. Real-time force predictions in Fig. 4F and Supplementary Video 2 demonstrate well-aligned predicted forces and ground truths. The half-violin plots (Supplementary Fig. 6C) illustrate

high-accuracy force prediction in both normal (−8 N to 0 N) and shear (−3 N to 3 N) directions. Additional results from the unseen group (Supplementary Fig. 7) further validate our model's robustness and generalizability.

## Skin hardness effect and compensation
The deformable skin of tactile sensors is akin to human skin, where its hardness property is often customized for specific tasks, and it also changes with long-term use and replacement due to wear and tear. The difference in material hardness introduces additional errors when

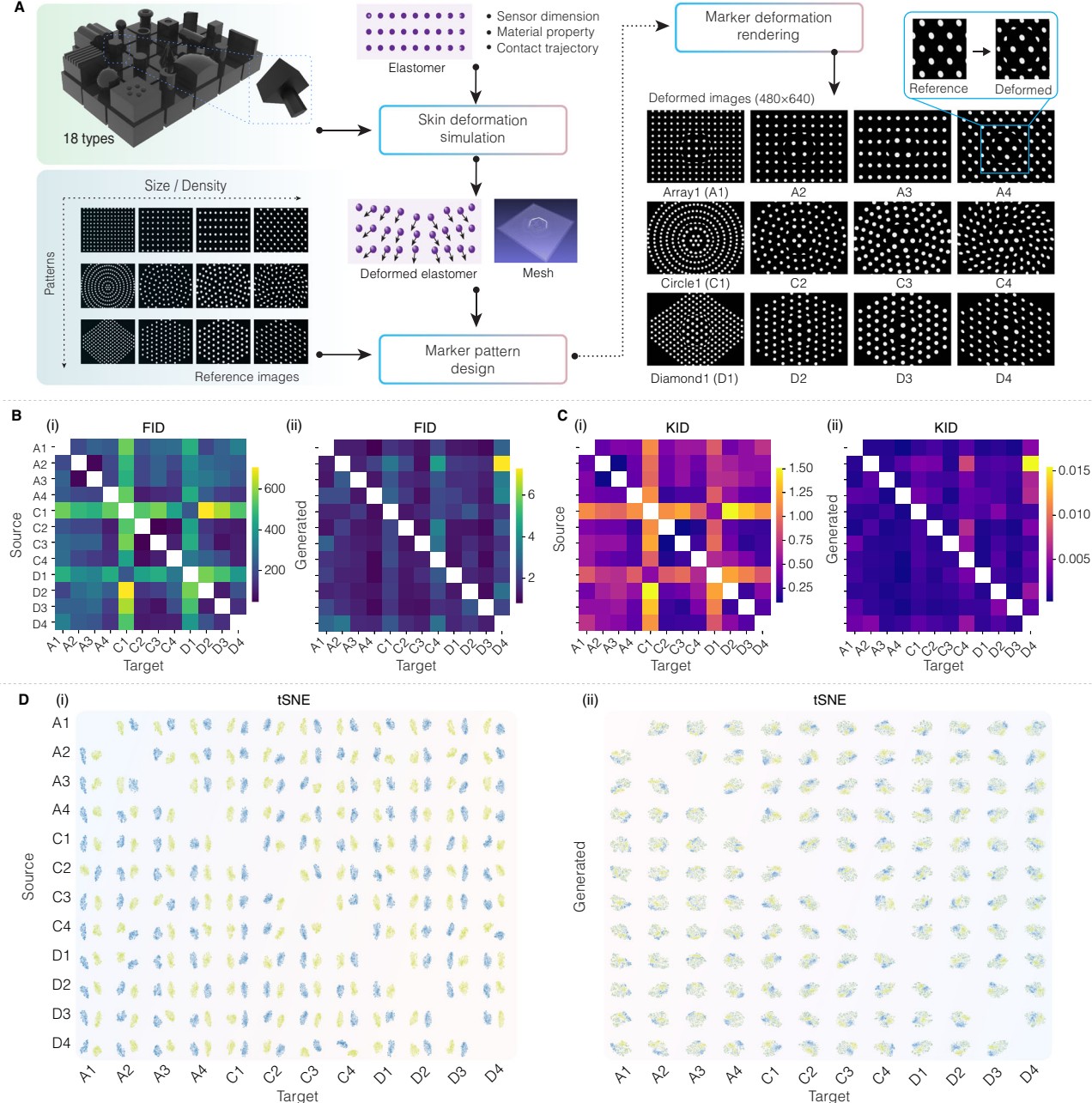

**Fig. 3 | Marker-to-marker translation in simulation. A** Pipeline for marker deformation simulation. Quantitative evaluation of marker-to-marker translation via heatmaps of (**B**) FID and (**C**) KID, comparing performance before (i) and after (ii) applying the M2M model across all 132 sensor combinations. **D** t-SNE feature space visualization illustrating the alignment between target images and generated images across all sensor combinations, before (i) and after (ii) applying the M2M model.

transferring the force labels (Fig. 5A). We investigate the skin hardness effect and compensation method in this section. Seven silicone elastomers with varied hardness are fabricated by controlling the base-to-activator (B:A) ratio from ratio-6 (r6) to ratio-18 (r18) with a step of 2 (see Methods and Supplementary Fig. 9A). The higher the ratio, the softer the skin becomes. We measure the force-normalized depth curve (Fig. 5B) for each skin during loading and unloading phases. Due to the hysteresis property, the contact force in the loading phase is found greater than that in the unloading phase at the same depth. This effect reveals a limitation of traditional methods that predict forces using single images[17] as the same contact image may correspond to different contact forces during the loading and unloading phases. The measured relationships (Fig. 5C) between the shear-to-normal force ratio $F_S/F_N$ and shear displacement reveal distinct coefficients of

friction across seven elastomers (see Supplementary Text 9). Representative images from the seven sensors with varying hardness are showcased in Supplementary Fig. 8. The distributions of data points in different force ranges are shown in Supplementary Fig. 10A.

We divide the transfer directions into two groups: (1) hard-to-soft and (2) soft-to-hard. In Fig. 5E–i, combinations like 6_8, 6_10, and 6_12 indicate progressive transfer to softer elastomers (slightly softer, softer, and much softer), whereas 8_6, 10_6, and 12_6 indicate the inverse, i.e., transfer to harder elastomers. These combinations reflect the long-term aging behavior of silicone elastomers. Before material compensation, normal-force errors generally grow with increasing hardness gap (Fig. 5E). On average, the normal-force error is 1.41 N for the hard-to-soft group and 1.03 N for the soft-to-hard group. For shear forces, average errors are 0.18 N (x) and 0.20 N (y) for hard-to-soft, and

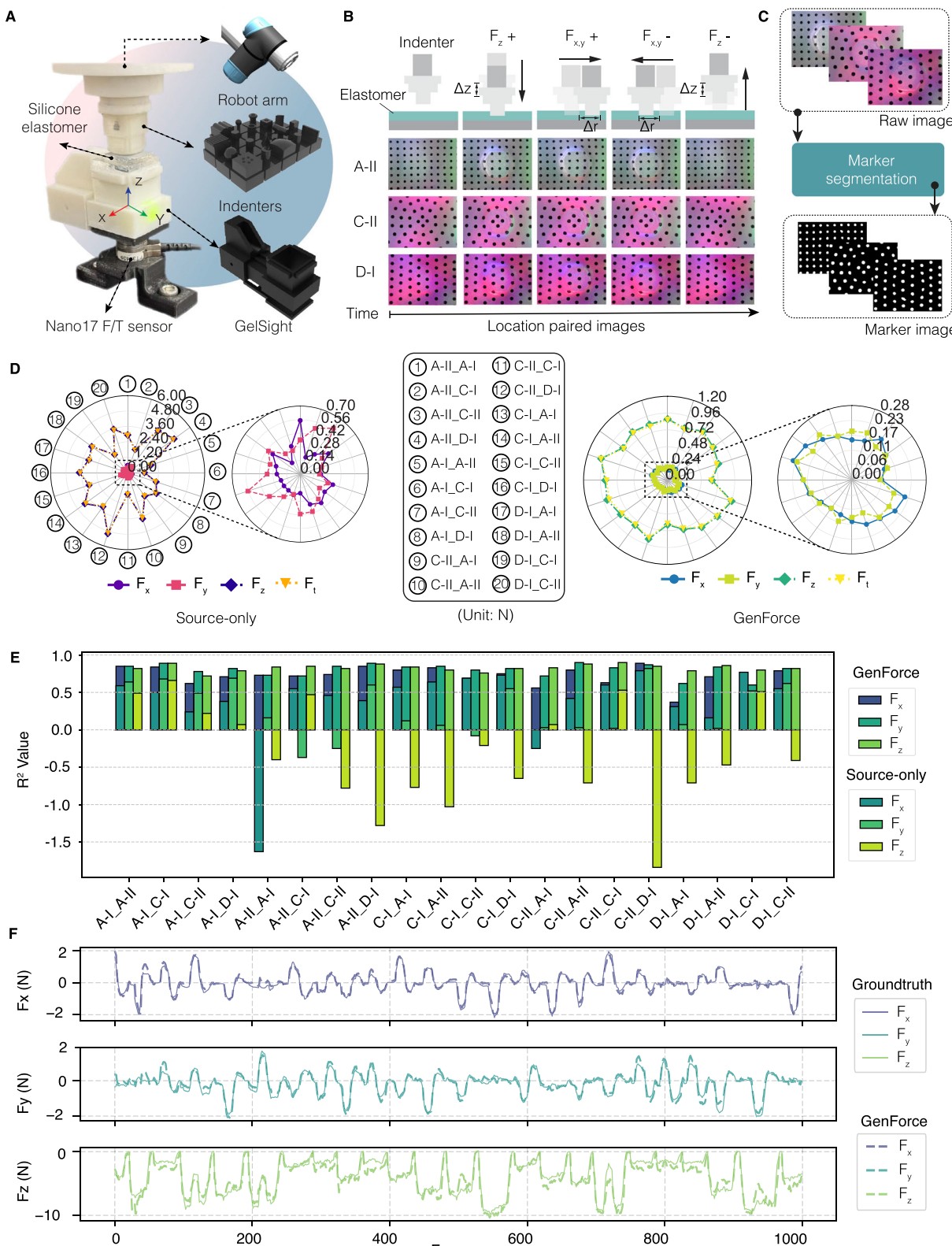

**Fig. 4 | Homogeneous tactile force translation. A** Data collection setup. **B** Data collection trajectory used to acquire sequential tactile images under normal and shear forces. **C** Marker segmentation process for RGB tactile images. **D** Radar plots comparing force prediction errors between the source-only method and the GenForce model. $F_t$ denotes total force. All units are in Newton (N). **E** Histogram of the coefficient of determination ($R^2$) values for predicted forces using source-only method and GenForce model. **F** Real-time force prediction trace over 1000 frames after using the GenForce model for the A-II_D-I group.

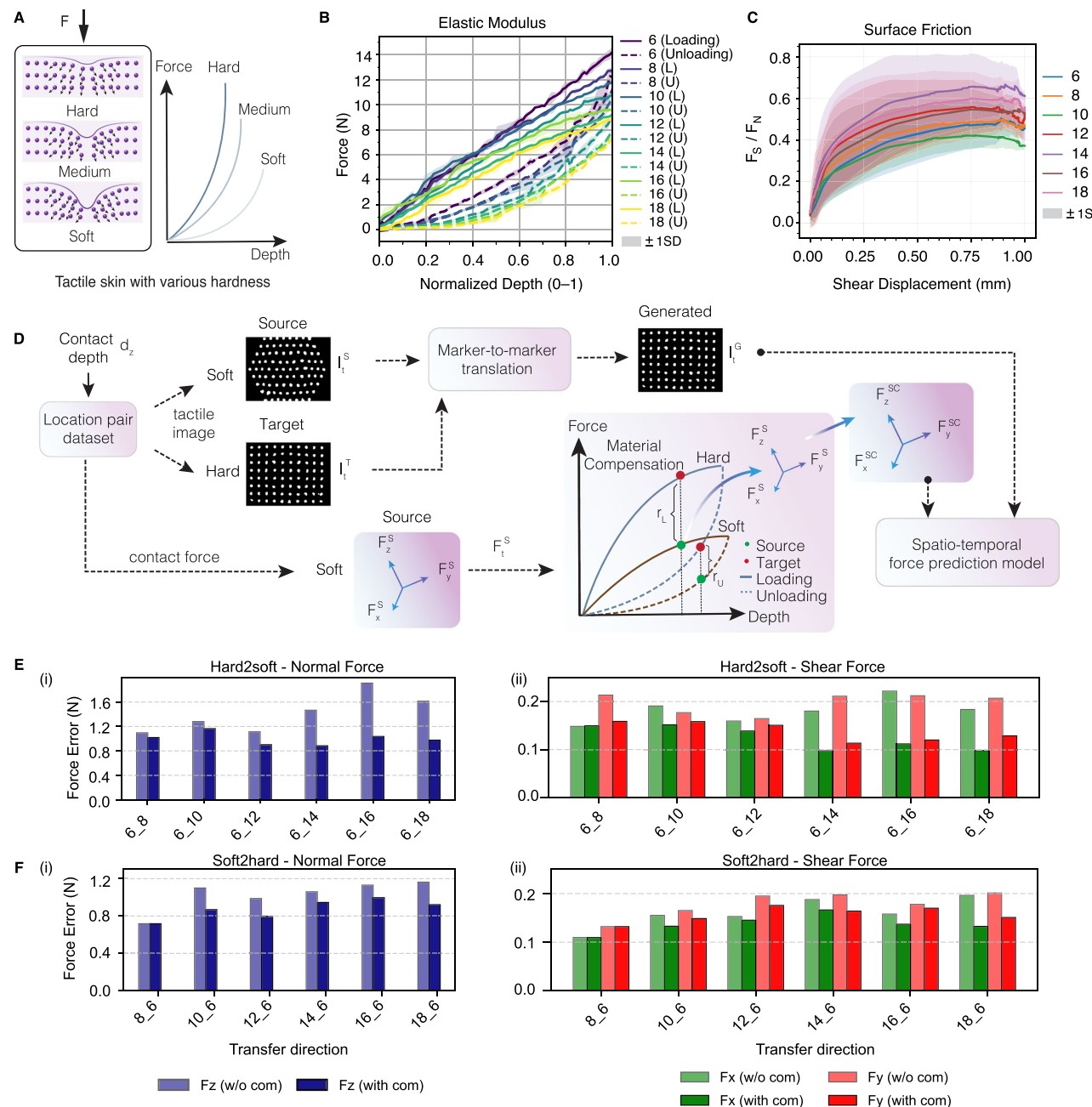

**Fig. 5 | Material hardness effect and compensation. A** Schematic representation of tactile skins with varying hardness levels and their corresponding force-depth relationships. **B** Measured force-normalized depth curves (mean ±1SD) for seven elastomers during loading and unloading phases, illustrating their distinct hardness and hysteresis properties. **C** Relationship between the shear-to-normal force ratio $F_S/F_N$ and shear displacement (mean ±1SD) of the seven elastomers, measured across various contact points. **D** The material compensation process, which leverages material priors to correct force labels during both loading and unloading phases. Force prediction errors (i-ii) for GenForce models without (w/o) and with material compensation in hard-to-soft group (**E**) and soft-to-hard group (**F**).

0.16 N (x) and 0.18 N (y) for soft-to-hard. Shear force errors are more stable across both groups, because applied shear forces are smaller than normal forces (see Supplementary Fig. 10A). Additional results for other 15 combinations are provided in Supplementary Fig. 9D–E.

To mitigate the increased force errors due to material hardness gaps, we propose a material compensation method (Fig. 5B) that uses material priors to correct force labels before training (see theoretical derivation in Supplementary Text 9). The priors are mainly the fitted elastic modulus profiles, i.e., force-normalized depth profiles shown in Supplementary Fig. 9C. Intuitively, this process scales up the force magnitude when transferring from soft-to-hard skin and scales it down in the opposite direction. As shown in Fig. 5E, F, this improves accuracy

across both normal and shear forces. After compensation, the average normal-force error drops to 0.99 N in the hard-to-soft group (30% reduction) and 0.87 N in the soft-to-hard group (16% reduction). For shear forces, the average errors in the hard-to-soft group reduce to 0.10 N on the x-axis (44% reduction) and 0.14 N on the y-axis (30% reduction); in the soft-to-hard group, errors reduce to 0.13 N ($F_x$, 19% reduction) and 0.15 N ($F_y$, 19% reduction). The method yields lower errors in 95% of all combinations within the hard-to-soft group (including those in Supplementary Fig. 9D-E) and in 57% of the soft-to-hard group. A representative case, transferring from r18 (soft) to r6 (much harder) skins in Supplementary Fig. 10C, shows improvements in $R^2$ values from 0.73 ($F_x$), 0.79 ($F_y$), and 0.78 ($F_z$) to 0.87, 0.86, and

0.84 across the three axes. Real-time visualizations in Supplementary Fig. 10B and Supplementary Video 2 further illustrate the effectiveness of the material compensation, which is often overlooked in traditional studies despite its critical role in force sensing.

### Learning across heterogeneous tactile sensors

Heterogeneous tactile sensors are designed with varying sensing principles, structural designs, and material properties to mimic human mechanoreceptors, presenting significant challenges for cross-sensor learning. For example, TacTip[16] features 127 markers distributed on a surface with a curvature of 41.5 mm and a diameter of 40 mm, and allows for a maximum indentation of 5 mm. By contrast, GelSight[13] is equipped with 56 markers within a volume of 25×25×4 mm, with a maximum indentation of 1.5 mm. uSkin[15] has dimensions of 24.6×22.6×5 mm and outputs multichannel signals via 4×4 taxels (16 markers), where each taxel can withstand a maximum force of 20 N.

We unify three distinct tactile signals (Supplementary Fig. 13A) into marker representation. For electronic sensor arrays, we develop a signal-to-marker pipeline (Fig. 6A) that converts multichannel raw signals into marker displacements and diameter changes (see more details in Supplementary Fig. 11A–B). An example conversion from uSkin (three-axis sensing) is demonstrated in Supplementary Fig. 12B. Although some electronic sensor arrays only measure pressure (z-axis) at each taxel, unlike magnetic sensors that offer three-axis measurement, our model is capable of handling this scenario, as verified by using only the z-component of uSkin, denoted as uSkin (z-axis) (see Supplementary Text 10 and Supplementary Video 3). For TacTip, we extract markers using a segmentation method (Supplementary Fig. 11C). The measured force-normalized depth relationships for the three sensors with varying indentation depths: 1 mm (uSkin), 1 mm (GelSight), and 4.5 mm (TacTip) are shown in Supplementary Fig. 15A. Despite TacTip's deeper contact depth, its force at maximum normalized depth is still four times smaller than that of uSkin and two times smaller than that of GelSight due to the extremely soft gel used as the soft skin[16]. The coefficients of friction are measured and shown in Supplementary Fig. 16A. Representative tactile images for the three heterogeneous tactile sensors are shown in Supplementary Fig. 13B–D. The distribution of data points collected across different force ranges is shown in Supplementary Fig. 18B.

The improved FID and KID scores, along with the aligned feature space in tSNE (Supplementary Fig. 14), suggest the M2M model's generalization capability across heterogeneous sensors. We showcase generated images and source images in Fig. 6B and the Supplementary Video 1. For force prediction, we observe that (Fig. 6C–i), the $R^2$ values are negative across all three axes when transferring from lower marker density to higher density using source-only method, i.e., uSkin_TacTip, uSkin_GelSight, and GelSight_TacTip. The maximum MAE reaches 7.76 N for $F_z$, 0.59 N for $F_y$, and 0.37 N for $F_x$. By contrast, when transferring from high density to lower density, the remaining three combinations exhibit better performance but still have high errors. These large force errors are also coupled with the material effects, which is prominent when transferring from the uSkin sensor, with the hardest skin, to the other two sensors with softer skins. After using GenForce but without compensation, force prediction errors (Supplementary Fig. 17A) improve substantially. However, due to the remaining gaps in skin hardness, GelSight_TacTip and uSkin_TacTip still exhibit negative $R^2$ values. The visualization in Supplementary Fig. 17B also demonstrates the non-negligible errors between predicted forces and ground truths due to material effects.

Upon applying material compensation, all $R^2$ values (Fig. 6C–ii) improve to positive values, following an ascending performance order: GelSight_uSkin, uSkin_GelSight, TacTip_GelSight, GelSight_TacTip, TacTip_uSkin, and uSkin_TacTip. This order correlates with the differences in material hardness and marker densities. Transfers between GelSight and uSkin show lower errors compared to combinations

involving TacTip due to their similar material properties. The average of MAE (Fig. 6C–iii and Fig. 6C–iv) over all six combinations decreases below 0.92 N for normal force, while $F_x$ and $F_y$ reduce below 0.22 N and 0.3 N, respectively. Notably, the uSkin_TacTip group shows a 93% improvement of $F_z$ in MAE (from 7.76 N to 0.52 N) and a 66% improvement of $F_y$ (from 0.59 N to 0.2 N). The continuous force predictions (see Fig. 6D and Supplementary Fig. 18A) also demonstrate the mitigation of material effects. In Supplementary Fig. 18C, the force errors for all combinations are centered around zero within ranges of −4N to 0 N in the normal direction and −3N to 3 N in the shear direction, demonstrating both the accuracy and reliability of our model in the most challenging task of heterogeneous translation.

We also qualitatively evaluate our model in real-time across six transfer groups under more dynamic conditions (see Fig. 6E and Supplementary Video 3). Tactile sensors are mounted on an ATI Nano17 F/T sensor, and we apply random forces using four daily objects unseen in our dataset with different shapes and materials, including a screwdriver, a glue stick, a plastic pizza, and a LEGO block (Supplementary Fig. 19). A human operator performs five unstructured, dynamic contact actions commonly encountered in robot manipulation tasks, including pressing, rubbing, rolling, pushing, and pulling, as well as continuous combinations of these on the sensor surfaces. The transfer groups cover both the homogeneous and heterogeneous translations described earlier. As shown in Supplementary Video 3 and Supplementary Figs. 20–24, all test groups exhibit strong generalizability to new objects and achieve low force prediction errors when compared against a commercial high-accuracy F/T sensor.

### Transferable force sensing in robot manipulation

To demonstrate the applicability of our model in robot manipulation, we install heterogeneous tactile sensors on a robot hand to show the transferable force sensing and control during daily-object grasping, slip detection and compensation tasks. In the robot grasping task, we mount a flat-surface vision-based tactile sensor (GelSight, marker pattern A-II) on the left finger of the robot hand while installing a flat-surface magnetic tactile sensor (uSkin with three-axis sensing capability) on the right finger. We transfer force prediction models to these two sensors from a third flat-surface vision-based tactile sensor (GelSight, marker pattern D-I). The robot grasping setup can be seen in Supplementary Fig. 27A–B. The task requires the robot to grasp nine daily objects with different sizes, shapes and materials without damaging them. These objects include a potato chip, a grape, a strawberry, an orange, a plum, a wood block, a glue stick, a meat box and a tea box, all of which are unseen in the training dataset. During grasping, the robot hand is controlled with a shared proportional controller to apply fixed normal forces ranging from 0.6 N to 1.2 N by using one of the installed sensors. As shown in Supplementary Fig. 27B and Supplementary Videos 4-5, the robot equipped with transferable force sensing successfully grasps all objects without causing damage. Even with fragile objects such as chips and fresh fruits, the robot achieves delicate grasping.

For the second task, we extend the robot grasping scenario to include slip detection and compensation via multi-sensor force coordination (Fig. 7). A curved-surface vision-based TacTip sensor is mounted on the left finger of the robot hand, while a three-axis magnetic uSkin sensor is mounted on the right (Fig. 7A). The y-axis of the uSkin is inverted due to the mirrored configuration. The force prediction model for the TacTip is transferred from a GelSight (D-I), and the force prediction model for the uSkin is transferred from a TacTip. These two transfer directions are designed to demonstrate the generalizability of our model across diverse transfer groups for the real-world robotic tasks. The task proceeds through several stages: moving down, grasping, lifting, slip detection and compensation at the top position, releasing, and returning to the home position. The force controller of the robot hand is active only during the grasping and slip

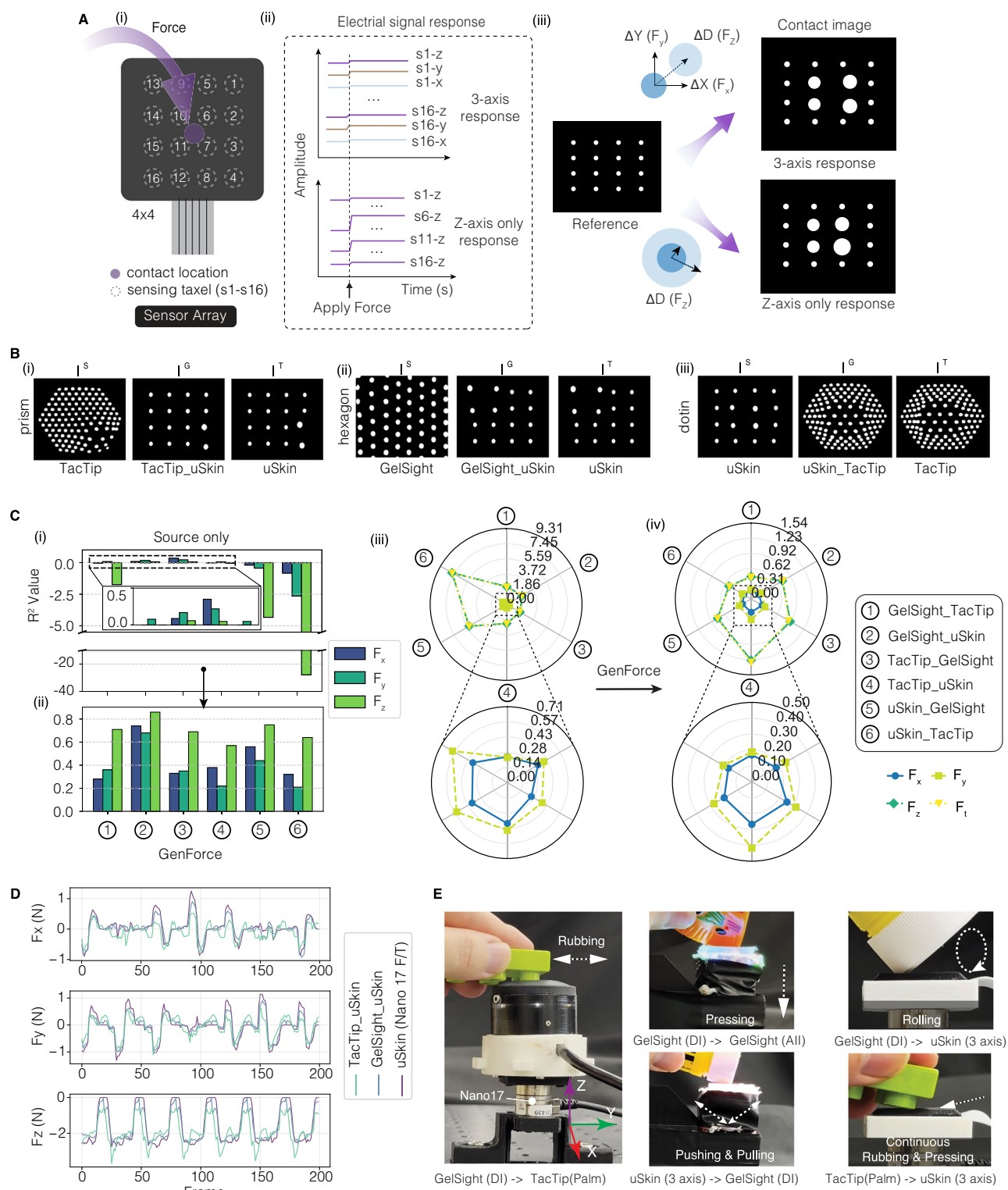

**Fig. 6 | Heterogeneous tactile force translation. A** Signal-to-marker conversion process. The multichannel electrical signals (ii) from a sensor array (i) are unified into marker images (iii) accommodating sensors with either 3-axis sensing capability or z-axis-only sensing capability. **B** Representative examples of tactile images generated after M2M translation. **C** Histogram of $R^2$ values (i-ii) and radar plot (iii-iv) of force MAE, comparing the source-only method against the GenForce

model with material compensation. **D** Real-time demonstration of force prediction with uSkin (3-axis) as the target domain, utilizing the GenForce model with material compensation. **E** Force prediction using GenForce model during dynamic contact events, qualitatively benchmarked against an ATI nano 17 F/T sensor (see Supplementary Video 2 and 3).

detection stages. We use three objects with different sizes and surface conditions from the YCB dataset: a strawberry (rough), a banana (moderately smooth) and an egg (smooth). External forces are applied by a human operator at the top position to induce slip. The robot

grasps objects using the mean normal force ($F_z$) from both sensors until it reaches 1 N. The robot detects slip if changes of shear force ($F_x$ or $F_y$) in either the TacTip or uSkin exceed 0.2 N. Upon detection, it compensates for the slip by narrowing the gripper width using a

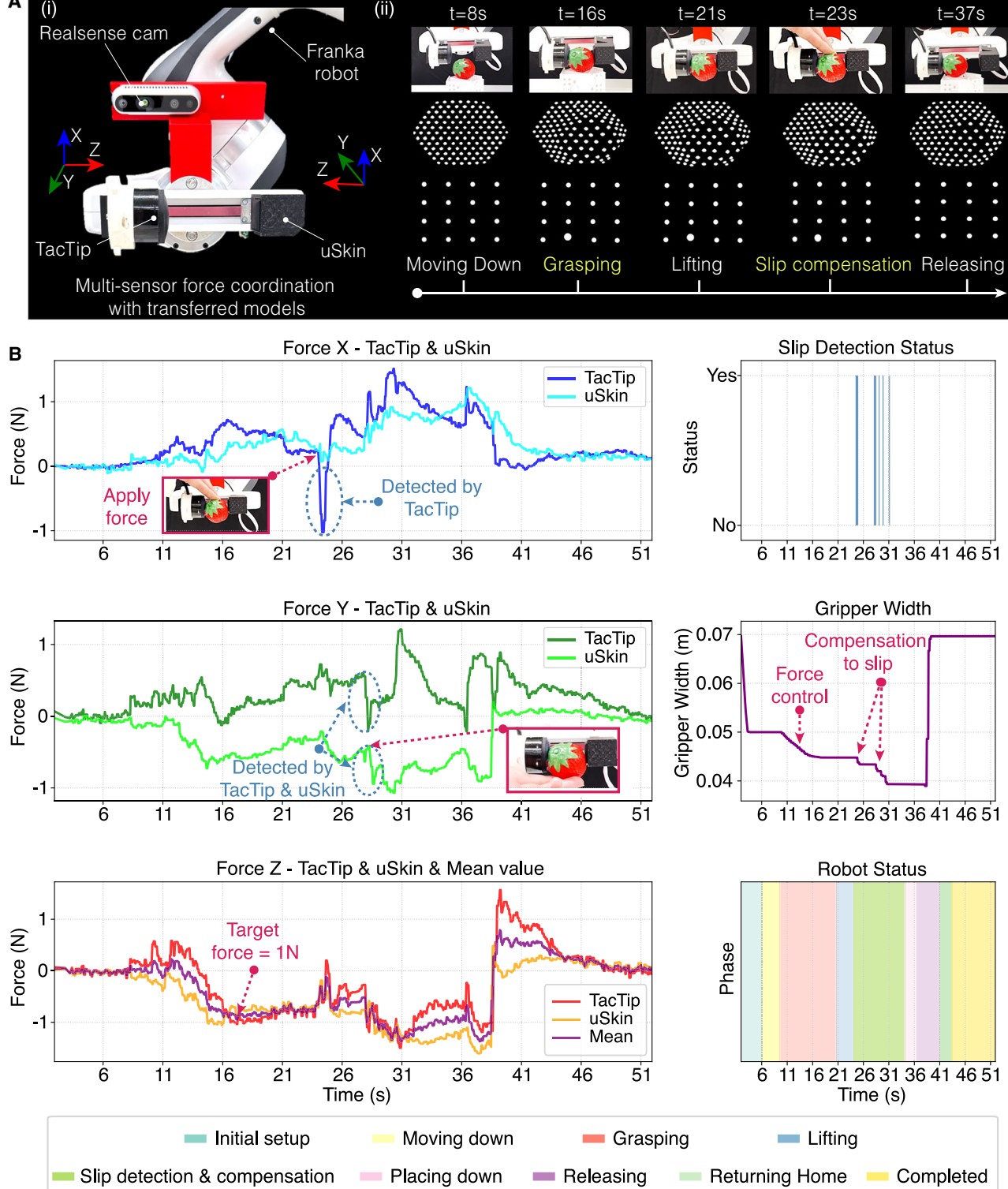

**Fig. 7 | Robot grasping and slip compensation with transferable force sensing.** **A** Multi-sensor force coordination enabled by transferred force models. (i) A Franka Hand equipped with a TacTip sensor on the left finger and a uSkin sensor on the right finger. (ii) Image sequence depicting the robot grasping a strawberry using force control, slip detection, and slip compensation. **B** Real-time measurements of forces, slip-detection status, gripper width, and robot state during the grasping task shown in (A-ii). Additional examples are provided in Supplementary Videos 4–7.

weighted sum of changes of shear force from both sensors (see Methods). As shown in Fig. 7B and Supplementary Video 7, the robot completes all stages successfully by coordinating the multi-sensor forces. By using this multi-sensor strategy, the mean normal force exhibits smoother profiles compared to those of a single sensor, which

is beneficial for stabilizing the grasping process against sudden changes of force in individual sensors. If one sensor indicates a significant change of force while the other remains stable, the controller can quickly identify slip; for instance, in Fig. 7B, the first slip event is detected by the TacTip but not by the uSkin. Conversely, when slip is

detected, the weighted sum of changes of force from both sensors enables an adaptive adjustment of the gripper width. We also provide additional demonstrations of this task with four objects (Supplementary Fig. 27C) in Supplementary Video 6 to illustrate force control using a single sensor. The experiments described above successfully demonstrate the practical applications of our GenForce model in robot manipulation, establishing a new paradigm for coordinating heterogeneous sensors on robotic hands via transferable force sensing.

## Discussion

In summary, this study addresses the fundamental challenge of transferring force sensing capabilities across diverse tactile sensors. While advances inspired by human mechanoreceptors have led to diverse bioinspired tactile sensors, few studies have focused on bridging these sensors or transferring tactile experience between them. This capability is critical for large-scale tactile sensing systems, where collecting labeled data for each individual sensor is impractical, yet the coordination of multiple sensors is essential.

Traditional paradigms for endowing tactile sensors with force sensing capabilities present significant limitations. For instance, Finite Element Method (FEM) approaches[40] rely on sensor-specific stiffness matrices and often assume linear deformation of soft elastomers[41]. While learning-based methods[17] excel at handling nonlinear responses, they necessitate costly force/torque sensors (e.g., Nano17) and extensive data collection for each new sensor. GenForce addresses these limitations by mimicking the human tactile memory system to unify, adapt, and transfer tactile experiences, enabling sensors to acquire force prediction capabilities by directly learning from one another.

Unlike recent methods[24–28] that focus solely on vision-based tactile sensors using learned latent representations, which typically require extensive data and are restricted to homogeneous sensors, our GenForce model operates directly at the tactile-signal level and is generalizable across diverse tactile sensors. By employing a unified marker representation, analogous to human sensory encoding, our method explicitly transfers skin deformation information between sensors through marker-to-marker translation and validates the transferable force sensing in robot manipulation tasks with multi-sensor coordination.

The key contributions of our study can be summarized as:

1. Cost-efficient and user-friendly: Our approach alleviates the need for expensive force/torque sensors when training force prediction models for new sensors. It only requires a small amount of location-paired data, less than 10% of the force-paired images used in our experiments, substantially reducing time and cost.
2. Foundational and Scalable: The marker-to-marker translation, based on an image-conditioned diffusion model, enables transfer of marker representations across a wide variety of sensors using a single model. Once translation is established between sensors, no additional data collection is required, making the model inherently scalable as sensor networks grow.
3. Generalizability: Our model generalizes across diverse tactile sensors, functioning independently of sensing principles, structural designs, or material properties. This is validated on simulated data with 132 sensor combinations and real-world data with 74 combinations, encompassing both homogeneous and heterogeneous translations.
4. Accuracy and Reliability: The force prediction errors are tightly centered around zero within the range of −4 N to 0 N in the normal direction and −3 N to 3 N in the shear direction across all transfer combinations. The model is also validated in real-time against ATI nano 17 F/T sensor under dynamic contact events commonly encountered in manipulation tasks.
5. Applicability: We demonstrate the practical utility of transferable force sensing in robot grasping and slip compensation tasks

involving multi-sensor coordination. The transferred force models exhibit robustness against environmental noise arising from sensor fabrication, data collection, marker conversion, and the translation process.

Despite its robustness, our model exhibits certain failure cases:

1. We observed a flicker effect in homogeneous translation when transferring to GelSight (A-I) (see Supplementary Fig. 45A and Video 1 for the AII_A-I group). This issue stems from the data collection process, where indenters with large contact areas, such as a hemisphere, cause an upward shift of the elastomer, disturbing the image style of A−I. However, this artifact does not compromise the subsequent force prediction performance of the AII_A-I group (see Fig. 4D–E).
2. Another challenge arises in heterogeneous translation (Supplementary Fig. 45B) when the converted markers are too small and too sensitive in the changes of marker size and marker displacement, making the M2M model hard to train. To mitigate this, optimized configurations are detailed in Supplementary Text 8 and Supplementary Fig. 12A.
3. In real-time deployment, the force prediction model exhibits an initial zero-shift during the first few seconds and a mild shift under static loading. The initial zero-shift is mitigated by baseline subtraction, whereas the static shift stems from the elastomer's intrinsic hysteresis. Additionally, when grasping metallic objects, the uSkin sensor may fail due to magnetic field interference. Nonetheless, manipulation tasks are successfully completed thanks to the gripper's heterogeneous sensor configuration (see Supplementary Videos 4–7).

Some limitations of our model and possible future works:

1. A prerequisite for the model is the need for at least one sensor fully calibrated with force-labeled data and material priors. Such a calibrated sensor can be sourced from our dataset or other existing sensors, followed by collecting a small set of location-paired tactile images for the target sensors. Regarding material priors, recent in-situ methods[42,43] offer an alternative to traditional characterization, circumventing the need for complex hardware such as F/T sensors or moving stages. Looking ahead, a pivotal direction is the development of a foundational tactile database like ImageNet[44] that eliminates the need for individual data collection and material calibration. Given our model's adaptability, such a dataset would be inherently scalable by leveraging our data collection process, enabling users worldwide to rapidly deploy high-performance force sensing.
2. As the transferred force prediction models are based on unsupervised learning, their accuracy may not fully match that of supervised learning methods trained on extensive labeled datasets. However, for applications demanding state-of-the-art accuracy, these transferred models can serve as robust pre-trained models that can be further fine-tuned with minimal data.
3. This approach has been designed and tested for tactile sensors that naturally produce outputs that can be expressed as marker-based or taxel-based 2D deformation patterns. Extending this framework to tactile sensors lacking explicit spatial taxels remains an open challenge, such as Electrical Impedance Tomography (EIT) sensors[45].

Beyond force sensing, our three key contributions, a unified marker representation, marker-to-marker translation, and a spatio-temporal regression model, can readily extend to broader robotics applications by simply modifying the output layer, offering immense potential value. Promising avenues include in-hand pose estimation[46], 3D reconstruction[14], and other tasks critical for dexterous manipulation. By establishing a unified framework for transferable sensing

across diverse tactile sensors, our work paves the way for cost-effective and scalable tactile sensing systems in next-generation tactile robots.

# Methods

## Model derivation

**Problem setting.** In a general case, we have one or more calibrated tactile sensors, referred to as source domains $\{S_i\}_{i=1}^{n}, n \in \mathbb{N}$ with paired tactile signal-force data $\{I^{S_i}, F^{S_i}\}$, while uncalibrated tactile sensors, referred to as target domains $\{T_j\}_{j=1}^{m}, m \in \mathbb{N}$, are without access to force labels $\{F^{T_j}\}$ for tactile signals $\{I^{T_j}\}$. The problem is how to leverage the collected force labels $\{F^{S_i}\}$ from $\{S_i\}_{i=1}^{n}$ to endow sensors $\{T_j\}_{j=1}^{m}$ with force prediction capability. Notably, the tactile signal can be tactile images or multichannel electrical signals.

**Overview.** Above problem setting raises a cross-sensor force calibration challenge in a many-to-many paradigm regardless of discrepancy in sensing principles, structure designs and material properties. A solution building upon such a problem setting is able to establish a foundation for transferable force sensing across various homogeneous (Fig. 2A) or heterogeneous tactile sensors (Fig. 2B). Our proposed GenForce model can be divided into three steps: (1) unified marker representation; (2) marker-to-marker translation; (3) spatiotemporal force prediction.

**Derivation of unified marker representation.** When forces are applied to tactile sensors (Fig. 1D), the soft skin deforms and the skin deformation is measured as different types of tactile signals, due to difference in sensing principles. The differences in physical configuration, such as illuminations and cameras in vision-based tactile sensors, or electrode layout and resolution in sensor arrays, lead to further differences in measured signals. To eliminate these input disparities similar to how human mechanoreceptors translate stimuli into sensory encoding, we unify the tactile signals from various tactile sensors as marker representation. Compared with other modality (see Supplementary Text 1), the marker representation unifies the tactile signals induced by skin deformations as marker deformation and displacement, ensuring a consistent image representation essential for cross-sensor translation by using image translation methods[47,48]. The marker pattern is common in tactile sensors. In vision-based tactile sensors, it can be physically designed on the soft skin, such as GelSight[13], DIGIT[49] or transferred from markerless skin[50] by using deep learning method (see Supplementary Fig. 38). While in electronic sensor arrays, the electrodes can also be regarded as another type of marker in either lab prototype[8] or commercial product[15], which can also be converted as marker images from multichannel electrical signals using our signal-to-marker technique (see Supplementary Fig. 11A–B).

**Derivation of marker-to-marker translation.** By unifying tactile representation as marker images, we can transfer the collected data across tactile sensors via image-to-image translation technique at the image level[47,48], as the differences now mainly lie in marker pattern. We enable this transfer across marker images through a marker-to-marker (M2M) translation technique (Fig. 2C) that bridges differences regarding marker size, density, and distribution. To align the image distributions of the source domains $p(I^{S_i})$ with target domains $p(I^{T_j})$, we train an image conditioned diffusion model $G(I_t^{S_i}, I_0^{T_j})$ to map $I_t^{S_i}$ to $I_t^{T_j}$ at deformed time step $t, t \in \mathbb{N}$ such that $G : I_t^{S_i} \rightarrow I_t^{T_j}$. The conditional inputs used here are the reference images $I_0^{T_j}$, captured when target sensors are in a non-contact state. Once the model $G$ is trained, $I_t^{S_i}$ can be translated into $I_t^{G_i}$, which matches the image styles of $I_t^{T_j}$ while preserving the deformation information of $I_t^{S_i}$. Note that, we only need to train one single model $G$ available to selectively transfer from sensor $i$ to sensor $j$ by using image conditions.

**Derivation of spatiotemporal force prediction.** Upon marker-to-marker translation, we can construct new datasets $\{I^{G_i}, F^{S_i}\}$ from existing datasets $\{I^{S_i}, F^{S_i}\}$ to train force prediction models $\hat{h}_{i \rightarrow j}$ for new sensors $T_j$, where we focus on three-axis force prediction $F = (F_x, F_y, F_z)$ in this paper. Specifically, we leverage force prediction models depicted in Fig. 2D to take spatiotemporal information into consideration. In addition, we consider the material hardness effect and formulate a material compensation method (see Fig. 5D). As such, force prediction models $\hat{h}_{i \rightarrow j}$ can now be trained with the transferred dataset $\{I^{G_i}, F^{S_i}\}$ and then predict forces $F^{T_j}$ with images $I^{T_j}$ from new sensors.

## Marker-to-marker translation model

The M2M model (see Fig. 2C) comprises two primary components: a marker encoder-decoder[51] and an image-conditioned diffusion model[52]. We adapt the variational autoencoder (VAE) architecture from SD-Turbo[52] as the marker encoder-decoder. The conditional diffusion model is based on the UNet[53] architecture from SD-Turbo combined with a DDPM Scheduler[54]. To optimize the model's performance while maintaining parameter efficiency, we implement Low-Rank Adaptation (LoRA)[55] to key network components, including attention layers, convolutional layers, and projection layers for efficient fine-tuning. The model is mainly trained with one NVIDIA A100 GPU with 80GB memory. We employ an 80-20 train-test split on our dataset. All images are preprocessed from 640×480 pixels to a uniform size of 256×256 and normalized to [0,1] range. We use AdamW optimizer with an initial learning rate of $5 \times 10^{-6}$, betas = (0.9, 0.999), epsilon=$1 \times 10^{-8}$, and weight decay of $1 \times 10^{-2}$. Details of the model architecture and training process are shown in the Supplementary Text 2. The ablation study is available in Supplementary Text 4.

## Spatiotemporal force prediction model

The spatiotemporal architecture has four components (see Fig. 2D): (1) a marker feature encoder from RAFT[56] that extracts spatial representations from sequential tactile images, (2) a spatiotemporal module using ConvGRU[56] that captures temporal dependencies while preserving spatial information, (3) a ResNet[57] module that hierarchically processes features through residual blocks, and (4) a regression head with a multilayer perceptron that transforms the processed features into force predictions. We use the same image preprocessing as M2M, while the forces are processed with min-max normalization. We train the model with sequential images with variable length. The training process follows a two-stage approach: first, we pre-train the model on with transferred data with a learning rate of 0.1 for 40 epochs, then fine-tune it with a learning rate of $1 \times 10^{-3}$ for another 20 epochs. We use SGD optimization with momentum of 0.9 and weight decay of $5 \times 10^{-4}$. The detailed model architecture and training process are demonstrated in the Supplementary Text 3.

## Simulation of marker deformation

This pipeline (Fig. 3A) enables the synthesis of deformed marker images similar to those from real-world sensors. The pipeline comprises three primary stages: (1) skin deformation simulation; (2) marker pattern design; (3) marker deformation rendering. We model the skin deformation simulation environment in Tacchi[58] where sensor dimension, material property, and contact trajectory are configured to generate deformed meshes (see Supplementary Fig. 1C) in varying locations. The simulation model comprises a $20 \times 20$ mm elastomer with a thickness of 4 mm. The elastic modulus is set as $1.45 \times 10^{5}$ Pa with a Poisson's ratio of 0.45. We employ 18 different primitive indenters (Supplementary Fig. 1A) following the contact trajectory shown in Supplementary Fig. 1B and the Supplementary Text 5 to get deformed meshes. Upon obtaining the deformed meshes, the meshes are imported into Blender for marker image rendering. Before rendering, we draw 12 types of reference marker images (see Fig. 3A) with

different marker patterns, density, and size using CAD software. For each mesh, we apply the reference marker image as the texture of the deformed mesh and use UV mapping specifically to surfaces with negative z-axis normal (see Supplementary Fig. 1C). The rendering environment is configured with a 35 mm focal length camera, oriented to capture the deformed surface. To ensure visibility of the marker patterns, we set the light strength to 60.0 units. The rendered outputs are 640 × 480 images. In total, we get 810 marker images per marker pattern and a total of 9720 images across all reference marker images. It is worth noting that this dataset is scalable by adding varieties of marker patterns, sensor dimensions, material properties, and contact trajectories.

### Fabrication of tactile sensors

We primarily utilize two types of vision-based tactile sensors: GelSight[13] and TacTip[12], both fabricated in-house. Related fabrication details for TacTip (palm shape, TacPalm) can be found in ref. [16]. Additionally, we incorporated an off-the-shelf magnetic-based tactile sensor, the uSkin (Patch version) from XELA robotics with of a 4×4 grid of taxels, where each taxel measures 3-axis deformation. The GelSight sensor integrates a Logitech C270 HD webcam, 3D-printed housing components (fabricated using Bambu Lab P1S), laser-cut acrylic boards (25 × 25 mm), LED lights with RGB color, and silicone elastomers. To achieve varying illumination conditions, we attach resistors to the RGB lights, producing three GelSight variants with different light colors: sensors with marker Array-II, Circle-II, and Diamond-I patterns. The sensors Array-I and Circle-I use the same hardware as Array-II and Circle-II respectively differing only in marker patterns with slightly random shift and rotation to replicate changes that occur on long-term use. For material compensation testing, we utilized white LEDs. The marker patterns used can be found in Supplementary Fig. 4A. Details for the soft skin fabrication can be found in Supplementary Fig. 2A and Supplementary Text 6.

### Data collection in real world

The data collection setup consists of five main components shown in Fig. 4A: a UR5e robot arm, a ATI Nano17 force/torque sensor, 3D-printed indenters, tactile sensors, and a 3D-printing support base. Each indenter is mounted on the robot arm with fixed initial positions and orientations. The contact trajectory for the real-world data collection is depicted in Supplementary Fig. 2B with parameters in the Supplementary Text 7. We choose five surface points on the skin. For each point, the indenters move towards different target points with varying depths and orientations in four actions: moving down, moving laterally, returning laterally, moving up. The movement speed was set to 25% for GelSight and uSkin sensors, and 40% for TacTip on the UR5e touch panel. For material compensation, we collected approximately 30,000 force-image pairs per skin for force transfer and 3,400 location-paired images per skin for M2M model finetuning. For heterogenous translation, approximately 60,000 force-image paired data are obtained per sensor, including 11,520 location-paired images.

### Marker conversion

For GelSight and TacTip, we use marker segmentation as illustrated in Supplementary Fig. 4B and Supplementary Fig. 11C. The marker segmentation process comprises a coarse-to-fine process. For GelSight, we execute rough extraction by combining the thresholding with Gaussian blur. For TacTip, we crop marker area from the raw RGB image and normalize brightness, followed by identifying markers using thresholding. In the fine extraction stage, we employ an EfficientSAM[59] model for both sensors to precisely segment individual markers by using the position information from the rough marks. For uSkin sensor we use signal-to-marker conversion (Supplementary Fig. 11A-B). The raw sensor data contains 16 three-dimensional vectors representing the taxel deformations. We convert the raw signals to marker images by using the parameters in the Supplementary Text 8.

### Material compensation

According to the relationship of force and indentation depth in Supplementary Text 9, we measure the force-normalized depth ($F - d_z$) relationship for homogeneous translation (Fig. 5B) and heterogeneous translation (Supplementary Fig. 15A). For each sensor, we conduct three indentations using the prism indenter (Supplementary Fig. 1A) with a rigid surface. The indenter is pressed down to a maximum depth $d_{max}$ of 1.4 mm and then released. We collect paired force-depth data during both loading and unloading phases according to the parameters in the Supplementary Text 9. We can then get the fitted lines of the $F - d_z$ curves (see Supplementary Fig. 9C, 15A-iii and 15A-iv), using two-degree polynomials. Then, the force label $F^S$ can be corrected to $F^{SC}$ with either the ratio of $r_L$ or $r_U$ depending on the contact is in loading phase L or unloading phase U according to the compensation process in the Supplementary Text 9.

### Robot grasping and slip compensation

We use a Franka FR3 robot arm with a Franka hand (maximum opening 0.08 m), an Intel RealSense camera, fifteen daily objects (nine for grasping and six for slip compensation), and three tactile sensors. For robot grasping in the first task, we implement a proportional control loop that modulates the gripper width based on normal force feedback from one sensor. This high-level controller runs at 10 Hz with a gain of $K_p = 0.0004$. During execution, the gripper closes at a slow speed of 0.005 m/s from an object-dependent initial width (ranging from 0.025 m for grapes to 0.075 m for oranges). The target normal force is adapted to the object's fragility and is set at 0.6 N, 0.8 N, or 1.2 N (e.g., lower forces for fragile items like chips and grapes, and higher forces for rigid items like a meat box). Specifically, the same grasping controller is used in the second task, but it utilizes the mean normal force from both sensors as feedback. For robot slip detection and compensation, the robot only detects slip at the top position with a controller runs at 20 Hz for 10 seconds, keeping a 0.5-second window with three historical force measurements. If the change of shear force from any sensors over the window exceeds $\tau = 0.2$ N, we tighten the grasp width $\Delta w$ by considering both sensors and command the changes of gripper width at a rate of 0.015 m/s:

$$\Delta w = \frac{\Delta w_{base}}{|N|} \sum_{i=1}^{N} \frac{\Delta F_i}{\tau} \tag{1}$$

where $\Delta w_{base} = 1$ mm is the base compensation width, $N$ is the number of sensors currently detecting slip (i.e., where $\Delta F_i > \tau$), $\Delta F_i$ is change of shear force of the $i$-th sensor.

## Data availability

All data supporting the findings described in this manuscript are available in the article and the Supplementary Information. Source data have been deposited in the Figshare database at https://doi.org/10.6084/m9.figshare.28842896[60].

## Code availability

All code and machine learning model used in this study are available in the Github at https://github.com/Zhuochenn/GenForce_Code[61].

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

## Acknowledgements

This work is supported in part by EPSRC project "ViTac: Visual-Tactile Synergy for Handling Flexible Materials" (EP/T033517/2, S.L.). We thank King's College London (2025), King's Computational Research, Engineering and Technology Environment (CREATE) for providing HPC.

## Author contributions

Z.C. and S.L. conceived the systematic study. Z.C. proposed the model, implemented the code, fabricated the sensors, collected the data, wrote the manuscript and created the figures. J.D. and N.O advised on model implementation and marker segmentation. X.Z. advised on marker simulation and provided TacTip sensor. Z.W., Y.Z., and Y.W. contributed to data collection. E. S. P. funded the Franka FR3 robot arm and supervised experiments regarding robot manipulation. N.L. funded the TacTip sensor and supervised experiments regarding TacTip. L.J. funded the uSkin sensor and supervised experiments regarding uSkin. J.D. and S.L. supervised, funded and managed the research, while providing hardware and computing resources. All authors helped revised and polished the manuscript.

## Competing interests

The authors declare no competing interests.
