## [Transparent Peer Review file · Nature Communications]

Training Tactile Sensors to Learn Force Sensing from Each Other

Corresponding Author: Professor Shan Luo

A version of this paper was originally rejected for publication by Nature Communications, however that decision was reconsidered after appeal by the authors.

Version 0:

Reviewer comments:

Reviewer #1

(Remarks to the Author)

This paper proposes a universal force sensing framework called GenForce for achieving transferable force perception between both homogeneous and heterogeneous tactile sensors. This work unifies the inputs of three different tactile sensors by constructing binary marker images, and achieves unsupervised force label transfer to another sensor by using a marker-to-marker translation module to transfer the deformation style from the source sensor to the target sensor.

My primary concerns are as follows:

1. **Method Design and Experiments on Material Compensation:**

This work collects and fits the "loading" and "unloading" force–depth curves of both the source and target sensors to empirically compensate for force prediction when transferring to materials with different hardness levels. However, a more detailed explanation is needed regarding how the method compensates for forces when the object moves laterally across the sensor surface. Does the material compensation account only for the vertical force F_z ? Furthermore, in Figure 5, is the consistently low prediction error in the x and y directions due to the similar friction coefficients of these homogeneous sensors? If so, how would variations in friction coefficients affect the transferability of shear force prediction? The authors should include more experimental results to support this point.

2. **Efficiency of Unified Marker Representation:**

In this work, all tactile sensor inputs are segmented or directly converted into binary marker images and processed by convolutional networks. However, would directly inputting the marker coordinates (and depth, if available) be a more efficient and accurate approach? Can this unified marker representation efficiently handle dense tactile 3D point clouds (e.g., GelSight Mini), or is it limited in that regard? Is this method unable to directly handle vision-based tactile sensors that do not have marker points? The authors should provide more detailed experiments and analyses.

3. **Real-world Robotic Applications:**

I recognize the great potential of this work in various tactile tasks. It would be even better if experiments and analyses demonstrating its application in real-world slip detection or robotic manipulation were provided.

4. **Quality of Figures:**

All the figures in this paper appear blurry when enlarged and should be replaced with vector graphics.

5. **The Rigor of the Biological Inspiration:**

This work draws inspiration from the human ability to transfer tactile sensations between the fingers and the palm, aiming to illustrate that humans can achieve generalized force perception through information sharing across different sensory

receptors. However, the skin on the human fingers and palm should belong to the same tactile system and represents a scenario of "homogeneous" sensors, rather than a transfer between heterogeneous sensors. The authors should revise this part of the Introduction with greater rigor.

Overall, this work systematically integrates different sensors in the force prediction context and demonstrates practical value across various robotic and tactile tasks, providing valuable insights for future research.

(Remarks on code availability)

The updated code provides a detailed README file to elaborately introduce the procedure of Data Collection, Model Training. However, I did not try to install and run the code.

Reviewer #2

(Remarks to the Author)

The paper presents a pipeline to generate tactile responses and corresponding force profiles across tactile sensors. The pipeline includes i) a module to obtain a binary marker-based representation of the tactile sensor responses, ii) a M2M module, based on latent diffusion models and trained in simulation, that, using a few paired reference images from a calibrated sensor, converts the marker-style of one sensor into another, iii) a module that, given the translated marker images, trains a spatiotemporal force-prediction model to estimate 3-axis forces (x, y, z) from the tactile sequence.

The work targets a very important problem faced by the tactile sensors' community related to the generation of tactile datasets since their collection is time-consuming and they are generally not reusable across tactile sensor technologies due to differences in the sensor hardware. The problem of generating tactile images across different tactile sensor technologies has already been presented in previous papers cited in this work, where the transfer has been done between a camera based tactile sensor and a capacitive based tactile array. With respect to the cited paper, this work faces a simpler problem when generating tactile dataset across sensors modalities, as the sensors they selected (camera-based sensors and the magnetic-based sensor) are all providing both normal and shear components and comes with marker-based representations, however they present a modular architecture that enable generalisation within the marker-based classes they are focusing on as well as the generation of the force profiles. The amount of analysis performed cover also a wide range of different cases.

The paper is structured such that it is possible to easily follow all the steps of the pipeline but could be improved to achieve more clarity as some parts are dense of details which are also repeated in the supplementary materials. Some parts of the paper clearly look like they were rewritten by a chatbot. While the use of AI tools for improving the English for not native speakers might be acceptable, the use of chatbot should be disclosed explicitly as per the journal's guidelines.

Comments on the stated contribution claims:

The claim of eliminating the need for expensive force/torque sensors seems overstated as there is the need to calibrate at least one of the sensors.

While the architecture has been designed to be modular and extendable, the model assumes a specific marker structure (circular, optical, regular) across sensors and would probably require some adaptation as well as new calibrated paired data for sensors that are structurally different (e.g., capacitive and resistive).

Your model has been trained and evaluated only on marker-based tactile images coming from simulation and real-world. This type of representation is common within the space of optical tactile sensors and magnetic based tactile sensor arrays which have been proposed in this work.

Evidence that it can generalize to fundamentally different modalities, such as capacitive arrays (no markers, no shear force), piezoelectric or resistive pressure sensors have not been presented.

Furthermore, the whole architecture M2M +GenForce looks like dependent on "materials-priors" which cannot generalize well to new materials not seen during M2M training, account for ageing or continuous variation in the stiffness (e.g., slightly stiffer vs. much stiffer materials). For these reasons, this claim appears to be an overstatement and to support it the study should include at least a sensor which is fundamentally different or state clearly the limit of this generalizability.

The +/-4N range is suitable for many robotics application like grasp, push, pull and the error in that range is impressive. This good result has been obtained thanks also to the material compensation step that address differences in the elastomers mechanical properties, however the results compare force predictions across sensors (which shows consistency) rather than against a trusted external ground-truth force sensor (like ATI F/T). This is necessary for claiming accuracy which is important for the control.

Methodology:

- The architecture is explained in detail, but there is the lack of critical ablation studies which would enable to understand the role of each element in the pipeline in the overall performance of the system and how to change parameters to extend the work. This limits the reproducibility and the extendibility of the presented work.

- There is not a systematic analysis on when the model would fail and in general the limitations of the work are not discussed.

- The whole work is motivated by providing robots the ability to unify and transfer sensory information across multiple fingers ensuring stability and dexterity during object manipulation, however a real-world manipulation task where such capabilities, enabled by the presented architecture, are evaluated is not presented. The whole work stops by demonstrating the generation of the tactile marker-based images and the force estimation. Online performances are not discussed a part what can be inferred from the videos (which is the frequency at which we can get the force estimation for example) so it is not clear if this can be really used for control.

- In the Discussion section it is mentioned large-scale tactile sensing. When integrated on a robotic hand the sensor might need to adapt to non-flat surfaces, however the presented work doesn't seem to address this aspect. It would be better to clarify this.

- Data collection in the real world, why the movement speed is different between GelSight and uSkin and TacPalm? Also, what is the maximum speed that can be achieved? The data collection in the real world is impressive, how long it took to collect all the data? This looks contradicting the purpose of the system to reduce the need of tactile dataset acquisition.

- Relationship of force and indentation depth: why a linear model is used? This might be the reason why the accuracy is limited to the 4N.

- Material compensation process: what are the material priors you are referring to? Is the young's modulus?

- Real- world data for the force estimation model have been acquired with controlled indentations at predefined locations and depths involving static or slowly varying contacts. How this architecture performs in case of more dynamic contact events typical for manipulation tasks?

Minor:

- The term "Labor-friendly" is usually used to indicate about someone or something supportive of workers' welfare and rights, it doesn't seem the appropriate adjective in this case.

- The only biomimetic sensor in the study is the TacPalm while the text looks like referring to all of them as biomimetic.

- Please proofread again the text for more appropriate words ("disparity" among sensors, "from seen indenters") and for typos (e.g. corase-to-fine process..).

- Even if the generated marker-based images are not linked to any specific sensor technology It would be good to have an image that shows generated marker-based images vs real marker-based images associated to a specific sensor to check noise or error. It is difficult to understand how well they are w.r.t real world sensors as there are not real-world tasks.

- When referring to tactile memory and transfer there is a specific reference to "use index finger memory to interpret palm stimulus" in the text, however, please note that tactile memory is not strictly localised so I would avoid being so specific as this level of granularity of spatial memory transfer is not proven.

- Figure 1A is not very informative.

- Figure 5D is very difficult to read as it is too small

- The videos are low resolution (especially the written text). It is not clear in the video which sensor is represented in the RGB image and marker-based image, is it the groundtruth?

(Remarks on code availability)

Version 1:

Reviewer comments:

Reviewer #1

(Remarks to the Author)

Thank you to the authors for the improvements in shear force compensation, visuo-tactile sensor transfer, real-world experiments, and the motivation explanation, which make this work more complete. My main concerns have been addressed, but there is still a minor typo:

There is a symbol error in Equation (15) — it has two $F^{\{SC\}}_{\{Shear\}}$.

(Remarks on code availability)

Reviewer #2

(Remarks to the Author)

Thank you for the substantial revisions and the extensive experiments added to the manuscript and supplementary material to address the reviewer's comments. The manuscript has evolved substantially, and the idea of creating a shared tactile representation for cross-sensor force prediction is interesting and timely. Your expanded experiments and supplementary material help clarify several aspects of the method. At the same time, a number of elements would benefit from further clarification, restructuring, or moderation to ensure the contribution aligns fully with the evidence presented. The suggestions below are offered to help strengthen the paper in this direction.

1) To address the reviewer's concern about the claim that the method allow generalization "across tactile technologies," it has been included in the Discussion section, a statement suggesting that the approach could extend to any tactile sensor by dividing the sensing surface into artificial "taxels" to produce a marker-like displacement field. Since this idea is not demonstrated experimentally, and may not hold for sensors whose outputs do not encode surface deformation in a physically meaningful way, it would be helpful to frame this more conservatively. The evidence in the paper shows that the method works well for sensors that "naturally" produce marker-based or taxel-based 2D deformation patterns (e.g., GelSight, TacTip, uSkin). For tactile technologies that do not exhibit such outputs (e.g., resistive textile based sensors, whisker sensors, EIT sensor, etc), the feasibility of generating an equivalent marker representation remains untested. Moderating the claim explicitly stating that the approach is designed for sensors with this family of outputs, would make the contribution clearer and more aligned with the demonstrated results.

2) The revised manuscript is clearly well-polished, and the use of AI-based tools for grammatical refinement is entirely acceptable. However, some resulting terminology (e.g., "slip avoidance," "short-term shear variation") does not align with established vocabulary in tactile sensing or robotic manipulation. Slip detection, slip compensation / grasp stabilisation, and rate of change of shear force constitute better terminology. Please ensuring that the wording remains consistent with the terminology used in the robotics community to improve clarity and make the experimental descriptions easier to interpret.

3. Several main figures, particularly Figs. 4–7, are dense, and hard to read due to the small size, making it difficult to understand key components without consulting the supplementary material. As a result, it is difficult to understand the full meaning of the results without repeatedly referring to the supplementary figures, which contain essential information. This constant switching between documents interrupts the flow of reading and makes it harder to follow the narrative. I would suggest a reorganisation of the figures, bringing the most important supplementary elements into the main figures and enlarging the size to make the main text more self-contained and improve the readability.

3. In Fig. 7, only TacTip's predicted forces are shown. It is therefore difficult to determine how uSkin contributes to grasp control, or how the two sensors' outputs are combined. Since the Panda gripper uses a single actuator, explaining precisely how slip detection is triggered and how both sensors influence the control loop would make the demonstration stronger and easier to interpret. Including uSkin's predicted forces in the plot would help support and validate the multi-sensor aspect of the experiment.

4. The dynamic-contact results in Supplementary Figs. 18–24 provide helpful qualitative insight, but the human-operated trajectories are not repeatable and cannot serve as quantitative demonstrations of dynamic performance. Clarifying that these results are qualitative and avoiding strong claims based solely on them would make the interpretation more consistent with the presented evidence.

5. Thank you for your detailed comparison with [30]. Some of the points you raise, particularly the force-prediction capability, are very useful to understand the novelty of your method. However, several other entries in Supplementary Table 1 are presented only conceptually, since no experimental evidence is provided to support them. It would strengthen the manuscript to adjust the wording so that the comparison reflects only what is directly demonstrated. More generally, I am not convinced that the table is necessary; a clear and focused supporting explanation in the text could convey the same information without requiring a separate table.

6. Please provide a clear definition of the "source-only" baseline, as it is referenced multiple times. Readers who are not already familiar with this term may have difficulty understanding the comparisons without an explicit description.

I hope these suggestions are helpful to improve the final version of the paper.

(Remarks on code availability)

Response Letter

We thank the reviewers for their helpful and constructive comments on our manuscript, which have significantly improved the quality of our work. Below, we have carefully considered all the comments from reviewers and provided detailed responses to each point. The *reviewers' comments* are in *italics* text and *blue color*, and **our responses** are in **normal** text and **black color**. The **revisions** in the **revised manuscript and Supplementary Information** are in **red color**. The **highlight** uses **bold font** and **yellow background**.

Responses to Comments from Reviewer# 1

Reviewer #1 (Remarks to the Author):

This paper proposes a universal force sensing framework called GenForce for achieving transferable force perception between both homogeneous and heterogeneous tactile sensors. This work unifies the inputs of three different tactile sensors by constructing binary marker images, and achieves unsupervised force label transfer to another sensor by using a marker-to-marker translation module to transfer the deformation style from the source sensor to the target sensor.

Response: We sincerely thank the reviewer for the thorough evaluation of our manuscript. Thank you for acknowledging the main contributions of our GenForce model, which achieves unsupervised, transferable force prediction between tactile sensors by unifying tactile signals into marker images and leveraging a marker-to-marker translation and a temporal force prediction model. In response to your constructive comments, we have **substantially revised and strengthened the manuscript**. We have **added new experiments** and **detailed analyses** to address each concern. We also added **new real-robot experiments with force sensing and control** to **strengthen the applicability of our work**. **In total, we revised the manuscript with 7 main figures, 43 supplementary figures, 4 supplementary tables, 7 supplementary videos (including 5 new videos and 2 videos remade from the previous 8 videos), and 1 project page (Link)**. All textual changes are highlighted in red in the revised manuscript, and every point is addressed in the detailed, point-by-point response below, with explicit pointers to the relevant pages, lines, and figure, table, video references.

My primary concerns are as follows:

*1. ****Method Design and Experiments on Material Compensation:*****

This work collects and fits the "loading" and "unloading" force–depth curves of both the source and target sensors to empirically compensate for force prediction when transferring to materials with different hardness levels. However, a more detailed explanation is needed regarding how the method compensates for forces when the object moves laterally across the sensor surface. Does the material compensation account only for the vertical force F_z ?

Response: Thank you for the constructive comments regarding the method design and the experiments related to material compensation. We apologize for not clearly explaining the compensation of shear force in our original manuscript. In the revised manuscript, we have

added a theoretical formulation to provide a more comprehensive explanation of force compensation, especially for shear force. The main modifications in our Supplementary Information are as follows.

Main modification in our Supplementary Information can be found as follows.

(Line 298 - 419) “

Text 9. Material Compensation

9.1 Relationship of Normal Force and Elastic Modulus

According to contact mechanics, when a flat rigid indenter applies normal force F_{normal} in the z direction on an elastic specimen's surface⁶⁶, the relationship between force F_{normal} and indentation depth d_z is given by:

$$F_{normal} = \alpha E^* d_z \quad (6)$$

where α is a geometric constant specific to the indenter, and E^* represents effective modulus, i.e. combined modulus of the indenter and the specimen.

For multi-material contacts, the effective modulus satisfies,

$$\frac{1}{E^*} = \frac{1-\nu^2}{E} + \frac{1-\nu'^2}{E'}, \quad \frac{1}{G^*} = \frac{2-\nu}{4G} + \frac{2-\nu'}{4G'}, \quad G = \frac{E}{2(1+\nu)} \quad (7)$$

Where E' and ν' , and E and ν , describe the elastic modulus and Poisson's ratio of the indenter and the specimen respectively. In the context of contact of a rigid indenter with a soft elastomer, the effective modulus E^* and G^* are determined by the elastomer E and G ⁶⁶.

Based on Equation (6), we can derive that in a fixed contact depth:

$$F_{normal} \propto E \quad (8)$$

9.2 Relationship of Shear Force and Elastic Modulus

When considering contacts which are loaded both in the normal direction z and in the tangential direction x and y , it is the ‘‘Cattaneo–Mindlin problem’’⁶⁷. We consider the solution of the tangential contact problem using the Method of Dimensionality Reduction (MDR)⁶⁷. The shear force F_{shear} is described by a rigid lateral shift $u^{(0)}$ of the rigid indenter over an elastic base characterized by effective modulus E^* (normal) and G^* (shear). The contact has outer radius a ; the inner stick region extends to radius c and the outer annulus ($c < x < a$) slips region.

By MDR superposition, the total shear force is the sum of a stick-core term and a slip-annulus term,

$$F_{shear} = F_{stick} + F_{slip}, F_{stick} = 2cG^* u^{(0)}, F_{slip} = \mu[F_{normal}(a) - F_{normal}(c)]. \quad (9)$$

Here μ is the Coulomb friction coefficient and $F_{normal}(a)$ is the normal force associated with a contact of radius a . The MDR problem provides the normal force as

$$F_{normal}(a) = 2E^* \int_0^a [d(a) - d(x)] dx \quad (10)$$

Where indentation depth $d(x) = g(x)$, $g(x)$ is the shape profile of the indenter in a one-dimensional.

The stick–slip interface is determined by

$$u^{(0)} = \mu \frac{E^*}{G^*} [d(a) - d(c)] \quad (11)$$

which gives the stick–slip boundary with radius c .

Eliminating $u^{(0)}$ from the decomposition using the displacement criterion yields a compact expression for the shear force:

$$\begin{aligned} F_{shear} &= 2cG^* u^{(0)} + \mu[F_{normal}(a) - F_{normal}(c)] \\ &= \mu\{[F_{normal}(a) - F_{normal}(c)] + 2cE^*[d(a) - d(c)]\} \end{aligned} \quad (12)$$

Because $F_{normal}(\cdot) \propto E^*$ and $d(\cdot)$ is purely geometric, both bracketed terms in the compact expression are linear in E^* . The shear force approximately satisfies,

$$F_{shear} \propto \gamma E \quad (13)$$

Where γ is the factor related to the difference in geometry g , imposed tangential shift u , modulus ratio E^*/G^* (effective Poisson ratio) and friction coefficient μ . The effective modulus E^* and G^* are determined by the elastomer E and G similar to normal force.

9.3 Compensation for normal force and shear force

Base above relationship, to compensate the normal force, we can approximately multiply the existing force label F_{normal}^S from the source sensor S with the ratio of modulus to get compensated force label F_{normal}^{SC}

$$F_{normal}^{SC} \approx \frac{E^T}{E^S} F_{normal}^S \quad (14)$$

Regarding shear force F_{shear}^S , the shear force can be compensated:

$$F_{shear}^{SC} \approx \gamma \frac{E^T}{E^S} F_{shear}^S \quad (15)$$

where friction weighting factor γ is set as a hyperparameter.

Specifically, the compensation for differences in modulus $\frac{E^T}{E^S}$ for both normal force and shear force happens when training the force prediction model. The corrected force label $F^{SC} = \{F_{normal}^{SC}, F_{shear}^{SC}\}$ can be obtained when loading the force-image pair data. Specifically, we use contact depth d_z to index force f_z^S and f_z^T from the curves of normal force - depth in both the source sensor S and the target sensor T . Note that, the indentation depth is normalized for its easy to comparing sensors with different indentation depth due to size and material differences. The compensation ratio r can then be calculated in an incremental form:

$$r = \frac{f_z^T}{f_z^S} - 1 \quad (16)$$

We introduce two additional hyperparameters: starting depth d_0 ($0 < d_0 < d_{max}$) and correction weight λ ($0 < \lambda < 1$) to control the amount of compensation. The compensation starts when the contact depth d_z exceeds d_0 . λ controls the compensation weight. These parameters used in our paper are obtained via grid search (see Supplementary Table 1 and Supplementary Table 2). Thus, the corrected force label F^{SC} can be derived as:

$$F^{SC} = F^S \cdot (1 + \lambda r) \quad (17)$$

Where r is r_L in loading phase while r_U in the unloading phasing indexed with the contact location d_z to catch the hysteresis property of elastomer:

$$r = \begin{cases} r_L, & \text{if } d_z > d_0 \text{ and } d_z \in L \\ 0, & \text{if } d_z \leq d_0 \\ r_U, & \text{if } d_z > d_0 \text{ and } d_z \in U \end{cases} \quad (18)$$

9.4 Choice of Material Priors

We introduce the choice of the two material priors, including elastic modulus E and friction weighting factor γ .

To get the E if not knowing the material property, we can choose below ways to acquire depending on the experimental conditions: (1) traditional materials characterization method with high accuracy but need to dismantle the sensor or break the elastomer; (2) in-situ calibration using force gauges and indenters without dismantling the sensor and breaking the elastomer⁴³; (3) in-situ mechanical calibration method without use of force/torque sensors and the need for a robotic arm with a low-cost calibration devices⁴⁴. In our test, the elastomer has non-linear mechanical property, we estimate E by measuring the relationship between applied normal forces F_N and normalized indentation depth d_z using a rigid indenter (*prism*, contact area 5×7 mm) referring to equation (6). We vertically move the indenter from a non-contact position to place near the maximum depths of sensors: GelSight (1mm), uSkin (1mm), TacTip

(4.5mm). We measure the force response both in loading and unloading stage in different indentation depth due to hysteresis property. Then, we normalize indentation depths to the range of 0 to 1 so that those images collected in a step of 0.2 mm (25% of maximum depth 1mm) in GelSight and uSkin can be paired with TacTip with a step of 1.125mm (25% of maximum depth of 4.5mm). The curves are then fitted with a two-degree polynomial using the mean and variance values during three indentations shown in Supplementary Figure 8C and 13A.

However, friction weighting factor γ is hard to calibrate which involves many variables. Here, we provide empirical factors for γ only using in the real-time deployment stage to compensate the differences in geometry g , tangential shift u , modulus ratio E^* / G^* (effective Poisson ratio) and the friction coefficient μ . In the training stage, we only compensate for the shear force with E and set γ as 1, which is verified good performance in improvement of shear force in our experimental results.

To further understand the complexity of compensating shear force and the choice of our empirical factors, we exam the statistical relationship between shear force F_S and shear displacement δ (0-1mm) from both homogeneous sensors (see Supplementary Figure 8B) and heterogeneous sensors (see Supplementary Figure 13B). We use the *sphere_s* indenter (Supplementary Figure 1A, diameter of 8mm) to apply forces in different depth and direction by referring to our data collection trajectory. This resulted curve (Supplementary Figure 8B) matches the relationship shown in [68] with shear and slip stages. This curve can be further processed to the relationship between F_S / F_N and shear displacement δ (Fig. 5D), which implies the friction coefficient μ between the contact pair of indenter and elastomer. From the curves of $F_S - \delta$ (Supplementary Figure 8B) in homogeneous sensors, we can see that even in the same shear displacement, the F_S varies due to difference in hardness although we use same materials for the pigment layer on the surface. That explains the necessity of compensate $\frac{E^T}{E^S}$ in the shear force. On the other hand, the mean friction coefficient μ varies from 0.4-0.7 (see Fig. 5D and Supplementary Figure 14A) if we look at the slip region [68] in displacement of 0.6-0.8mm in curves of $F_S / F_N - \delta$. The variation range expands from 0.3-0.8 (see Fig. 5D) when considering the standard variation. Similar mean friction coefficient μ (0.5-0.7) and variation range (0.3-0.9) can be found in heterogeneous sensors (see Supplementary Figure 14B and Supplementary Figure 13B). This variation further validates that the shear force is further affected by differences in geometry, hardness et.al. Hence, to compensate γ , we empirically choose friction weighting factor γ as 2 for homogeneous translation while 3 for the heterogeneous translation by referring to the variation range of friction coefficient.

References:

43. Zhao, C., Ren, J., Yu, H. & Ma, D. In-situ Mechanical Calibration for Vision-based Tactile Sensors. in 2023 IEEE International Conference on Robotics and Automation (ICRA) 10387–10393 (2023).
44. Li, M., Zhang, L., Zhou, Y. H., Li, T. & Jiang, Y. EasyCalib: Simple and Low-Cost In-Situ Calibration for Force Reconstruction With Vision-Based Tactile Sensors. IEEE Robot Autom Lett 9, 7803–7810 (2024).
66. Fischer-Cripps, A. C. Introduction to Contact Mechanics. (Springer New York, NY, 2010). <https://doi.org/10.1007/978-0-387-68188-7>.
67. Popov, V. L., Heß, M., & Willert, E. Handbook of contact mechanics: exact solutions of axisymmetric contact problems. Springer Nature (2019).
68. Yuan, W., Li, R., Srinivasan, M. A., & Adelson, E. H. Measurement of shear and slip with a GelSight tactile sensor. In 2015 IEEE International Conference on Robotics and Automation (ICRA) 304-311 (2015).

Furthermore, in Figure 5, is the consistently low prediction error in the x and y directions due to the similar friction coefficients of these homogeneous sensors? If so, how would variations in friction coefficients affect the transferability of shear force prediction? The authors should include more experimental results to support this point.”

Response: Thank you for your comments on the low prediction error for the shear force. We apologize for not clearly explaining the surface conditions of the elastomers used in Fig. 5 in our original manuscript. The elastomers used in **Fig.5** were fabricated from the same material, XPA565 (base A and activator B), but with different mixing ratios. Although the materials used for the pigment layers (i.e., the outermost layers) are the same, these elastomers exhibit different friction coefficients. Specifically, we measured the relationship between F_S / F_N and shear displacement δ (0-1mm) for seven elastomers at different surface points and depths using our data collection trajectory. As shown in **Fig. 5D** and **Supplementary Figure 14**, the mean friction coefficient varies from 0.4 to 0.7 in the steady region with a lateral displacement of 0.6–0.8 mm, which corresponds to the slip state [68]. The coefficient range further expands to 0.3–0.8 when considering the standard deviation, due to changes in material hardness, contact depth, movement directions, and contact positions.

To clearly demonstrate how variations in friction coefficients affect the transferability of shear force prediction, we have included additional experimental results and conducted a detailed analysis, as you suggested. As shown in **Fig. 5E(ii) and F(ii)**, the mean shear-force prediction errors before compensation, across all transfer groups, are below 0.2 N (test range: –3 N to 3 N; see Supplementary Figure 9A), owing to the strong performance of our model. The shear-force errors generally increase with larger gaps in hardness and friction coefficients. After applying our material compensation, the prediction errors in the shear direction decrease further. The improvements are most pronounced in groups with large gaps in hardness and friction coefficients, such as *6_14*, *6_16*, *6_18*, and *18_6*. From **Fig. 5G**, we see that the material compensation alleviates the underestimation of force caused by differences in hardness and

friction coefficients. This verifies the transferability of our model across variations in friction coefficients and hardness.

In addition, we tested the friction coefficients among heterogeneous sensors, as shown in **Supplementary Figures 13 and 14**. Similar mean friction coefficients μ (0.5-0.7) and variation range (0.3-0.9) were observed. Improvements in shear force prediction among heterogeneous sensors are also evident in **Fig. 6E**. To verify the accuracy of shear force and normal force in real-world dynamic events, we conducted new dynamic force experiments with daily objects (**Supplementary Figure 17**), compared in real time against ATI Nano17 F/T sensors. As shown in **Supplementary Figures 18–22 and Supplementary Video 3**, all transfer groups demonstrated impressive performance in shear force in terms of accuracy, consistency, and fast response, further verifying the method’s applicability in real-world scenarios.

Main modification in revised manuscript can be found as follows.

(Line 200-227) “

We divide the transfer direction into two groups: (1) *hard-to-soft* and (2) *soft-to-hard*. In **Fig.5E-i**, combinations like *6_8*, *6_10*, and *6_12* indicate progressively transfer to harder elastomer (slightly harder, harder, much harder), whereas *8_6*, *10_6*, and *12_6* indicate the inverse, i.e., softer transfer. These combinations reflect the long-term aging behavior of silicone elastomers. Before material compensation, normal-force errors generally grow with increasing hardness gap (**Fig. 5E**). On average, the normal-force error is 1.41 N for *hard-to-soft* and 1.03 N for *soft-to-hard*. For shear forces, average errors are 0.18 N (x) and 0.20 N (y) in *hard-to-soft*, and 0.16 N (x) and 0.18 N (y) in *soft-to-hard*. Shear errors are more stable across both groups, because applied shear forces are smaller than normal forces and less sensitive to hardness gaps (see Supplementary Text 9A). Additional results for 15 other translation combinations are provided in Supplementary Figure 8D–E.

To mitigate the increased force errors due to material hardness gaps, we propose a material compensation method (**Fig. 5B**) that uses material priors to correct force labels before training. The priors are the fitted elastic modulus profiles, i.e force-normalized depth profiles shown in Supplementary Figure 8C. Intuitively, this process increases the amount of force when transferring from soft to hard skin and decreases it in the opposite direction. As shown in **Fig. 5E-F**, this improves accuracy across both normal and shear forces. After compensation, average normal-force error drops to 0.99 N in the *hard-to-soft* group (30% reduction) and 0.87 N in the *soft-to-hard* group (16% reduction). For shear forces, average errors in the *hard-to-soft* group reduce to 0.10 N on the x-axis (44% reduction) and 0.14 N on the y-axis (30% reduction); in the *soft-to-hard* group, errors reduce to 0.13 N (F_x , 19% reduction) and 0.15 N (F_y , 19% reduction). The method yields lower errors in 95% of all combinations within the *hard-to-soft* group (including those in Supplementary Figure 8D) and in 57% of the *soft-to-hard* group. A representative case, transferring from r18 (soft) to r6 (much harder) skins in **Fig. 5G**, shows improvements in R^2 values from 0.73 (F_x), 0.79(F_y), and 0.78 (F_z) to 0.87, 0.86, and 0.84 across the three axes. Real-time visualizations in Supplementary Figure 9B and Supplementary Video 2 further illustrate the effectiveness of the material compensation, which is ignored by traditional studies while is very important in real-world robot force and tactile sensing.”

(Line 282-290) “

We also evaluated our model in real-time across six transfer groups under more dynamic conditions (see Supplementary Video 3). Tactile sensors were mounted on an ATI Nano17 F/T sensor, and we applied forces using four daily objects with different shapes and materials, including screwdriver, glue stick, plastic pizza, and a LEGO block (Supplementary Figure 17). A human operator performed five dynamic contact events, including press, rub, roll, push and pull, as well as continuous combinations of these on sensor surfaces (Fig. 6G). The transfer groups covered both homogeneous and heterogeneous translations described earlier. As shown in Supplementary Video 3 and Supplementary Figure 18-22, all test groups exhibited fast, accurate and robust responses comparable to a commercial high-accuracy F/T sensor.”

Fig. 5 | Material hardness effect and compensation. (C) Measured force-normalized depth curves for the seven elastomers during loading and unloading phases, demonstrating elastomers’ hardness and hysteresis property. (D) Relationship between the shear-to-normal force F_s / F_N and shear displacement of seven elastomers measured across different contact points by using our data collection trajectory. (E-F) Force prediction errors (i-ii) before and after using material compensation in *hard-to-soft* group (E) and *soft-to-hard* group (F). (G) Fit of force prediction to ground truth demonstrates the effectiveness of material compensation when transferring from a sensor with r18 (soft) skin to r6 (hard) skin before (i) and after (ii) using material compensation.

Supplementary Figure 8. Material compensation for force prediction. (A) Marker-based elastomers fabricated with seven base-to-activator ratios, where higher ratios yield softer elastomer and lower ratios produce harder elastomer. **(B)** Relationship of shear force and shear displacement for seven elastomers. **(C)** Relationship of force and normalized depth for seven elastomers with two-degree polynomial fitted lines in loading and unloading stage. **(D-E)** Normal and shear force prediction error changes after using material compensation in hard2soft (D) direction and soft2hard direction (E).

Supplementary Figure 9. Material compensation results for force prediction. (A) Distribution of data points

Supplementary Figure 13. Material property of three heterogeneous sensors. (B) Relationship of shear force and shear displacement (i) for three heterogeneous sensors. The curve (i) is then divided with normal force to show the relationship of F_S/F_N and shear displacement (ii).

Supplementary Figure 14. Friction of coefficient of heterogeneous sensors (A) and homogeneous sensors (B) with different hardness. The Friction of coefficient is calculated by the mean and std from the curve of F_S/F_N to shear displacement in the completely friction area (shear displacement in [0.6,0.8] mm).

Fig. 6 | Heterogeneous tactile force translation results. (E) Histogram of R^2 values (i-ii) and radar plot (iii-iv) of MAE before using GenForce and after using GenForce with material compensation.

Supplementary Figure 17. Objects used for dynamic force test compared with ATI nano17 F/T sensor in real-time. We use four objects with different materials, sizes and shapes, including a screwdriver, glue stick, Lego block (YCB) and pizza (YCB).

Supplementary Figure 18. Real-time force prediction when pressing on a GelSight sensor. (A) Demonstration of test object, contact event, tactile sensor and nano17. The force model is transferred from a GelSight (D-I) sensor. **(B-D)** Force prediction performance in Z-axis, X-axis and Y-axis respectively.

Supplementary Figure 19. Real-time force prediction when rubbing on a TacTip sensor. (A) Demonstration of test object, contact event, tactile sensor and nano17. The force model is transferred from a GelSight (D-I) sensor. **(B-D)** Force prediction performance in Z-axis, X-axis and Y-axis respectively.

Supplementary Figure 20. Real-time force prediction when rolling on a uSkin (three-axis) sensor. (A) Demonstration of test object, contact event, tactile sensor and nano17. The force model is transferred from a GelSight (D-I) sensor. **(B-D)** Force prediction performance in Z-axis, X-axis and Y-axis respectively.

Supplementary Figure 21. Real-time force prediction when continuously rubbing and pressing on a uSkin sensor. (A) Demonstration of test object, contact event, tactile sensor and nano17. The force model is transferred from a TacTip (palm) sensor. **(B-D)** Force prediction performance in Z-axis, X-axis and Y-axis respectively.

Supplementary Figure 22. Real-time force prediction when pushing and pulling on a GelSight (D-I) sensor. (A) Demonstration of test object, contact event, tactile sensor and nano17. The force model is transferred from a uSkin (3-axis) sensor. **(B-D)** Force prediction performance in Z-axis, X-axis and Y-axis respectively.

References:

68. Yuan, W., Li, R., Srinivasan, M. A., & Adelson, E. H. Measurement of shear and slip with a GelSight tactile sensor. In 2015 IEEE International Conference on Robotics and Automation (ICRA) 304-311 (2015).

2. **Efficiency of Unified Marker Representation:**

In this work, all tactile sensor inputs are segmented or directly converted into binary marker images and processed by convolutional networks. However, would directly inputting the marker coordinates (and depth, if available) be a more efficient and accurate approach? Can this unified marker representation efficiently handle dense tactile 3D point clouds (e.g., GelSight Mini), or is it limited in that regard?

Response: Thank you for the constructive comments regarding the efficiency of using marker images as unified representations across tactile sensors, especially in comparison with marker coordinates, depth map, and 3D point clouds. As you suggested, we have clarified our motivations for choosing marker-based representations by considering efficiency, robustness, and shared features across tactile sensors, as outlined below and detailed in the revised Supplementary Information.

(Line21-57)

“Text 1. Binary marker images as unified tactile representation

Among tactile modality such as marker coordinates, marker images, depth map and 3D point cloud, we choose binary marker images as unified representation by considering the efficiency, robustness and common features among tactile sensors as follow:

- 1. Efficiency:** In our work, the whole process of marker conversion and force prediction is fast in all sensors, which can be run in real-time around 29.6 Hz (the camera is 30fps) and demonstrated in our real robot applications with force control (see **Supplementary Video 3-7**). However, the marker coordinates normally need one more operation following marker segmentation in vision-based tactile sensors. They are extracted based on the marker images by finding the marker centroid with 2D displacement (X axis and Y axis, without information in Z axis). This process also loses information such as marker deformation compared with marker images, which provide essential information for the normal force. Depth map and 3D point clouds is normally used in vision-based tactile sensors calculated by calibration method¹³ and neural networks (such as GelSight-mini[R1]), thus are more complex and less efficient (higher computational cost) than 2D marker images.
- 2. Robustness:** Marker representations are 2D black-white images with binary format (0, 255). This data format is friendly to model training compared with RGB images ranging from 0-255 without considering image background and contact geometry. The npy saving format is uncompressed while only takes 39KB by packing into binary bit (see Supplementary Figure 34). However, the jpg file takes 57KB (30%+ memory cost than binary image) and introduces ringing and nonzero values near edges due to its compressed format. The CNN model is also robust to missing markers, irregular markers and random noises we introduced in our dataset (see Supplementary Figure 31). However, marker coordinates are easy to lose track due to the missing/overlapped markers under large deformation (see Supplementary Figure 32) or artifacts appearing on the image. The missing coordinate is unfriendly to model design and training. While depth map/3D point cloud are hard to extract from sensors with markers (see Supplementary Figure 33).
- 3. Common features among tactile sensors:** Binary marker images can be unified and easy to be converted from both vision-based tactile sensors or non-vision based tactile sensors. For vision-based tactile sensors, we can convert it from markerless tactile images by using regressive network (see Supplementary Figure 36) or segment from tactile images with physical markers (see Supplementary Figure 4 and Supplementary Figure 10). For non-vision based tactile sensors, the marker images can be converted from multichannel signals regardless of three-axis deformation sensing or pressure-only sensing in each taxel (see Supplementary Figure 10). However, the marker coordinates are hard to be obtained in non-vision based tactile sensors, such as electronic sensor arrays with pressure-only sensing. For 3D point cloud/depth map, it is hard to generalize in sensor arrays with sparse distributions. ”

Supplementary Figure 4. Marker-to-marker translation results for homogeneous sensors. (B) Marker segmentation pipeline for the GelSight sensors, including rough processing and fine processing. The rough process can run in real-time more than 29.6Hz together with force prediction when the FPS of the camera is 30 Hz.

Supplementary Figure 10. Marker conversion for heterogeneous sensors. (A) Signal-to-marker process for sensor arrays capable of sensing deformation in three-axis in each taxel. **(B)** Signal-to-marker process for sensor arrays capable of sensing deformation in z-axis in each taxel. **(C)** Marker segmentation pipeline for TacTip. The rough process can run in real-time more than 29.6Hz together with force prediction when the FPS of the camera is 30 Hz.

Supplementary Figure 31. Irregular and non-circular markers in our dataset. (A) Real-world marker images with explosive area, irregular edge and missing part. (B) Simulated marker images with elliptical shape, high-density and distortion.

Supplementary Figure 32. Failure cases of marker coordinates. (A) Missing marker and losing tracking of marker coordinate (i) compared with normal tracking in static images. (B) Missing markers and losing track of marker coordinates in sequential images due to large displacement.

Supplementary Figure 33. Failure cases of depth image from vision-based tactile sensors. The depth images of contact area are hard to extract from RGB images using a Depth Anything model.

Supplementary Figure 34. Comparison of npy and jpg saving format for marker images. The npy saving format is uncompressed while only takes 39KB by packing into binary bit. However the jpg file takes 57KB and introduces ringing and nonzero values near edges due to its compressed format.

References:

13. Yuan, Wenzhen, Siyuan Dong, and Edward H. Adelson. "Gelsight: High-resolution robot tactile sensors for estimating geometry and force." *Sensors* 17.12 (2017): 2762. R1. <https://github.com/gelsightinc/g robotics/blob/main/utilities/reconstruction.py> (GelSightmini-3d reconstruction)

Is this method unable to directly handle vision-based tactile sensors that do not have marker points? The authors should provide more detailed experiments and analyses.

Response: Thank you for the constructive comments regarding the handling of markerless vision-based tactile sensors. The key to using our model with such sensors is to generate marker images from markerless inputs. To achieve this, we use a pipeline called “markerless-to-marker translation,” proposed in our previous work [52]. As shown in **Supplementary Figure 36A**, this pipeline employs a lightweight regression network, DeepLabV3, well known in real-time semantic segmentation, to predict dense pixel displacements of markers predefined in a

reference image. The new experimental results in **Supplementary Figure 36B** demonstrate successful translation from markerless contact images to marker images. The predicted markers can then be used within our GenForce pipeline for both marker-to-marker translation and force prediction. This process enables our GenForce model to operate with vision-based tactile sensors that do not have physical markers.

To clarify this, we add modifications to our supplementary information as follows.

(Line 48-57)

“Common features among tactile sensors: Binary marker images can be unified and easy to be converted from both vision-based tactile sensors or non-vision based tactile sensors. For vision-based tactile sensors, we can convert it from markerless tactile images by using regressive network (see Supplementary Figure 36) or segment from tactile images with physical markers (see Supplementary Figure 4 and Supplementary Figure 10). For non-vision based tactile sensors, the marker images can be converted from multichannel signals regardless of three-axis deformation sensing or pressure-only sensing in each taxel (see Supplementary Figure 10).”

Supplementary Figure 36. Markerless-to-marker translation. (A) Pipeline for markerless to marker translation by using a lightweight real-time segmentation model DeepLabV3 [52]. (B) Examples for translating contact image without (w/o) marker to binary marker images.

[52] Ou, Ni, Zhuo Chen, and Shan Luo. "Marker or markerless? mode-switchable optical tactile sensing for diverse robot tasks." *IEEE Robotics and Automation Letters* (2024).

3. ***Real-world Robotic Applications:***

I recognize the great potential of this work in various tactile tasks. It would be even better if experiments and analyses demonstrating its application in real-world slip detection or robotic manipulation were provided.

Response: Thank you for the constructive comments regarding real-world robot manipulation tasks using our models. As you suggested, we conducted two new experiments on daily-object robot grasping and on slip detection and avoidance using the transferred force. In particular, we show that, with the transferred force, we can also transfer force controllers for the robot grasping task across heterogeneous sensors. As shown in Fig. 7, task (1), daily-object grasping, examines the application of the transferred normal force with a shared proportional force controller. Task (2), slip detection and avoidance, examines both normal and shear force sensing using a shared proportional force controller and a slip-reactive controller.

To clarify this, we add a new section to our revised manuscripts and add **Supplementary Video 3-7**.

(Line 292-323)

“Transferable force sensing and control in robot manipulation

To demonstrate the applicability of our model in real world, we install heterogeneous tactile sensors on a robot arm to show the transferable force sensing and control in daily-object robot grasping, slip detection and avoidance (See Supplementary Figure 25A). The robot setup can be seen in **Fig. 7A-i**. In the robot grasping task, we mount a flat-surface vision-based tactile sensor (GelSight, marker pattern A-II) on the left finger of robot and a flat-surface magnetic tactile sensor (uSkin with three-axis sensing capability) on the right finger. We transfer the force prediction model to above two sensors by using a third flat-surface vision-based tactile sensor (GelSight, marker pattern A-II) tactile sensor with our GenForce model. The task requires robot to grasp nine daily objects (See Supplementary Figure 25B) with different sizes, shapes and material without damage them. Those objects include potato chip, grape, strawberry, orange, plum, wood block, glue stick, meat box and tea box, which are unseen in the training dataset. During the grasping, the arm is controlled with a proportional controller to grasp those objects with fixed normal forces ranging from 0.6N to 1.2N. Both sensors are shared with the same force controller. As shown in **Fig. 7C**, Supplementary Video 4 and Supplementary Video 5, both sensors can equip the robot arm an accurate force sensing so that the robot arm can successfully grasp all objects with target forces without damage. Even for challenging objects such as chips and fresh fruits, the robot can achieve delicate grasping by using the transferred force prediction model combining with force control.

For the second task, we extend the grasping experiment (see **Fig. 7A-ii**; Supplementary Videos 6 and 7) to include slip detection and avoidance. A curved-surface, vision-based TacTip sensor (palm shape) is mounted on the left finger, and a three-axis magnetic uSkin sensor is mounted on the right. The TacTip’s force prediction model is transferred from GelSight (D-I), and the uSkin model is subsequently transferred from the TacTip. The task proceeds through several

stages—moving down, proportional-control grasping, lifting, slip detection and avoidance at the top position, release, and return to home—with the force controller active only during the grasp and slip-detection phases. We evaluate four objects—banana, plum, meat box, and glue stick. Beyond completing the grasp, external forces are applied by a human at the top position to induce slip. The robot detects slip via changes in shear force and responds by narrowing the gripper width. As shown in **Fig. 7B** and Supplementary Videos 6–7, the system completes all stages successfully, demonstrating the practical applications of our model in real robotic tasks.”

(Line 572-584) “

Robot grasping and slip control

We use a Franka FR3 robot arm with a Franka hand (max opening 0.08 m), an Intel RealSense camera, daily objects (9 for grasping and 4 for slip control), and tactile sensors. For grasping, normal force predicted from GelSight and uSkin feed into a simple proportional force controller. The gripper closes at 0.005 m/s from an object-dependent initial width (about 0.025 m for grapes up to 0.075 m for oranges). Depending on fragility, the target force is 0.6 N, 0.8 N, or 1.2 N (e.g., lower for chips and grapes, higher for rigid items like a meat box). Force controller runs at 10 Hz with $K_p=0.0004$. The slip-control experiments use the same setup, switching the tactile sensors to TacTip and uSkin. Here we pair the same proportional controller with a top-position slip check: after lifting, we monitor F_x and F_y at 20 Hz for 10 s, keeping a 0.5 s window. If the short-term shear force variation over three samples in either axis exceeds 0.3 N, we tighten the grasp by 0.001 m and command the change at 0.015 m/s. See more details in Supplementary Video 4–7.”

Supplementary Figure 25. Applications of transferable force sensing in real robot tasks. (A) Demonstration of transferable force in robotic force control. (B) Setup for robot grasping, slip detection and avoidance using heterogeneous sensors. (C) Daily objects used for robot grasping with transferable force control.

Fig. 7 | Robot grasping and slip avoidance with transferable force sensing. (A) Transferable force sensing in robot slip detection and avoidance. (A-i) Robot arm equipped with TacTip (palm) and uSkin (three-axis). (A-ii) Sequence of robot grasping an object with force control, slip detection, and avoidance. (B) Real-time measurement of forces, slip-detection status, gripper width, and robot status during grasping task of A-ii. (C) Daily objects grasping with transferable force sensing and control using GelSight (A-II) and uSkin (three-axis). All forces models used in each sensor are transferred from other sensors. See more details in our Supplementary Video 4-7.

4. ***Quality of Figures:***

All the figures in this paper appear blurry when enlarged and should be replaced with vector graphics.

Response: Thank you for highlighting the figure quality issue. We have replaced all figures with high-resolution, high-quality vector graphics.

5. ***The Rigor of the Biological Inspiration:***

This work draws inspiration from the human ability to transfer tactile sensations between the fingers and the palm, aiming to illustrate that humans can achieve generalized force perception through information sharing across different sensory receptors. However, the skin on the human fingers and palm should belong to the same tactile system and represents a scenario of "homogeneous" sensors, rather than a transfer between heterogeneous sensors. The authors should revise this part of the Introduction with greater rigor.

Response: Thank you for your comments on the biological inspiration. We agree that the skin on the fingers and palm is homogeneous within the human tactile system. As suggested, we have revised the relevant part of the Introduction for rigor. In the revised text, we (i) clearly state that “homogeneous” and “heterogeneous” refer only to tactile sensors: homogeneous sensors share the same sensing principle but differ in configuration (e.g., the GelSight family), whereas heterogeneous sensors employ distinct sensing modalities and material properties (e.g., GelSight vs. uSkin); and (ii) adopt a broader, bio-inspired phrasing by avoiding fine-grained claims such as transferring tactile sensations between specific fingers and the palm, instead describing transfer across skin regions on the hand, consistent with references [21–22].

(Line 17-20) “

Abstract: Humans achieve stable and dexterous object manipulation by coordinating grasp forces across multiple fingers and palms, facilitated by a unified tactile memory system in the somatosensory cortex. This system encodes and stores tactile experiences across skin regions, enabling the flexible reuse and transfer of touch information.”

(Line 24-26) “

We demonstrate that GenForce generalizes across both homogeneous sensors with varying configurations and heterogeneous sensors with distinct sensing modalities and material properties.”

(Line 63-68) “

The ability to unify and transfer sensory information across skin is a critical aspect of human control over grasp forces, ensuring both stability and dexterity during object manipulation^{18,19}. This capability is equally essential for robots equipped with tactile sensors to perform in-hand object manipulation tasks in unstructured environments²⁰. In humans, the tactile memory system (**Fig. 1C**) enables the storage and retrieval of experienced tactile information, such as haptic stimuli, across skin regions on hands^{21,22}.”

(Line 71-74) “

Mimicking this unified representation and transferable tactile sensing could enable tactile sensors to learn from each other, reuse collected tactile experience and transfer tactile sensing across robot hands, enhancing their dexterity and adaptability. ”

(Line 96-99) “

GenForce further improves force accuracy by compensating for material differences. Extensively validated for generalizability, accuracy, and robustness, GenForce works across diverse tactile sensors, including flat-surface vision-based tactile sensors (GelSight¹³), flat-surface electronic sensor arrays (uSkin¹⁵) with either three-axis sensing or z-axis-only sensing, and curved-surface vision-based tactile sensors (TacTip¹⁶), spanning varying configurations (Fig. 2A-B).”

References:

21. Gallace, A. & Spence, C. The cognitive and neural correlates of tactile memory. *Psychol Bull* 135, 380–406 (2009).
22. Harris, J. A., Harris, I. M. & Diamond, M. E. The Topography of Tactile Working Memory. *The Journal of Neuroscience* 21, 8262–8269 (2001).

Overall, this work systematically integrates different sensors in the force prediction context and demonstrates practical value across various robotic and tactile tasks, providing valuable insights for future research.

Response: Thank you for the thoughtful and encouraging assessment.

Reviewer #1 (Remarks on code availability):

The updated code provides a detailed README file to elaborately introduce the procedure of Data Collection, Model Training. However, I did not try to install and run the code.

Response: Thank you for the positive feedback on our code and procedures and we appreciate your time. We plan to open-source our code and dataset upon acceptance, and we will continue to update the codebase based on feedback from the tactile community.

We sincerely appreciate your insightful feedback. Your comments have greatly improved the quality and clarity of our manuscript. We hope that the revised version, together with our detailed responses, addresses all of your concerns. Thanks !

Reviewer #2 (Remarks to the Author):

The paper presents a pipeline to generate tactile responses and corresponding force profiles across tactile sensors. The pipeline includes i) a module to obtain a binary marker-based representation of the tactile sensor responses, ii) a M2M module, based on latent diffusion models and trained in simulation, that, using a few paired reference images from a calibrated sensor, converts the marker-style of one sensor into another, iii) a module that, given the translated marker images, trains a spatiotemporal force-prediction model to estimate 3-axis forces (x, y, z) from the tactile sequence.

Response: We thank the reviewer for the thorough evaluation and for recognizing the main contributions of our work. As summarized, our pipeline comprises: (i) a module that converts raw tactile signals into a unified, binary, marker-based representation across different tactile sensors; (ii) a marker-to-marker (M2M) translation module based on latent diffusion, pretrained in simulation and adapted to diverse real tactile sensors using a small set of paired images; and (iii) a spatiotemporal model that enables target sensors to predict 3-axis forces in an unsupervised manner by learning from translated marker sequences without collecting new force labels. In response to your constructive comments, we have **substantially revised and strengthened the manuscript**. We have **added new experiments** and **detailed analyses** to address each concern. We also added **new real-robot experiments with force sensing and control** to **strengthen the applicability of our work**. **In total, we revised the manuscript with 7 main figures, 43 supplementary figures, 4 supplementary tables, 7 supplementary videos (including 5 new videos and 2 videos remade from the previous 8 videos), and 1 project page (Link)**. All textual changes are highlighted in red in the revised manuscript, and every point is addressed in the detailed, point-by-point response below, with explicit pointers to the relevant pages, lines, and figure, table, video references.

The work targets a very important problem faced by the tactile sensors' community related to the generation of tactile datasets since their collection is time-consuming and they are generally not reusable across tactile sensor technologies due to differences in the sensor hardware.

Response: Thank you for recognizing the importance of our target problem to the tactile sensing community. By introducing a unified, marker-based representation and a cross-sensor translation pipeline, our work takes a direct step toward addressing this challenge and enables the reuse of existing force-labeled datasets, thereby reducing both data collection time and cost.

The problem of generating tactile images across different tactile sensor technologies has already been presented in previous papers cited in this work, where the transfer has been done between a camera based tactile sensor and a capacitive based tactile array. With respect to the cited paper, this work faces a simpler problem when generating tactile dataset across sensors modalities, as the sensors they selected (camera-based sensors and the magnetic-based sensor) are all providing both normal and shear components and comes with marker-based representations, however they present a modular architecture that enable

generalisation within the marker-based classes they are focusing on as well as the generation of the force profiles. The amount of analysis performed cover also a wide range of different cases.

Response: Thank you for drawing attention to a related work we cited in our original manuscript and for encouraging a deeper comparison. While prior studies have explored transfer between a camera-based tactile sensor and a capacitive array, our work differs substantively in problem formulation, tactile representation, model design, transfer direction, and real-robot validation. We clarify these distinctions below and provide a consolidated comparison in the revised manuscript.

1. Problem formulation and transfer directionality

- **Prior work [30] (a preprint at the time of our submission)** translates grayscale images from a high-resolution vision-based sensor (Digit) to a lower-resolution capacitive array (Cyskin) using a Pix2Pix-style model. As reported, this translation is asymmetric (only support high resolution \rightarrow low resolution) and does not generalize in the reverse direction, also implying a separate model is required for each ordered sensor pair.
- Our approach targets many-to-many transfer across multiple sensors and directions using a single translation model. This is enabled by a sensor-agnostic, binary, marker-based representation and an image-conditioned latent diffusion model, eliminating per-pair, per-direction retraining.

2. Unified representation with explicit 3-axis semantics

- In [30], normal-force readings from Cyskin are interpolated into grayscale images, which encode z-axis information only and do not explicitly carry x/y shear force. Hence, grayscale encodings are not well-suited to representing shear, making it hard to extend to sensor arrays with 3-axis force sensing.
- Our unified marker representation uses marker size to encode normal force (z) and marker displacement to encode shear (x,y). This representation naturally supports 3-axis sensors (e.g., uSkin) while is also compatible with z-only arrays by varying marker size alone. We have added new experiments and videos to support the transfer to sensor array capable of z-axis only force sensing in **Supplementary Figure 10, Supplementary Figure 23, Supplementary Figure 24 and Supplementary Video 3.**

3. Model scalability and generalization across sensors and forces

- Work [30] uses a pixel-to-pixel translation mode, and as noted by the authors, exhibits limitations in transfer direction and generalization in different forces.
- Our marker-to-marker translator is based on latent diffusion and enables scaling to additional sensors without training separated models as the number of sensors grows. Our model is also tested to marker images applied in wide force ranges.

4. Robot task validation: from forces translation to force controller translation

- Prior work primarily reports quantitative metrics on translated images, leaving real-world task performance unclear.
- We validate our model on real-time 3-axis force prediction (see **Supplementary Video 3**) and test the accuracy compared with a commercial ATI Nano17 F/T sensor, demonstrating robustness across sensors, dynamic contact events, and force ranges. We

further showcase applicability in robotic manipulation task (object grasping, slip detection and avoidance) using transferred force and force control (see **Supplementary Video 4-7**).

We summarize the key differences in Supplementary Table 1. To the best of our knowledge, our work is the first to address transferable force sensing across tactile sensors of different modalities. We believe this contribution is of significant interest to the tactile sensing and robot manipulation community. It offers timely, practical value for real-world robot manipulation tasks with transferable force sensing and force control.

To clarify this, main modifications to the revised manuscript can be found as follows.

(Line 83-86) “Some efforts to directly transform tactile signals across sensors [31–34] similarly fall short on one-to-one translation, as they overlook the importance of a unified tactile representation among sensors and lack of generalizability to diverse tactile sensors (see **Supplementary Table 1**).”

Supplementary Table 1. A comparison between work [30] and our work*

Aspect	work [30]	This work (GenForce)
Representation	Grayscale image (interpolated z-axis pressure value)	Binary marker image (marker size→z, marker displacement→x,y)
Sensors	Digit (vision), Cyskin (capacitive, z-only)	Vision and magnetic (e.g., GelSight, TacTip, uSkin; three-axis capable)
Transfer direction	One-to-one, high resolution→low resolution only	Many-to-many, bidirectional
Model	Pix2Pix GAN	Image-conditioned latent diffusion
Materials / Geometry	Flat surfaces	Flat and curved surfaces, varied materials
Force sensing	\	Demonstrated across wide force ranges
Applicability	\	Robot grasping, slip detection/avoidance
Scalability	Separated model per sensor pair	Scales to new sensors within one model
Generalizability	Only test in two sensors and one direction	Test in 5+7+3 sensors and 132 (sim) + 74 (real) transfer direction

*Scalability means a model can transfers across selected sensors in one model, rather than training separated models for each sensor pair. With the sensor types growing, the later one is extremely hard to manage.

*We test across 5 GelSight sensors with different illumination colours and markers, 7 GelSight with different elastomer hardness and 3 heterogenous sensors, i.e. GelSight, TacTip, uSkin.

*Magnetic tactile sensors with three-axis sensing capability can reduce to z-axis only similar to capacitive and resistive sensor in output signal by just using the z-component of each texels, while the inverse direction is impossible.

Reference:

30. Grella, F., Albin, A., Cannata, G., & Maiolino, P. (2024). Touch-to-Touch Translation-- Learning the Mapping Between Heterogeneous Tactile Sensing Technologies. arXiv preprint arXiv:2411.02187.

The paper is structured such that it is possible to easily follow all the steps of the pipeline but could be improved to achieve more clarity as some parts are dense of details which are also repeated in the supplementary materials. Some parts of the paper clearly look like they were rewritten by a chatbot. While the use of AI tools for improving the English for not native speakers might be acceptable, the use of chatbot should be disclosed explicitly as per the journal's guidelines.

Response: Thank you for acknowledging our paper's structure and for your constructive comments on improving clarity, especially regarding repeated content between the main paper and the supplementary materials. As suggested, we carefully revised the pipeline description across both documents to make it more fluent and compact. We only used a chatbot to polish wording and syntax, and we have disclosed this usage in the revised manuscript as recommended.

(Line 773-774) “

Additional information:

- AI tools were used solely for grammar checking in this manuscript.”

Comments on the stated contribution claims:

The claim of eliminating the need for expensive force/torque sensors seems overstated as there is the need to calibrate at least one of the sensors.

Response: We acknowledge that at least one calibrated sensor is required, and we apologize for not stating this clearly in the original paper. To clarify, our claim of eliminating the need for expensive force/torque sensors relies on the availability of our open-sourced dataset (release upon acceptance) : users without force/torque sensors can select a sensor in our dataset that is closest to theirs and use our pipeline to calibrate their sensors without collecting new force labels. We also aim to encourage the community to contribute additional datasets after publication, expanding a shared force-sensing database. We believe that our model, along with the data collection process, indenters, and code, can serve as a standardized, efficient, and high-accuracy pipeline for force prediction in tactile sensors, enabling worldwide users to contribute via the same procedure. To prevent misunderstandings or overstatements, we have added a clear clarification in the Discussion section.

Main modifications can be found in our revised manuscript as follows.

(Line 387-409) “

Some limitations of our models and possible future work:

- We still require at least one sensor that is fully calibrated with force-labeled data and material priors. Such a calibrated sensor can be sourced from our dataset or from other existing sensors, followed by collecting a small number of locations-pair reference images with fixed trajectory. For material priors, instead of using traditional material characterization method, some recent in-situ method^{43,44} can be referred to avoid use F/T sensor, moving stage and robot arm. Looking ahead, an important direction is to build foundational tactile database like ImageNet⁴⁵ that totally removes the need for any data collection and material calibrations. Given our model’s adaptability and data reusability, the dataset is inherently scalable using our data collection trajectory and could enable users worldwide to rapidly equip their own sensors with high-performance force sensing. ”

References:

43. Zhao, C., Ren, J., Yu, H. & Ma, D. In-situ Mechanical Calibration for Vision-based Tactile Sensors. in 2023 IEEE International Conference on Robotics and Automation (ICRA) 10387–10393 (2023).
44. Li, M., Zhang, L., Zhou, Y. H., Li, T. & Jiang, Y. EasyCalib: Simple and Low-Cost In-Situ Calibration for Force Reconstruction With Vision-Based Tactile Sensors. IEEE Robot Autom Lett 9, 7803–7810 (2024).
45. Deng, J. et al. ImageNet: A large-scale hierarchical image database. in 2009 IEEE Conference on Computer Vision and Pattern Recognition 248–255 (2009).

While the architecture has been designed to be modular and extendable, the model assumes a specific marker structure (circular, optical, regular) across sensors and would probably require some adaptation as well as new calibrated paired data for sensors that are structurally different (e.g., capacitive and resistive).

Response: Thank you for the positive feedback on our model’s modularity and extensibility. The unified marker representation is a central component of our work, designed to bridge tactile signals across sensors because marker patterns are widely used in tactile sensing. This binary image encodes dynamic deformation information for force prediction and is straightforward to train within image-to-image translation models due to its binary format and low noise. To obtain this representation, we adapt the pipeline in two categories (see Supplementary Figure 4 and see Supplementary Figure 10): (1) marker segmentation for vision-based tactile sensors; and (2) signal-to-marker processing for electronic sensor arrays.

Specifically, for sensor arrays, whether resistive, capacitive, or magnetic, we can map the normal-direction tactile response to marker size (see **Supplementary Figure 10**) and, when available (e.g., uSkin), map shear response to marker displacement. If only z-axis signals are available, we represent normal force solely through marker size. The reference marker image used for mapping can be flexibly designed with varying marker size, density, and spatial distribution. Additionally, the mapping sensitivity for both marker size and displacement can

be tuned via parameters during the conversion from electrical signals (see **Supplementary Figure 12A**). We have tested our model with diverse marker images using a simulated dataset (12 types) and real-world datasets spanning multiple tactile sensors, including 5 homogeneous translations, 7 material compensations, and 3 heterogeneous translations. Despite missing markers, irregular edges, and non-circular dots in our datasets, the model reliably transfers across these conditions (see **Supplementary Figure 32**).

The newly collected, small set of location-paired data is used to bridge deformation across tactile sensors. It is easy and fast to acquire using our data-collection trajectory, and it does not require force labels from expensive commercial sensors such as the ATI Nano17 F/T. For each paired sample at a single contact point, acquisition takes under 3 seconds, and the total time per indenter is under 7 minutes in our experiments (see **Supplementary Figure 2B**).

Main modifications to the revised supplementary information can be found as follow.

(Line21-57)

“Text 1. Binary marker images as unified tactile representation

Among tactile modality such as marker coordinates, marker images, depth map and 3D point cloud, we choose binary marker images as unified representation by considering the efficiency, robustness and common features among tactile sensors as follow:

- 1. Efficiency:** In our work, the whole process of marker conversion and force prediction is fast in all sensors, which can be run in real-time around 29.6 Hz (the camera is 30fps) and demonstrated in our real robot applications with force control (see **Supplementary Video 3-7**). However, the marker coordinates normally need one more operation following marker segmentation in vision-based tactile sensors. They are extracted based on the marker images by finding the marker centroid with 2D displacement (X axis and Y axis, without information in Z axis). This process also loses information such as marker deformation compared with marker images, which provide essential information for the normal force. Depth map and 3D point clouds is normally used in vision-based tactile sensors calculated by calibration method¹³ and neural networks (such as GelSight-mini[R1]), thus are more complex and less efficient (higher computational cost) than 2D marker images.
- 2. Robustness:** Marker representations are 2D black-white images with binary format (0, 255). This data format is friendly to model training compared with RGB images ranging from 0-255 without considering image background and contact geometry. The npy saving format is uncompressed while only takes 39KB by packing into binary bit (see Supplementary Figure 34). However, the jpg file takes 57KB (30%+ memory cost than binary image) and introduces ringing and nonzero values near edges due to its compressed format. The CNN model is also robust to missing markers, irregular markers and random noises we introduced in our dataset (see Supplementary Figure 31). However, marker coordinates are easy to lose track due to the missing/overlapped markers under large deformation (see Supplementary Figure 32) or artifacts appearing on the image. The missing coordinate is unfriendly to model design and training. While

depth map/3D point cloud are hard to extract from sensors with markers (see Supplementary Figure 33).

- 3. Common features among tactile sensors:** Binary marker images can be unified and easy to be converted from both vision-based tactile sensors or non-vision based tactile sensors. For vision-based tactile sensors, we can convert it from markerless tactile images by using regressive network (see Supplementary Figure 36) or segment from tactile images with physical markers (see Supplementary Figure 4 and Supplementary Figure 10). For non-vision based tactile sensors, the marker images can be converted from multichannel signals regardless of three-axis deformation sensing or pressure-only sensing in each taxel (see Supplementary Figure 10). However, the marker coordinates are hard to be obtained in non-vision based tactile sensors, such as electronic sensor arrays with pressure-only sensing. For 3D point cloud/depth map, it is hard to generalize in sensor arrays with sparse distributions. ”

Your model has been trained and evaluated only on marker-based tactile images coming from simulation and real-world. This type of representation is common within the space of optical tactile sensors and magnetic based tactile sensor arrays which have been proposed in this work. Evidence that it can generalize to fundamentally different modalities, such as capacitive arrays (no markers, no shear force), piezoelectric or resistive pressure sensors have not been presented.

Response: Thank you for the constructive comments regarding evidence of our model’s generalizability to other modalities, including sensor arrays without markers and without shear sensing, particularly capacitive and resistive pressure sensors. We apologize for not clearly explaining how our signal-to-marker processing applies to sensors with only z-axis (pressure) signals and how our model handles the absence of shear.

First, we clarify that sensor arrays sensing only normal pressure at each taxel are fully supported by our signal-to-marker pipeline (see **Supplementary Figure 10**). The advantage of the marker image is its ability to encode three-axis responses per taxel: marker size for the z-axis and marker displacement for the x- and y-axis. When applying our method to arrays that respond only to normal forces, we simply map the z-direction measurement to marker size and omit the displacement component. This preserves the essential deformation cues for force prediction while maintaining compatibility with our image-to-image translation framework.

Although the magnetic tactile sensor (uSkin) provides x- and y-axis information in our tests, we can use only its z component to mimic the electrical measurements of typical capacitive or resistive pressure sensors. In this setting, the multichannel uSkin signal is downsampled to retain only the third (z) channel per taxel. During the signal-to-marker stage, the predefined markers are constrained to change only in size under external force, without x- or y-axis displacement. Based on this modality, termed “uSkin (z-axis)”, we train our model to transfer it across a flat-surface vision-based tactile sensor (GelSight, marker pattern D–I) and a curved-surface vision-based tactile sensor (TacTip, palm shape).

As shown in **Supplementary Figure 23**, the model successfully transfers images from GelSight (D-I) and TacTip (palm) to uSkin (z-axis). Within the contact region, only marker size changes appear in the generated images, matching the target modality. We then use the transferred uSkin (z-axis) images to train a force-prediction model and deploy it in real time, benchmarking against an ATI Nano17 F/T sensor with three daily objects (see **Supplementary Figure 24 and Supplementary Video 3**). Because x-y information is absent, pressure-only arrays are not expected to produce precise shear predictions; accordingly, we evaluate only normal-force prediction using the transferred model from a vision-based sensor. The results show accurate, real-time normal-force estimation, demonstrating that our approach generalizes to modalities without shear sensing and without intrinsic marker shear displacement.

To clarify this, we add modifications to our revised manuscripts as follows.

(Line 420-448)“

Text 10. Performance on sensor arrays with pressure sensing

For sensor arrays, whether resistive, capacitive, or magnetic, we can map the tactile response in normal direction to marker size (see Supplementary Figure 10), while response in shear into marker displacement if available, such as uSkin. If only z-axis signals are available, we represent the normal force by only using marker size. The reference marker image used for mapping tactile response from sensor array can be designed in varied marker size, density and distribution. The mapping sensitivity of the marker changes in size and displacement based on the reference marker image can also be adjusted by parameters when converting from the electrical changes (see Supplementary Figure 12A).

To test the transferability of our model among sensor arrays with only pressure sensing, we downsample the $4 \times 4 \times 3$ multichannel signal of uSkin (three-axis) into 4×4 channels only with measurement from the z-channel in each texel. Then, in the signal-to-marker stage, the predefined markers can only change the size due to external forces without the displacement in x and y axis. Based on this, we train our model using the modality, named uSkin (z-axis), to transfer across flat-surface vision-based tactile sensor (GelSight, marker pattern D-I) and curved-surface vision-based tactile sensor (TacTip, palm shape).

As shown in Supplementary Figure 23, the model successfully transfers the images from the GelSight (D-I) and TacTip (palm) to uSkin (z-axis). In the contact area, only marker size changes are observed in the generated images similar to the target images. We then use the transferred uSkin (z-axis) image to train the force prediction model and deploy in real-time to compared it with ATI nano17 F/T sensor by using three daily objects (see Supplementary Figure 24 and Supplementary Video 3). Due to the loss of information in x and y directions, it is known that the sensor array with only z-axis sensing capabilities is hard to get precise shear force prediction. In this case, we only show its capability to predict normal force in real time by using the transferred model from a vision-based tactile sensor. We can see that the sensor is still able to respond to high-accuracy forces in real-time. This experiment verifies our model is

also generalizable to modality without shear force and intrinsically no marker shear displacement.”

Supplementary Figure 10. Marker conversion for heterogeneous sensors. (A) Signal-to-marker process for sensor arrays capable of sensing deformation in three-axis in each taxel. (B) Signal-to-marker process for sensor arrays capable of sensing deformation in z-axis in each taxel. (C) Marker segmentation pipeline for TacTip. The rough process can run in real-time more than 29.6Hz together with force prediction when the FPS of the camera is 30 Hz.

Supplementary Figure 23. Marker-to-marker result among GelSight (D-I), TacTip (palm) and uSkin with only z-axis component. (A) Transferring marker image from GelSight (D-I) to uSkin (z-axis). (B) Transferring from marker image TacTip (palm) to uSkin (z-axis). Note that, no shear displacement in uSkin’s marker image due to only use signal from z-axis, similar to other type of sensor array only with z-axis (pressure) sensing capability.

Supplementary Figure 24. Dynamic force test for uSkin (z-axis) with force model transferred from GelSight (D-I). (A) Continuous pressing with a Lego block. (B) Rubbing with a plastic Pizza from YCB. (C) Pushing with a glue stick. All test is compared with ATI nano 17. See more details in our Supplementary Video 3.

Furthermore, the whole architecture M2M + GenForce looks like dependent on “materials-priors” which cannot generalize well to new materials not seen during M2M training, account for ageing or continuous variation in the stiffness (e.g., slightly stiffer vs. much stiffer materials). For these reasons, this claim appears to be an overstatement and to support it the study should include at least a sensor which is fundamentally different or state clearly the limit of this generalizability.

Response: Thank you for the insightful comments on the material priors and concerns about generalizability to new materials. We apologize for not clarifying this earlier. In our material compensation section, we evaluate seven different elastomers to capture stiffness variations, also reflecting aging and gradual changes in tactile sensors, with their properties of elastic modulus reported in **Fig.5C**. Among heterogeneous sensors, three sensors also span different hardness levels (see **Supplementary Figure 13A**). Our method remains effective for bidirectional translation across sensors with slightly to substantially stiffer materials demonstrated with the experimental results in **Fig.5**. In addition, following prior suggestions, we introduced a new experiment using uSkin (z-axis) to further assess its availability in transferring across sensors without shear. We have expanded the manuscript to include these results and a detailed discussion to gradually stiffer materials.

Main modifications can be found below:

(Line 200-227) “

We divide the transfer direction into two groups: (1) *hard-to-soft* and (2) *soft-to-hard*. In **Fig.5E-i**, combinations like *6_8*, *6_10*, and *6_12* indicate progressively transfer to harder elastomer (slightly harder, harder, much harder), whereas *8_6*, *10_6*, and *12_6* indicate the inverse, i.e., softer transfer. These combinations reflect the long-term aging behavior of silicone elastomers. Before material compensation, normal-force errors generally grow with increasing hardness gap (**Fig. 5E**). On average, the normal-force error is 1.41 N for *hard-to-soft* and 1.03 N for *soft-to-hard*. For shear forces, average errors are 0.18 N (x) and 0.20 N (y) in *hard-to-soft*, and 0.16 N (x) and 0.18 N (y) in *soft-to-hard*. Shear errors are more stable across both groups, because applied shear forces are smaller than normal forces and less sensitive to hardness gaps (see Supplementary Text 9A). Additional results for 15 other translation combinations are provided in Supplementary Figure 8D–E.

To mitigate the increased force errors due to material hardness gaps, we propose a material compensation method (**Fig. 5B**) that uses material priors to correct force labels before training. The priors are the fitted elastic modulus profiles, i.e force-normalized depth profiles shown in Supplementary Figure 8C. Intuitively, this process increases the amount of force when transferring from soft to hard skin and decreases it in the opposite direction. As shown in **Fig. 5E-F**, this improves accuracy across both normal and shear forces. After compensation, average normal-force error drops to 0.99 N in the *hard-to-soft* group (30% reduction) and 0.87 N in the *soft-to-hard* group (16% reduction). For shear forces, average errors in the *hard-to-soft* group reduce to 0.10 N on the x-axis (44% reduction) and 0.14 N on the y-axis (30% reduction); in the *soft-to-hard* group, errors reduce to 0.13 N (F_x , 19% reduction) and 0.15 N (F_y , 19% reduction). The method yields lower errors in 95% of all combinations within the *hard-to-soft*

group (including those in Supplementary Figure 8D) and in 57% of the *soft-to-hard* group. A representative case, transferring from r18 (soft) to r6 (much harder) skins in **Fig. 5G**, shows improvements in R^2 values from 0.73 (F_x), 0.79(F_y), and 0.78 (F_z) to 0.87, 0.86, and 0.84 across the three axes. Real-time visualizations in Supplementary Figure 9B and Supplementary Video 2 further illustrate the effectiveness of the material compensation, which is ignored by traditional studies while is very important in real-world robot force and tactile sensing.”

(Line 387-409) “

Some limitations of our models and possible future work:

- We still require at least one sensor that is fully calibrated with force-labeled data and material priors. Such a calibrated sensor can be sourced from our dataset or from other existing sensors, followed by collecting a small number of locations-pair reference images with fixed trajectory. For material priors, instead of using traditional material characterization method, some recent in-situ method^{43,44} can be referred to avoid use F/T sensor, moving stage and robot arm. Looking ahead, an important direction is to build foundational tactile database like ImageNet⁴⁵ that totally removes the need for any data collection and material calibrations. Given our model’s adaptability and data reusability, the dataset is inherently scalable using our data collection trajectory and could enable users worldwide to rapidly equip their own sensors with high-performance force sensing.
- As the force prediction model is based on unsupervised learning by transferring data from other tactile sensors, it is still not comparable to the accuracy of supervised learning which demands high-volume labelled data. However, if the user wants to get very high-accuracy force prediction, this model is expected to be very good pretrained model to be finetuned with only a few amounts of force labels.
- It remains an open question how this model can be extended to transfer across finger-sized tactile sensors, large-area electronic skins, and sensors lacking explicit taxels, such as those based on electrical impedance tomography⁴⁶ sensors. We propose that this challenge could be addressed by dividing large-area skins into smaller subareas, thereby enabling transfer to regions comparable in size to finger-sized sensors.

References:

43. Zhao, C., Ren, J., Yu, H. & Ma, D. In-situ Mechanical Calibration for Vision-based Tactile Sensors. in 2023 IEEE International Conference on Robotics and Automation (ICRA) 10387–10393 (2023).
44. Li, M., Zhang, L., Zhou, Y. H., Li, T. & Jiang, Y. EasyCalib: Simple and Low-Cost In-Situ Calibration for Force Reconstruction With Vision-Based Tactile Sensors. IEEE Robot Autom Lett 9, 7803–7810 (2024).
45. Deng, J. et al. ImageNet: A large-scale hierarchical image database. in 2009 IEEE Conference on Computer Vision and Pattern Recognition 248–255 (2009).

Fig. 5 | Material hardness effect and compensation. (A) Schematic representation of tactile skins with varying hardness and their corresponding force-depth relationships. (B) Material compensation process incorporates material priors to correct force labels during loading and unloading phases. (C) Measured force-normalized depth curves for the seven elastomers during loading and unloading phases, demonstrating elastomers' hardness and hysteresis property. (D) Relationship between the ratio of shear force by normal force (F_s/F_N) and shear displacement of seven elastomers measured across different contact point by using our data collection trajectory. (E-F) Force prediction errors (i-ii) before and after using material compensation in hard-to-soft group (E) and soft-to-hard group (F). (G) Fit of force prediction to ground truth demonstrates the effectiveness of material compensation when transferring from a sensor with r18 (soft) skin to r6 (hard) skin before (i) and after (ii) using material compensation.

The +/-4N range is suitable for many robotics application like grasp, push, pull and the error in that range is impressive.

Response: Thanks for your positive feedback on the performance on our force prediction model in the force range of +/-4N and find it impressive. We agree that this force range is

really useful for robot applications such grasping, pushing and pulling. In this work, we aim to not only quantitatively demonstrate the accuracy of our model, but also demonstrate its use in real robot application. We have conducted two new experiments on robot applications by using force control in our revised manuscripts (see **Supplementary Video 3-7**).

This good result has been obtained thanks also to the material compensation step that address differences in the elastomers mechanical properties, however the results compare force predictions across sensors (which shows consistency) rather than against a trusted external ground-truth force sensor (like ATI F/T). This is necessary for claiming accuracy which is important for the control.

Response: Thank you for the positive evaluation on the accuracy of our force prediction models and its applicability to real robotic tasks. We apologize for not clearly stating that the compared force data in our earlier videos were all collected from external ATI F/T sensors. We have remade these videos as **Supplementary Video 2** to make this distinction explicit. Additionally, following your suggestion, we conducted new experiments comparing transferred forces with an ATI F/T sensor using real-world, unseen objects to demonstrate our model’s real-time processing and the accuracy scale across several translation groups. Please also refer to **Supplementary Video 3**.

Methodology:

- The architecture is explained in detail, but there is the lack of critical ablation studies which would enable to understand the role of each element in the pipeline in the overall performance of the system and how to change parameters to extend the work. This limits the reproducibility and the extendibility of the presented work.

Response: Thank you for the constructive comments regarding ablation studies. We agree that essential studies on each module and parameter are important for the reproducibility and extendibility of our work. As you suggested, we conduct new experiments regarding the ablation study about our model architecture in marker-to-marker translation model and force prediction model respectively.

Main modifications can also be found in our revised supplementary information as follows.

(Line 196-256) “

Text 4. Ablation study

For marker-to-marker translation model, we firstly introduce a baseline, cycleGAN⁶⁵, as it is bi-directional translation compared with the pixel2pixel GAN with only unidirectional translation. However, we want highlight cycleGAN can only deal with the translation between two sensors, which is not able to transfer across more than two sensors within one model. We test its performance by using both RGB images and binary marker images. The models are trained in 100,000 iterations with the parameters in [65] . As shown in **Supplementary Figure**

37A, the model shows inferiority in correctly translating the illumination direction, the orientation of the indenters and the marker patterns by using RGB image. On the other hand, when only use marker images (see **Supplementary Figure 37B**), the predicted image did change compared with the source image or the model is hard to converge when transferring from grid pattern to circular pattern. This explains why we did not choose RGB images as the unified modality due to its more complex illumination properties and contact geometry, which also does not exist in electronic sensor arrays. The marker image is binary without background noise, but it is hard to train with cycleGAN due to its unpaired dataset setting and the model tries to learn the major image style information, such as the illumination or contact geometry information in the RGB images but it focuses on the black area in the marker images, which is hard to capture the marker deformation. While our models capture both adversarial loss and pixel-level loss with L_{gan} and L_{rec} respectively and has been testified to performs well in our original manuscript.

Regarding the model architecture, we firstly examine the role of two primary components proposed in our training objective, i.e. adversarial loss L_{gan} and reconstruction loss L_{rec} (L2 and LPIPS). Three models with L_{gan} , L_{rec} and $L_{gan} + L_{rec}$ respectively are trained in the sim dataset with most varied marker pattern and translation directions. Specifically, we use the weights $\lambda_{gan} = 0.5$, $\lambda_{Lpips} = 5.0$, and $\lambda_{L2} = 1.0$ from our original manuscript for model with $L_{gan} + L_{rec}$, but set $\lambda_{gan} = 0$ for model only with loss L_{rec} while set $\lambda_{Lpips} = 0$, $\lambda_{L2} = 0$ for model only with L_{gan} . All models are trained with 5 epochs. We show the loss curves and the marker-to-marker translation performance in **Supplementary Figure 38**. We find that models with L_{gan} only and L_{rec} only cannot predict correct target images, either with artefacts or gray masks. While models with $L_{gan} + L_{rec}$ shows good performance and can predict visual-similar target images without artefacts or gray mask observed in the generated image. The L_{rec} part also speeds up the convergence of L_{gan} in the model with $L_{gan} + L_{rec}$.

Then, we examined the contribution of the pretrained model with simulated dataset. We compare the performance of our model when training with heterogeneous translation dataset with or without (w/o) pretrained model. We use the weights $\lambda_{gan} = 0.5$, $\lambda_{Lpips} = 5.0$, and $\lambda_{L2} = 1.0$ from our original manuscript and train both models with 5 epochs. As shown in **Supplementary Figure 39**, the model without pretraining cannot converge and predict artefacts. While the model pretrained with simulated dataset show distinct convergence speed and is able to transfer from a low-resolution sensor (uSkin, 3-axis) to a high-resolution sensor (TacTip, palm) in a visual-similar pattern. This verifies the contribution of our simulated dataset and necessary of the pretraining stage.

For the force prediction part, we include two baseline models which are commonly used in tactile sensing community, i.e the resnet backbone and renet backbone with additional LSTM module. For our model, we compare the performance of the feature encoder, and feature

encoder followed with a convGRU module with above two baselines. The outputs from those four modules are then connected with post-processing part and a regression head to output three-axis forces. Four models are test with three heterogeneous sensors. For **Supplementary Figure 40**, four models are trained with 18 indenters to show the training performance. For **Supplementary Figure 41**, 12 indenters (seen group) are used in the training stage while 6 indenters (unseen group) are test in the test stage. All models are firstly trained with 20 epochs with learning with 0.1 and another 20 epochs with learning rate of 0.001. The experiments are calculated with the mean and std trained in three times with three random seed {0,10,20}. The experimental results show that our model shows the lowest force prediction in normal force and shear forces for all sensors.

We also discover that the training batch size (1, 2, 4) affect the performance of our model performance. The experimental results in **Supplementary Figure 42** shows that training with batch size with 1 show twice times larger force errors than models training more than 1. For our model, we normally train the model with batch size larger than 2. In addition, with the increase of trained indenters, the model performance increases accordingly as it learns more information from increased data.

”

Supplementary Figure 37. Tactile image translation performance for using cycleGAN. (A) Failure case for transferring using RGB image. (i) Incorrect orientation of indenter. (ii) Incorrect circular marker pattern (B) Failure case for transferring using marker image. (i) Fail to transfer marker pattern with two grid-like marker images. (ii) Fail to converge between a grid marker image and circular marker image.

Supplementary Figure 38. Ablation study for the loss function in marker-to-marker translation model. (A) Loss curves for models with adversarial loss, reconstruction loss and both respectively. (B) Marker-to-marker translation results for three types of loss function.

Supplementary Figure 39. Ablation study for the function of pretrained model with simulated data in marker-to-marker translation. (A) Loss curves for models with adversarial loss, reconstruction loss and both respectively. (B) Marker-to-marker translation results for three types of loss function.

Supplementary Figure 40. Ablation Study for Force Prediction Performance of Different Modules Trained with All Indenters. We compare the force prediction in three axis when using ResNet, Resnet with LSTM, Feature Encoder-Only (FE-only) and our models (FE+ConvGRU) in three different sensors. Our model shows the lowest errors among all groups when training with 18 indenters in the training stage. We use random seed {0,10,20} to train 20 epochs with learning rate 0.1 plus another 20 epochs with learning rate 0.001.

Supplementary Figure 41. Ablation Study for Force Prediction Performance of Different Modules Test with Indenters Unseen in Training. We show force prediction error for 6 indenters unseen in training stage compared with ground truth from ATI nano17. Our model demonstrates the lowest errors among all groups. We use random seed {0,10,20} to train 20 epochs with learning rate 0.1 plus another 20 epochs with learning rate 0.001.

Supplementary Figure 42. Ablation Study for Batch Size on Force Prediction Performance. We show the force prediction errors for all indenters seen in training stage compared with ground truth from ATI nano17. When batch size is set to 1, the model performs poor in normal force. When batch size larger than 2, the model performs well. We use random seed 0, training 20 epochs with learning rate 0.1.

References:

[65] Zhu, Jun-Yan, et al. "Unpaired image-to-image translation using cycle-consistent adversarial networks." *Proceedings of the IEEE international conference on computer vision*. 2017.

- There is not a systematic analysis on when the model would fail and in general the limitations of the work are not discussed.

Response: Thank you for the constructive suggestions regarding a systematic analysis on the failure case and limitation of this work. As you suggested, we carefully add this analysis in the

discussion part from different sides covering the image-to-image translation, force prediction, applicable sensor type, and the applications.

Main modifications in our revised manuscript can be found as follows.

(Line 368-408)

“

However, we still find our model shows few failure cases:

1. We observed a flicker effect in homogeneous translation when transferring to GelSight (A-I) (see Supplementary Figure 41 and Supplementary Video 1 for the AII_A-I group). This issue arises in data collection when indenters with large contact areas, such as a hemisphere, cause an upward shift of the elastomer of A-I due to inadequate adhesion to the underlying plate. This results in a slight change of image style in the dataset of A-I. However, this artifact does not affect the overall force prediction performance of GelSight (A-I) (see **Fig. 4F**).
2. Another failure case arises in heterogeneous translation when the converted markers from electronic sensor arrays are too small and very sensitive in marker size and displacement. This makes the M2M model hard to train. To alleviate this, the applicable configurations are provided in Supplementary Text 8 and Supplementary Figure 12A.
3. In real-time deployment, the force prediction model exhibits zero-shift over the first few seconds and mild shift under static loading. The initial zero-shift can be mitigated by subtracting a baseline, whereas the static shift stems from the elastomer's intrinsic hysteresis issues. Additionally, when grasping metallic objects, the uSkin sensor can fail due to magnetic field interference. Nonetheless, the manipulation tasks are still completed successfully because of the gripper's heterogeneous sensor configuration (see Supplementary Videos 4–7).

Some limitations of our models and possible future work:

1. We still require at least one sensor that is fully calibrated with force-labeled data and material priors. Such a calibrated sensor can be sourced from our dataset or from other existing sensors, followed by collecting a small number of locations-pair reference images with fixed trajectory. For material priors, instead of using traditional material characterization method, some recent in-situ method^{43,44} can be referred to avoid use F/T sensor, moving stage and robot arm. Looking ahead, an important direction is to build foundational tactile database like ImageNet⁴⁵ that totally removes the need for any data collection and material calibrations. Given our model's adaptability and data reusability, the dataset is inherently scalable using our data collection trajectory and could enable users worldwide to rapidly equip their own sensors with high-performance force sensing.
2. As the force prediction model is based on unsupervised learning by transferring data from other tactile sensors, it is still not comparable to the accuracy of supervised learning which demands high-volume labelled data. However, if the user wants to get very high-accuracy force prediction, this model is expected to be very good pretrained model to be finetuned with only a few amounts of force labels.

3. It remains an open question how this model can be extended to transfer across finger-sized tactile sensors, large-area electronic skins, and sensors lacking explicit taxels, such as those based on electrical impedance tomography⁴⁶ sensors. We propose that this challenge could be addressed by dividing large-area skins into smaller subareas, thereby enabling transfer to regions comparable in size to finger-sized sensors.”

References:

43. Zhao, C., Ren, J., Yu, H. & Ma, D. In-situ Mechanical Calibration for Vision-based Tactile Sensors. in 2023 IEEE International Conference on Robotics and Automation (ICRA) 10387–10393 (2023).
44. Li, M., Zhang, L., Zhou, Y. H., Li, T. & Jiang, Y. EasyCalib: Simple and Low-Cost In-Situ Calibration for Force Reconstruction With Vision-Based Tactile Sensors. IEEE Robot Autom Lett 9, 7803–7810 (2024).
45. Deng, J. et al. ImageNet: A large-scale hierarchical image database. in 2009 IEEE Conference on Computer Vision and Pattern Recognition 248–255 (2009).
46. Park, K. et al. A biomimetic elastomeric robot skin using electrical impedance and acoustic tomography for tactile sensing. Sci Robot 7, eabm7187 (2022).

Supplementary Figure 41. Failure cases in M2M translation. (A) The generated image (i) transferring from GelSight (A-II) to GelSight (A-I) fails due to the shift (ii) of the elastomer, which changes marker distribution of some collection images in GelSight (A-I). (B) The M2M is hard to converge (i) when the reference marker from uSkin (three-axis) is set to small when without contact, while too sensitive in marker size and displacement when in-contact.

- The whole work is motivated by providing robots the ability to unify and transfer sensory information across multiple fingers ensuring stability and dexterity during object manipulation, however a real-world manipulation task where such capabilities, enabled by

the presented architecture, are evaluated is not presented. The whole work stops by demonstrating the generation of the tactile marker-based images and the force estimation. Online performances are not discussed a part what can be inferred from the videos (which is the frequency at which we can get the force estimation for example) so it is not clear if this can be really used for control.

Response: Thank you for the constructive comments regarding real-world robot manipulation tasks using our models. As you suggested, we conducted two new experiments on daily-object robot grasping and on slip detection and avoidance using the transferred force. In particular, we show that, with the transferred force, we can also transfer force controllers for the robot grasping task across heterogeneous sensors. As shown in Fig. 7, task (1), daily-object grasping, examines the application of the transferred normal force with a shared proportional force controller. Task (2), slip detection and avoidance, examines both normal and shear force sensing using a shared proportional force controller and a slip-reactive controller.

To clarify this, we add a new section to our revised manuscripts and add **Supplementary Video 3-7**.

(Line 292-323)

“Transferable force sensing and control in robot manipulation

To demonstrate the applicability of our model in robot manipulation, we installed heterogeneous tactile sensors on a robot arm to show the transferable force sensing and control in daily-object robot grasping, slip detection and avoidance (See Supplementary Figure 25A). The robot setup can be seen in **Fig. 7A-i**. In the robot grasping task, we mounted a flat-surface vision-based tactile sensor (GelSight, marker pattern A-II) on the left finger of robot and a flat-surface magnetic tactile sensor (uSkin with three-axis sensing capability) on the right finger. We transferred the force prediction model to above two sensors by using a third flat-surface vision-based tactile sensor (GelSight, marker pattern A-II) tactile sensor. The task required robot to grasp nine daily objects (see Supplementary Figure 25B) with different sizes, shapes and material without damage them. Those objects include potato chip, grape, strawberry, orange, plum, wood block, glue stick, meat box and tea box, which are unseen in the training dataset. During the grasping, the arm was controlled with a proportional controller to grasp those objects with fixed normal forces ranging from 0.6N to 1.2N. Both sensors were shared with the same force controller. As shown in **Fig. 7C**, Supplementary Video 4 and Supplementary Video 5, both sensors can equip the robot arm an accurate force sensing so that the robot arm can successfully grasp all objects with target forces without damage. Even for challenging objects such as chips and fresh fruits, the robot can achieve delicate grasping by using the transferred force prediction model combining with force control.

For the second task, we extended the robot grasping (see **Fig. 7A-ii**; Supplementary Videos 6 and 7) to include slip detection and avoidance. A curved-surface vision-based TacTip sensor (palm shape) was mounted on the left finger, and a three-axis magnetic uSkin sensor was mounted on the right. The force prediction model of TacTip was transferred from GelSight

(D-I), and the model of uSkin was transferred from the TacTip. The task proceeded through several stages: moving down, proportional-control grasping, lifting up, slip detection and avoidance at the top position, releasing, and returning to home. The force controller was active only during grasping and slip detection. We evaluated four daily objects: banana, plum, meat box, and glue stick. Beyond completing the grasp, external forces were applied by a human operator at the top position to induce slip. The robot detected slip via changes in shear force and responded by narrowing the gripper width. As shown in **Fig. 7B** and Supplementary Videos 6–7, the robot completed all stages for tested objects successfully, demonstrating the practical applications of our model in robot manipulation tasks.”

(Line 572-584) “

Robot grasping and slip control

We use a Franka FR3 robot arm with a Franka hand (max opening 0.08 m), an Intel RealSense camera, daily objects (9 for grasping and 4 for slip control), and tactile sensors. For grasping, normal force predicted from GelSight and uSkin feed into a simple proportional force controller. The gripper closes at 0.005 m/s from an object-dependent initial width (about 0.025 m for grapes up to 0.075 m for oranges). Depending on fragility, the target force is 0.6 N, 0.8 N, or 1.2 N (e.g., lower for chips and grapes, higher for rigid items like a meat box). Force controller runs at 10 Hz with $K_p=0.0004$. The slip-control experiments use the same setup, switching the tactile sensors to TacTip and uSkin. Here we pair the same proportional controller with a top-position slip check: after lifting, we monitor F_x and F_y at 20 Hz for 10 s, keeping a 0.5 s window. If the short-term shear force variation over three samples in either axis exceeds 0.3 N, we tighten the grasp by 0.001 m and command the change at 0.015 m/s. See more details in Supplementary Video 4–7.

”

Supplementary Figure 25. Applications of transferable force sensing in real robot tasks. (A) Demonstration of transferable force in robotic force control. (B) Setup for robot grasping, slip detection and avoidance using heterogeneous sensors. (C) Daily objects used for robot grasping with transferable force control.

Fig. 7 | Robot grasping and slip avoidance with transferable force sensing. (A) Transferable force sensing in robot slip detection and avoidance. (A-i) Robot arm equipped with TacTip (palm) and uSkin (three-axis). (A-ii) Sequence of robot grasping an object with force control, slip detection, and avoidance. (B) Real-time measurement of forces, slip-detection status, gripper width, and robot status during grasping task of A-ii. (C) Daily objects grasping with transferable force sensing and control using GelSight (A-II) and uSkin (three-axis). All forces models used in each sensor are transferred from other sensors. See more details in our Supplementary Video 4-7.

- In the Discussion section it is mentioned large-scale tactile sensing. When integrated on a robotic hand the sensor might need to adapt to non-flat surfaces, however the presented work doesn't seem to address this aspect. It would be better to clarify this.

Response: Thank you for the constructive suggestions regarding the discussion on non-flat surfaces tactile sensing. Although we have included TacTip sensor (palm) shape with curved surface in our study, we agree that it is necessary to clarify on large-area skins and sensors with non-flat surfaces.

Main modifications in our revised manuscript can be found as follows.

(Line 404-408)

- “It remains an open question how this model can be extended to transfer across finger-sized tactile sensors, large-area electronic skins, and sensors lacking explicit taxels, such as those based on electrical impedance tomography⁴⁶ sensors. We propose that this challenge could be addressed by dividing large-area skins into smaller subareas, thereby enabling transfer to regions comparable in size to finger-sized sensors.”

Reference:

46. Park, K. et al. A biomimetic elastomeric robot skin using electrical impedance and acoustic tomography for tactile sensing. *Sci Robot* 7, eabm7187 (2022).

- Data collection in the real world, why the movement speed is different between GelSight and uSkin and TacPalm? Also, what is the maximum speed that can be achieved? The data collection in the real world is impressive, how long it took to collect all the data? This looks contradicting the purpose of the system to reduce the need of tactile dataset acquisition.

Response: Thank you for raising the concern about movement speed. We set the speed based on two considerations: (i) collecting sufficiently dense, high-resolution contact data to improve force-prediction accuracy, and (ii) managing data volume to avoid redundancy and excessive storage. To enhance coverage and diversity, we employ 18 different indenters while ensuring the trajectory sampling is fine enough to resolve local contact variations. At the same time, we limit the acquisition rate so that consecutive frames are not overly similar and do not overwhelm memory or processing. In practice, we balance movement speed, sampling frequency, and total data volume to achieve rich yet non-redundant training data.

For each contact point, the motion comprises four steps: moving downward, lateral outward, lateral inward, and upward. We cap each step to fewer than 10 frames, yielding approximately 20–40 frames per contact point. In the heterogeneous translation experiments, we sample 160 contact points per indenter trajectory. This results in a total of 106,278 images for GelSight (≈ 37 frames per point), 108,153 images for TacTip (palm; ≈ 37.5 frames per point), and 79,762 images for uSkin (≈ 28 frames per point). The movement speed is configured as a percentage of speed on the UR5e touch panel. Accounting for differences in sensor size and corresponding moving distances, we set 25% for GelSight and uSkin, and 40% for TacTip.

While we collect high-volume data to build the force-labeled dataset, the marker-to-marker translation model requires only minimal paired data. Specifically, for each contact point, the user only needs to acquire four location-paired images, i.e. those captured at the end of each motion step, without force labels. This amounts to 11,520 paired images per sensor for training the image translation model. During inference, we apply the trained M2M model to transfer all images with force labels from calibrated sensors to generated images, approximately 90% of these images are unseen in the training stage. We then use generated images (e.g., 106,278 images from GelSight) to train the force prediction models for the target sensors. Consequently, transferring a model from a calibrated sensor to a new sensor requires collecting only 11,520 unlabeled paired images.

As you suggested, we test the time cost by setting maximum speed as 100% on our UR5e touch panel to collect location-pair images. We find that each contact points can takes less than 2.8 seconds, which means that each indenter only takes less than 7.5 minutes in total. In addition, the number of indenters can be decided by the balance the force prediction accuracy and the time cost. We test the performance of force prediction using different numbers of indenter and the results are shown in **Supplementary Table 2**. Compared with traditional force calibration, our approach reduces data by about 90%, eliminates the need for expensive high-accuracy FT sensors by leveraging force labels from calibrated sensors or our dataset, and substantially reduce calibration time.

Supplementary Figure 2. Data collection in real world. (B) Data collection trajectories are used in real world. An elastomer is divided into five surface contact points. For each surface point, the indenter will move in depth with four actions: moving downward, moving lateral outward, moving lateral inward, and moving upward for a contact point. The movement for a point in depth takes less than 3s, while the total time cost is less than 7.5 mins.

Supplementary Table 2. Ablation Study for the Number of Indenters in Seen Group on Force Prediction Performance

Number of Indenters (seen)	GelSight		
	F _x (N)	F _y (N)	F _z (N)
4	0.199	0.102	0.593
8	0.138	0.076	0.433
12	0.066	0.069	0.399

* Shown with force prediction error when tested **in unseen group** compared with ground truth from ATI nano17.

* We use our model (FE+convGRU) in this test.

* Random seed 0, training 20 epochs with learning rate 0.1.

(Line 346-349) “

1. Cost-efficient and user-friendly: Our approach alleviates the need for expensive force/torque sensors when training force prediction model for new sensors. It only requires few location-paired data, less than 10% of the force-paired images used in our experiments, substantially reducing time and cost. ”

(Line 387-398) “

Some limitations of our models and possible future work:

1. We still require at least one sensor that is fully calibrated with force-labeled data and material priors. Such a calibrated sensor can be sourced from our dataset or from other existing sensors, followed by collecting a small number of locations-pair reference images with fixed trajectory. For material priors, instead of using traditional material characterization method, some recent in-situ method^{43,44} can be referred to avoid use F/T sensor, moving stage and robot arm. Looking ahead, an important direction is to build foundational tactile database like ImageNet⁴⁵ that totally removes the need for any data collection and material calibrations. Given our model’s adaptability and data reusability, the dataset is inherently scalable using our data collection trajectory and could enable users worldwide to rapidly equip their own sensors with high-performance force sensing. ”

- Relationship of force and indentation depth: why a linear model is used? This might be the reason why the accuracy is limited to the 4N.

Response: Thanks for your comments. We are sorry that we did not clearly clarify the non-linear model we use in our paper. In our original manuscript, the relationship of force and indentation depth is fitted from the collected data during both loading and unloading phases using two-degree polynomials. Both in the loading and unloading direction is a **non-linear** model, which demonstrates the typical property of hyper-elastic material in **Fig. 5C**.

The reason why the accuracy is limited to 4N, we can see some of below reasons: (1) Most of our collected data points are crowded within -4N to 0N (shown in **Supplementary Figure 6A, 9A and 16B** below), thus it is easier to learn higher accuracy in this force range; (2) The material hysteresis problem and material harden problem of elastic material affect the saturation of tactile response, even in the loading and unloading stage, the same depth will result in different forces and the gaps increases with the applied force (see **Fig. 5C**). While the same depth (deformation) normally causes the same tactile response, which is hard to capture the differences.

The primary reasons our accuracy concentrates within 4 N are as follows: (i) the majority of training samples lie between -4 N and 0 N (**Supplementary Figure 6A, 9A, and 16B**), making this range better represented and thus easier to model with higher precision; and (ii) the elastomer exhibits material hysteresis and material hardening under large forces, which saturates the tactile response. During loading and unloading, the same indentation depth can yield different forces, with gaps growing at higher forces (see **Fig. 5C**). In contrast, the tactile image primarily reflects deformation, so identical depths tend to produce similar visual responses. Above reasons explain the lower accuracy at larger forces.

To clarify this, we add modification to our revised manuscript as follows:

(Line 565-570) “We collect paired force-depth data during both loading and unloading phases according to the parameters in the Supplementary Text 9. We can then get the fitted lines of the $F - d_z$ curves (see Supplementary Figure 8C, 13A-iii and 13A-iv), using two-degree polynomials.”

Fig. 5 | Material hardness effect and compensation. (C) Measured force-normalized depth curves for the seven elastomers during loading and unloading phases, demonstrating elastomers’ hardness and hysteresis property.

Supplementary Figure 6. Force prediction results for homogeneous sensor translation. (A) Distribution of data points collected across different force ranges for normal and shear forces across five sensors.

Supplementary Figure 9. Material compensation results for force prediction. (A) Distribution of data points collected across varying force ranges (normal and shear forces) for seven sensors.

Supplementary Figure 16. Results for heterogeneous translation with material compensation. (B) Distribution of data points collected across different force ranges for normal and shear forces across three heterogeneous sensors.

Supplementary Figure 8. Material compensation for force prediction. (B) Relationship of shear force and shear displacement for seven elastomers.

Supplementary Figure 13. Material property of three heterogeneous sensors. (A) Force-normalized depth curve of raw data (i-ii) and fitted data (iii-iv) in loading/unloading stages. Note that fitted curves are fitted with 2-degree polynomials. **(B)** Relationship of shear force and shear displacement (i) for three heterogeneous sensors. The curve (i) is then divided with normal force to show the relationship of F_S/F_N and shear displacement (ii).

- Material compensation process: what are the material priors you are referring to? Is the young's modulus?

Response: We apologize for not clearly describing the material processing and the material prior. In the revised manuscript, we provide detailed theoretical formulation and explanation of the compensation for normal force and shear force due to differences in elastic modulus and other factors such as friction coefficient.

Main modification in our Supplementary Information can be found as follows.

(Line 298 - 419) “

Text 9. Material Compensation

9.1 Relationship of Normal Force and Elastic Modulus

According to contact mechanics, when a flat rigid indenter applies normal force F_{normal} in the z direction on an elastic specimen's surface⁶⁶, the relationship between force F_{normal} and indentation depth d_z is given by:

$$F_{normal} = \alpha E^* d_z \quad (6)$$

where α is a geometric constant specific to the indenter, and E^* represents effective modulus, i.e. combined modulus of the indenter and the specimen.

For multi-material contacts, the effective modulus satisfies,

$$\frac{1}{E^*} = \frac{1-\nu^2}{E} + \frac{1-\nu'^2}{E'}, \quad \frac{1}{G^*} = \frac{2-\nu}{4G} + \frac{2-\nu'}{4G'}, \quad G = \frac{E}{2(1+\nu)} \quad (7)$$

Where E' and ν' , and E and ν , describe the elastic modulus and Poisson's ratio of the indenter and the specimen respectively. In the context of contact of a rigid indenter with a soft elastomer, the effective modulus E^* and G^* are determined by the elastomer E and G ⁶⁶.

Based on Equation (6), we can derive that in a fixed contact depth:

$$F_{normal} \propto E \quad (8)$$

9.2 Relationship of Shear Force and Elastic Modulus

When considering contacts which are loaded both in the normal direction z and in the tangential direction x and y , it is the ‘‘Cattaneo–Mindlin problem’’ ⁶⁷. We consider the solution of the tangential contact problem using the Method of Dimensionality Reduction (MDR) ⁶⁷. The shear force F_{shear} is described by a rigid lateral shift $u^{(0)}$ of the rigid indenter over an elastic base characterized by effective modulus E^* (normal) and G^* (shear). The contact has outer radius a ; the inner stick region extends to radius c and the outer annulus ($c < x < a$) slips region.

By MDR superposition, the total shear force is the sum of a stick-core term and a slip-annulus term,

$$F_{shear} = F_{stick} + F_{slip}, F_{stick} = 2cG^* u^{(0)}, F_{slip} = \mu[F_{normal}(a) - F_{normal}(c)]. \quad (9)$$

Here μ is the Coulomb friction coefficient and $F_{normal}(a)$ is the normal force associated with a contact of radius a . The MDR problem provides the normal force as

$$F_{normal}(a) = 2E^* \int_0^a [d(a) - d(x)] dx \quad (10)$$

Where indentation depth $d(x) = g(x)$, $g(x)$ is the shape profile of the indenter in a one-dimensional.

The stick–slip interface is determined by

$$u^{(0)} = \mu \frac{E^*}{G^*} [d(a) - d(c)] \quad (11)$$

which gives the stick–slip boundary with radius c .

Eliminating $u^{(0)}$ from the decomposition using the displacement criterion yields a compact expression for the shear force:

$$\begin{aligned} F_{shear} &= 2cG^*u^{(0)} + \mu[F_{normal}(a) - F_{normal}(c)] \\ &= \mu\{[F_{normal}(a) - F_{normal}(c)] + 2cE^*[d(a) - d(c)]\} \end{aligned} \quad (12)$$

Because $F_{normal}(\cdot) \propto E^*$ and $d(\cdot)$ is purely geometric, both bracketed terms in the compact expression are linear in E^* . The shear force approximately satisfies,

$$F_{shear} \propto \gamma E \quad (13)$$

Where γ is the factor related to the difference in geometry g , imposed tangential shift u , modulus ratio E^*/G^* (effective Poisson ratio) and friction coefficient μ . The effective modulus E^* and G^* are determined by the elastomer E and G similar to normal force.

9.3 Compensation for normal force and shear force

Base above relationship, to compensate the normal force, we can approximately multiply the existing force label F_{normal}^S from the source sensor S with the ratio of modulus to get compensated force label F_{normal}^{SC}

$$F_{normal}^{SC} \approx \frac{E^T}{E^S} F_{normal}^S \quad (14)$$

Regarding shear force F_{shear}^S , the shear force can be compensated:

$$F_{shear}^{SC} \approx \gamma \frac{E^T}{E^S} F_{shear}^S \quad (15)$$

where friction weighting factor γ is set as a hyperparameter.

Specifically, the compensation for differences in modulus $\frac{E^T}{E^S}$ for both normal force and shear force happens when training the force prediction model. The corrected force label $F^{SC} = \{F_{normal}^{SC}, F_{shear}^{SC}\}$ can be obtained when loading the force-image pair data. Specifically, we use contact depth d_z to index force f_z^S and f_z^T from the curves of normal force - depth in both the source sensor S and the target sensor T . Note that, the indentation depth is normalized for its easy to comparing sensors with different indentation depth due to size and material differences. The compensation ratio r can then be calculated in an incremental form:

$$r = \frac{f_z^T}{f_z^S} - 1 \quad (16)$$

We introduce two additional hyperparameters: starting depth d_0 ($0 < d_0 < d_{max}$) and correction weight λ ($0 < \lambda < 1$) to control the amount of compensation. The compensation starts when

the contact depth d_z exceeds d_0 . λ controls the compensation weight. These parameters used in our paper are obtained via grid search (see Supplementary Table 1 and Supplementary Table 2). Thus, the corrected force label F^{SC} can be derived as:

$$F^{SC} = F^S \cdot (1 + \lambda r) \quad (17)$$

Where r is r_L in loading phase while r_U in the unloading phasing indexed with the contact location d_z to catch the hysteresis property of elastomer:

$$r = \begin{cases} r_L, & \text{if } d_z > d_0 \text{ and } d_z \in L \\ 0, & \text{if } d_z \leq d_0 \\ r_U, & \text{if } d_z > d_0 \text{ and } d_z \in U \end{cases} \quad (18)$$

9.4 Choice of Material Priors

We introduce the choice of the two material priors, including elastic modulus E and friction weighting factor γ .

To get the E if not knowing the material property, we can choose below ways to acquire depending on the experimental conditions: (1) traditional materials characterization method with high accuracy but need to dismantle the sensor or break the elastomer; (2) in-situ calibration using force gauges and indenters without dismantling the sensor and breaking the elastomer⁴³; (3) in-situ mechanical calibration method without use of force/torque sensors and the need for a robotic arm with a low-cost calibration devices⁴⁴. In our test, the elastomer has non-linear mechanical property, we estimate E by measuring the relationship between applied normal forces F_N and normalized indentation depth d_z using a rigid indenter (*prism*, contact area 5×7 mm) referring to equation (6). We vertically move the indenter from a non-contact position to place near the maximum depths of sensors: GelSight (1mm), uSkin (1mm), TacTip (4.5mm). We measure the force response both in loading and unloading stage in different indentation depth due to hysteresis property. Then, we normalize indentation depths to the range of 0 to 1 so that those images collected in a step of 0.2 mm (25% of maximum depth 1mm) in GelSight and uSkin can be paired with TacTip with a step of 1.125mm (25% of maximum depth of 4.5mm). The curves are then fitted with a two-degree polynomial using the mean and variance values during three indentations shown in Supplementary Figure 8C and 13A.

However, friction weighting factor γ is hard to calibrate which involves many variables. Here, we provide empirical factors for γ only using in the real-time deployment stage to compensate the differences in geometry g , tangential shift u , modulus ratio E^* / G^* (effective Poisson ratio) and the friction coefficient μ . In the training stage, we only compensate for the shear force with E and set γ as 1, which is verified good performance in improvement of shear force in our experimental results.

To further understand the complexity of compensating shear force and the choice of our empirical factors, we exam the statistical relationship between shear force F_S and shear displacement δ (0-1mm) from both homogeneous sensors (see Supplementary Figure 8B) and heterogeneous sensors (see Supplementary Figure 13B). We use the *sphere_s* indenter (Supplementary Figure 1A, diameter of 8mm) to apply forces in different depth and direction by referring to our data collection trajectory. This resulted curve (Supplementary Figure 8B) matches the relationship shown in [68] with shear and slip stages. This curve can be further processed to the relationship between F_S / F_N and shear displacement δ (Fig. 5D), which implies the friction coefficient μ between the contact pair of indenter and elastomer. From the curves of $F_S - \delta$ (Supplementary Figure 8B) in homogeneous sensors, we can see that even in the same shear displacement, the F_S varies due to difference in hardness although we use same materials for the pigment layer on the surface. That explains the necessity of compensate $\frac{E^T}{E^S}$ in the shear force. On the other hand, the mean friction coefficient μ varies from 0.4-0.7 (see Fig. 5D and Supplementary Figure 14A) if we look at the slip region [68] in displacement of 0.6-0.8mm in curves of $F_S / F_N - \delta$. The variation range expands from 0.3-0.8 (see Fig. 5D) when considering the standard variation. Similar mean friction coefficient μ (0.5-0.7) and variation range (0.3-0.9) can be found in heterogeneous sensors (see Supplementary Figure 14B and Supplementary Figure 13B). This variation further validates that the shear force is further affected by differences in geometry, hardness et.al. Hence, to compensate γ , we empirically choose friction weighting factor γ as 2 for homogeneous translation while 3 for the heterogeneous translation by referring to the variation range of friction coefficient.

References:

43. Zhao, C., Ren, J., Yu, H. & Ma, D. In-situ Mechanical Calibration for Vision-based Tactile Sensors. in 2023 IEEE International Conference on Robotics and Automation (ICRA) 10387–10393 (2023).
44. Li, M., Zhang, L., Zhou, Y. H., Li, T. & Jiang, Y. EasyCalib: Simple and Low-Cost In-Situ Calibration for Force Reconstruction With Vision-Based Tactile Sensors. IEEE Robot Autom Lett 9, 7803–7810 (2024).
66. Fischer-Cripps, A. C. Introduction to Contact Mechanics. (Springer New York, NY, 2010). <https://doi.org/10.1007/978-0-387-68188-7>.
67. Popov, V. L., Heß, M., & Willert, E. Handbook of contact mechanics: exact solutions of axisymmetric contact problems. Springer Nature (2019).
68. Yuan, W., Li, R., Srinivasan, M. A., & Adelson, E. H. Measurement of shear and slip with a GelSight tactile sensor. In 2015 IEEE International Conference on Robotics and Automation (ICRA) 304-311 (2015).

- Real- world data for the force estimation model have been acquired with controlled indentations at predefined locations and depths involving static or slowly varying contacts. How this architecture performs in case of more dynamic contact events typical for

manipulation tasks?

Response: Thank you for the constructive suggestion on real-world testing with more dynamic contact events. Following your advice, we conducted three new experiments with diverse tactile sensors and different transfer directions to evaluate our model’s performance in robot manipulation tasks: (1) dynamic force prediction across diverse contact events, benchmarked against an ATI Nano17 F/T sensor; (2) daily-object robot grasping; and (3) slip detection and avoidance. All raw signals, RGB and marker images, and force predictions are visualized in real time using ROS 1. We also provide five new videos corresponding to these experiments to demonstrate the practical use of our model in real robot tasks.

Main modifications can be found in our revised manuscripts and **Supplementary Videos 3-7** as follows:

(Line 282-290) “

We also evaluated our model in real-time across six transfer groups under more dynamic conditions (see Supplementary Video 3). Tactile sensors were mounted on an ATI Nano17 F/T sensor, and we applied forces using four daily objects with different shapes and materials, including screwdriver, glue stick, plastic pizza, and a LEGO block (Supplementary Figure 17). A human operator performed five dynamic contact events, including press, rub, roll, push and pull, as well as continuous combinations of these on sensor surfaces (Fig. 6G). The transfer groups covered both homogeneous and heterogeneous translations described earlier. As shown in Supplementary Video 3 and Supplementary Figure 18-22, all test groups exhibited fast, accurate and robust responses comparable to a commercial high-accuracy F/T sensor.”

Supplementary Figure 17. Objects used for dynamic force test compared with ATI nano17 F/T sensor in real-time. We use four objects with different materials, sizes and shapes, including a screwdriver, glue stick, Lego block (YCB) and pizza (YCB).

Supplementary Figure 18. Real-time force prediction when pressing on a GelSight sensor. (A) Demonstration of test object, contact event, tactile sensor and nano17. The force model is transferred from a GelSight (D-I) sensor. **(B-D)** Force prediction performance in Z-axis, X-axis and Y-axis respectively.

Supplementary Figure 19. Real-time force prediction when rubbing on a TacTip sensor. (A) Demonstration of test object, contact event, tactile sensor and nano17. The force model is transferred from a GelSight (D-I) sensor. **(B-D)** Force prediction performance in Z-axis, X-axis and Y-axis respectively.

Supplementary Figure 20. Real-time force prediction when rolling on a uSkin (three-axis) sensor. (A) Demonstration of test object, contact event, tactile sensor and nano17. The force model is transferred from a GelSight (D-I) sensor. **(B-D)** Force prediction performance in Z-axis, X-axis and Y-axis respectively.

Supplementary Figure 20. Real-time force prediction when continuously rubbing and pressing on a uSkin sensor. (A) Demonstration of test object, contact event, tactile sensor and nano17. The force model is transferred from a TacTip (palm) sensor. **(B-D)** Force prediction performance in Z-axis, X-axis and Y-axis respectively.

Supplementary Figure 21. Real-time force prediction when pushing and pulling on a GelSight (D-I) sensor. (A) Demonstration of test object, contact event, tactile sensor and nano17. The force model is transferred from a uSkin (3-axis) sensor. (B-D) Force prediction performance in Z-axis, X-axis and Y-axis respectively.

Supplementary Figure 24. Dynamic force test for uSkin (z-axis) with force model transferred from GelSight (D-I). (A) Continuous pressing with a Lego block. (B) Rubbing with a plastic Pizza from YCB. (C) Pushing with a glue stick. All test is compared with ATI nano 17. See more details in our Supplementary Video 3.

Minor:

- The term “Labor-friendly” is usually used to indicate about someone or something supportive of workers' welfare and rights, it doesn't seem the appropriate adjective in this case.

Response: Thanks for your constructive suggestion on the phrasing. We have revise this word to “user-friendly” in our revised manuscript.

Main modification can be found as follows:

(Line 346) “1. Cost-efficient and user-friendly”

- The only biomimetic sensor in the study is the TacPalm while the text looks like referring to all of them as biomimetic.

Response: Thank you for pointing this out. We've clarified our terminology and revised the manuscript accordingly. We now only use “biomimetic” strictly for TacPlam (now is TacTip, Palm shape in the revised manuscript as TacTip is well-known in tactile community) sensor. We use “bio-inspired” for other tactile sensors.

Main modification can be found in our revised manuscript as follows:

(Line 41-43) “Inspired by this, tactile robotics aims to mirror the function of human mechanoreceptors to enhance dexterity and intelligence^{6,7} by developing various bio-inspired tactile sensing systems (Fig. 1A).”

(Line 779-782) Fig. 1 | Transferable force sensing. (A) Robot grasping objects with tactile sensors and force control mimics human actions with sensory receptors. These bio-inspired tactile sensors cannot transfer force data with each other due to differences in sensing principles, structural designs and material properties.”

- Please proofread again the text for more appropriate words (“disparity” among sensors, “from seen indenters”) and for typos (e.g. corase-to-fine process.).

Response: Thank you for the helpful suggestions regarding wording and typos. We have carefully proofread the manuscript and made the following changes to our revised manuscript.

(Line 130-131) “the difference is prominent with an average FID larger than 400 and an average KID larger than 0.75.”

(Line 186-187) “The difference in material hardness introduces additional errors when transferring the force labels (Fig. 5A).”

(Line 434-436) “**Derivation of unified marker representation.** When forces are applied to tactile sensors (Fig. 1D), the soft skin deforms and the skin deformation is measured as different types of tactile signals, due to difference in sensing principles.”

(Line 157-158) “The pretrained M2M model is then fine-tuned with location-paired marker images contacting with indenters from seen group.”

(Line 162-163) “The results tested only on indenters from unseen group are presented in Supplementary Figure 5C-E”

(Line 180-181) “Additional results tested on unseen group (Supplementary Figure 6C-D) further demonstrate our model’s effectiveness and generalizability.”

(Line 549) “The marker segmentation process comprises a coarse-to-fine process.”

- Even if the generated marker-based images are not linked to any specific sensor technology It would be good to have an image that shows generated marker-based images vs real marker-based images associated to a specific sensor to check noise or error. It is difficult to understand how well they are w.r.t real world sensors as there are not real-world tasks.

Response: Thank you for this insightful suggestion. We agree that the comparison between generated marker-based tactile images and real, sensor-specific marker images would help quantify the similarity. To address this, we have added side-by-side real-image comparison to the generated marker images in **Fig. 6, Supplementary Figure 1, and Supplementary Figure 5.**

Fig. 6 | Heterogeneous tactile force translation results. (D) Examples of tactile images after M2M translation.

Supplementary Figure 1. Simulation framework for marker deformation. (D) Examples of six randomly selected marker-to-marker translation results using simulated data.

Supplementary Figure 5. Marker-to-marker translation results for homogeneous sensors. (A) Randomly selected marker images after marker-to-marker translations across five GelSight sensors. Source image, generated images and target images are in the first to third column respectively. Applied indenters are labelled in the tip.

- When referring to tactile memory and transfer there is a specific reference to “use index finger memory to interpret palm stimulus” in the text, however, please note that tactile memory is not strictly localised so I would avoid being so specific as this level of granularity of spatial memory transfer is not proven.

Response: Thank you for the important clarification regarding tactile memory. We agree that the current literature does not establish spatially localized tactile memory transfer at the granularity implied by “using index finger memory to interpret palm stimulus.” To avoid such expression, we have revised relevant parts in our revised manuscript from localized level such as “fingers to palms” to skin level, i.e. “across skin regions on hands”.

The main modifications can be found in our revised manuscript as follows.

(Line 66-68) “In humans, the tactile memory system (Fig. 1C) enables the storage and retrieval of experienced tactile information, such as haptic stimuli, across skin regions on hands”

(Line 71-74) “Mimicking this unified representation and transferable tactile sensing could enable tactile sensors to learn from each other, reuse collected tactile experience and transfer tactile sensing across robot hands, enhancing their dexterity and adaptability”

(Line 783-784) “(C) Humans use a tactile memory system to estimate stimuli on unexperienced skin regions by retrieving tactile memories stored in the somatosensory cortex.”

(Line 787-789) “GenForce unifies tactile signals into marker representation, enables marker-to-marker translation across various sensors, and achieves high-accuracy force prediction on uncalibrated sensors using data transferred from calibrated sensors.”

- Figure 1A is not very informative.

Response: Thank you for pointing this out. We have revised **Figure 1A** in our revised manuscript to improve clarity and informativeness. The revised Figure 1A is shown as follows.

Fig. 1 | Transferable force sensing. (A) Robot grasping objects with tactile sensors and force control mimics human actions with sensory receptors. These bio-inspired tactile sensors cannot transfer force data with each other due to differences in sensing principles, structural designs and material properties.

- Figure 5D is very difficult to read as it is too small

Response: Thank you for the feedback on the Figure. We've improved **Figure 5** and addressed its readability in the revised manuscript. In addition, we also change all figures in our revised manuscript and supplementary information to vector graphic for better readability. The revision of Figure 5 is shown as follows.

Fig. 2 | Material hardness effect and compensation. (A) Schematic representation of tactile skins with varying hardness and their corresponding force-depth relationships. (B) Material compensation process incorporates material priors to correct force labels during loading and unloading phases. (C) Measured force-normalized depth curves for the seven elastomers during loading and unloading phases, demonstrating elastomers' hardness and hysteresis property. (D) Relationship between the shear-to-normal force F_S / F_N and shear displacement of seven elastomers measured across different contact points by using our data collection trajectory. (E-F) Force prediction errors (i-ii) before and after using material compensation in hard-to-soft group (E) and soft-to-hard group (F). (G) Fit of force prediction to ground truth demonstrates the effectiveness of material compensation when transferring from a sensor with r18 (soft) skin to r6 (hard) skin before (i) and after (ii) using material compensation.

- The videos are low resolution (especially the written text). It is not clear in the video which sensor is represented in the RGB image and marker-based image, is it the groundtruth?

Response: Thank you for the helpful feedback on the videos. We have recreated seven videos from our previous manuscript and organize them into two videos, i.e. **Supplementary Video 1-2**, with improvements in resolution, labelling, and enhanced clarity. In addition, we upload five new videos, i.e. **Supplementary Video 3-7**, for experiments regarding dynamic force prediction, robot grasping and slip detection. Detailed changes can be found in our new videos.

We sincerely appreciate your insightful feedback. Your comments have greatly improved the quality and clarity of our manuscript. We hope that the revised version, together with our detailed responses, addresses all of your concerns. Thanks !

Response Letter

We thank the reviewers again for their helpful and constructive comments on our manuscript, which have significantly improved the quality of our work. Below, we have carefully considered all the comments from reviewers and provided detailed responses to each point. The *reviewers' comments* are in *italics* text and *blue color*, and **our responses** are in **normal** text and **black color**. The **revisions** in the **revised manuscript and Supplementary Information** are in **red color**. The **highlight** uses **bold font**.

Responses to Comments from Reviewer# 1

Reviewer #1 (Remarks to the Author):

Thank you to the authors for the improvements in shear force compensation, visuo-tactile sensor transfer, real-world experiments, and the motivation explanation, which make this work more complete.

Response: We sincerely thank the reviewer for thorough evaluation of our manuscript again. Thank you for acknowledging the improvement of our revised manuscript, particularly in shear force compensation, visuo-tactile transfer, real-world robot experiments and motivation explanation.

My main concerns have been addressed, but there is still a minor typo: There is a symbol error in Equation (15) — it has two $F^{\{SC\}}_{Shear}$.

Response: We are sorry for the typo in Equation (15). The Equation (15) has been rectified in our revised supplementary information, which can be found as follows.

(Line 351-352) “

Regarding shear force F_{shear}^S , the shear force can be compensated:

$$F_{shear}^{SC} \approx \gamma \frac{E^T}{E^S} F_{shear}^S \quad (15)$$

We sincerely appreciate the reviewer's insightful feedback. The reviewer's comments have greatly improved the quality and clarity of our manuscript. Thanks!

Reviewer #2 (Remarks to the Author):

Thank you for the substantial revisions and the extensive experiments added to the manuscript and supplementary material to address the reviewer's comments. The manuscript has evolved substantially, and the idea of creating a shared tactile representation for cross-sensor force prediction is interesting and timely. Your expanded experiments and supplementary material help clarify several aspects of the method. At the same time, a number of elements would benefit from further clarification, restructuring, or moderation to ensure the contribution aligns fully with the evidence presented. The suggestions below are offered to help strengthen the paper in this direction.

Response: We thank the reviewer again for the thorough evaluation and constructive suggestions. We appreciate the reviewer's insightful feedback, which has helped us significantly strengthen the rigor of our work. **We have carefully revised the manuscript by adding more data to figures, moderating claims, restructuring figures, and rephrasing relevant terminologies to strictly align with the presented evidence.** All textual changes are highlighted in red in the revised manuscript, and every point is addressed in the detailed point-by-point response below, with specific references to the relevant lines, figures, tables, and videos.

1) To address the reviewer's concern about the claim that the method allow generalization "across tactile technologies," it has been included in the Discussion section, a statement suggesting that the approach could extend to any tactile sensor by dividing the sensing surface into artificial "taxels" to produce a marker-like displacement field. Since this idea is not demonstrated experimentally, and may not hold for sensors whose outputs do not encode surface deformation in a physically meaningful way, it would be helpful to frame this more conservatively. The evidence in the paper shows that the method works well for sensors that "naturally" produce marker-based or taxel-based 2D deformation patterns (e.g., GelSight, TacTip, uSkin). For tactile technologies that do not exhibit such outputs (e.g., resistive textile based sensors, whisker sensors, EIT sensor, etc), the feasibility of generating an equivalent marker representation remains untested. Moderating the claim explicitly stating that the approach is designed for sensors with this family of outputs, would make the contribution clearer and more aligned with the demonstrated results.

Response: We thank the reviewer for the constructive suggestions on moderating the claim of our contributions. We agree that it is **more rigorous to explicitly state** that our method is designed for sensors that produce an output that can be expressed as marker-based or taxel-based 2D deformation patterns. On the other hand, the feasibility of applying such a method to tactile sensors that do not exhibit markers or taxels, such as EIT sensors, remains an open question. **To moderate the claims, we have revised this paragraph according to the reviewer's suggestions.**

Main modifications to the revised manuscript can be found as follows.

(Line 423-426) “

3. This approach has been designed and tested for tactile sensors that naturally produce outputs that can be expressed as marker-based or taxel-based 2D deformation patterns. Extending this framework to tactile sensors lacking explicit spatial taxels remains an open challenge, such as Electrical Impedance Tomography (EIT) sensors⁴⁶.”

2) *The revised manuscript is clearly well-polished, and the use of AI-based tools for grammatical refinement is entirely acceptable. However, some resulting terminology (e.g., “slip avoidance,” “short-term shear variation”) does not align with established vocabulary in tactile sensing or robotic manipulation. Slip detection, slip compensation / grasp stabilisation, and rate of change of shear force constitute better terminology. Please ensuring that the wording remains consistent with the terminology used in the robotics community to improve clarity and make the experimental descriptions easier to interpret.*

Response: We appreciate the reviewer’s guidance on aligning our terminology with standard usage in the robotics community. We agree that terms such as "slip compensation" and "rate of change of shear force" are more precise and constitute better terminology. In response to the reviewer’s suggestions, we have updated the manuscript, figures and videos to replace resulting terminology (**e.g., replacing "slip avoidance" with "slip compensation," and "short-term shear variation" with "change of shear force"**) to ensure consistency with established vocabulary in tactile sensing and robotic manipulation. Furthermore, we have carefully proofread the manuscript to improve the clarity and quality of the language throughout the paper.

Main modifications to the revised manuscript can be found as follows.

(Line 28) “This transferable force sensing is also demonstrated with high performance in robot force control including daily-object grasping, slip detection and compensation.”

(Line 105) “including daily objects grasping, slip detection and compensation”

(Line 302) “show the transferable force sensing and control in daily-object robot grasping, slip detection and compensation”

(Line 316) “For the second task, we extend the robot grasping with slip detection and compensation by using multi-sensor force coordination”

(Line 323) “The task proceeds through several stages: moving down, grasping, lifting up, slip detection and compensation at the top position, releasing, and returning to home”

(Line 600) “For robot slip detection and compensation, the robot only detects slip at the top position with a controller runs at 20 Hz for 10 seconds”

(Line 856) “Robot grasping and slip compensation with transferable force sensing”

(Line 859) “A sequence showing the robot grasping a strawberry with force control, slip detection and slip compensation”

3) *Several main figures, particularly Figs. 4–7, are dense, and hard to read due to the small size, making it difficult to understand key components without consulting the supplementary*

material. As a result, it is difficult to understand the full meaning of the results without repeatedly referring to the supplementary figures, which contain essential information. This constant switching between documents interrupts the flow of reading and makes it harder to follow the narrative. I would suggest a reorganisation of the figures, bringing the most important supplementary elements into the main figures and enlarging the size to make the main text more self-contained and improve the readability.

Response: We thank the reviewer for the constructive suggestions on restructuring the figures to improve readability. We agree that Figures 4–7 were previously overly dense and that the reliance on supplementary material interrupted the narrative flow. In response to the reviewer’s suggestions, **we have comprehensively reorganized these figures**. We remove unessential parts into supplementary material and make the results in the main figures self-contained. We have enlarged the figure layouts to ensure all components are clearly legible. We also carefully check and reorganize some of figures in the supplementary information to improve the readability. The corresponding references for those figures are also revised in the main text.

Main modifications to the revised manuscript can be found as follows. (see next page due to size of figures)

Fig. 1 | Homogeneous tactile force translation. (A) Data collection setup. (B) Data collection trajectory for acquiring sequential tactile images under normal and shear forces. (C) Marker segmentation for RGB tactile images. (D) Comparison of force prediction errors when using source-only method and GenForce model in radar plots. F_t denotes the total force. All units are Newton (N). (E) Histogram of the coefficient of determination (R^2) values for predicted forces with source-only method and GenForce model. (F) Real-time force prediction over 1000 frames after using GenForce model in group A-II_D-I.

Fig. 2 | Material hardness effect and compensation. (A) Schematic representation of tactile skins with varying hardness and their corresponding force-depth relationships. (B) Measured force-normalized depth curves for seven elastomers during loading and unloading phases, demonstrating elastomers' hardness and hysteresis property. (C) Relationship between the shear-to-normal force F_s / F_N and shear displacement of seven elastomers measured across different contact points. (D) Material compensation process incorporates material priors to correct force labels during loading and unloading phases. (E-F) Force prediction errors (i-ii) for GenForce models without (w/o) and with material compensation in *hard-to-soft* group (E) and *soft-to-hard* group (F).

Fig. 3 | Heterogeneous tactile force translation. (A) Signal-to-marker process. The multichannel electrical signals (ii) from a sensor array (i) can be unified to marker images (iii) regardless of 3-axis sensing capability or z-axis-only sensing capability. (B) Examples of tactile images after M2M translation. (C) Histogram of R^2 values (i-ii) and radar plot (iii-iv) of force MAE when using source-only method and GenForce model with material compensation. (D) Real-time demonstration of force prediction when target domains are uSkin (3-axis) after using GenForce model with material compensation. (E) Force prediction using GenForce model with dynamic contact events compared with ATI nano 17 F/T sensor (see Supplementary Video 2 and 3).

4) In Fig. 7, only TacTip's predicted forces are shown. It is therefore difficult to determine how uSkin contributes to grasp control, or how the two sensors' outputs are combined. Since the Panda gripper uses a single actuator, explaining precisely how slip detection is triggered and how both sensors influence the control loop would make the demonstration stronger and easier to interpret. Including uSkin's predicted forces in the plot would help support and validate the multi-sensor aspect of the experiment.

Response: We thank the reviewer for the constructive suggestions regarding Figure 7. We agree that only displaying TacTip's predicted force is difficult to fully evaluate the multi-sensor functionality and the contributions of uSkin in control loop. In response to the reviewer's suggestions, we have made the following improvements:

1. **Updated Figure 7 and Supplementary Video 7 with uSkin's predicted forces:** We have added new experiments to validate multi-sensor force coordination in robot grasping and slip compensation. The updated figures and videos successfully demonstrate how both sensors influence the control loop. The previous Supplementary Video 6-7 are combined into Supplementary Video 6 to demonstrate single-sensor force control.
2. **Clarified Multi-sensor Coordination Logic:** We have expanded the text in the Results and Methods sections to detail the multi-sensor grasp control. We have clarified how force information from the two sensors is fused to control the single-actuator Panda gripper and to close the control loop. Specifically, for robot grasping, the controller utilizes the mean normal force from both sensors as feedback to modulate the gripper width. The averaged force demonstrates smoother profiles (see Figure 7) compared with single sensor, which is beneficial for stabilizing the grasping process due to sudden changes in one sensor. For robot slip detection and compensation, the controller keeps a 0.5 s window with three historical force measurements. If the change of shear force in one of sensors over the window exceeds 0.2 N, we tighten the grasp width by using the weighted sum of changes of shear force from both sensors and command changes of the gripper width at 0.015 m/s. By using this strategy, if one sensor indicates a significant change of force while the other remains stable, the controller can still quickly determine the presence of slip, as demonstrated in Figure 7. On the other hand, when slip is detected, the weighted sum of the gripper's width response can achieve a more adaptive response by combining changes of shear force from both sensors.

Main modifications to the revised manuscript can be found as follows.

(Line 316-343) “

For the second task, we extend the robot grasping with slip detection and compensation by using multi-sensor force coordination (Fig. 7). A curved-surface vision-based TacTip sensor is mounted on the left finger of the robot hand, while a three-axis magnetic uSkin sensor is mounted on the right (Fig. 7A). The y-axis of uSkin is inverted due to mirrored configuration. The force prediction model of TacTip is transferred from a GelSight (D-I), and the force prediction model of uSkin is transferred from a TacTip. These two transfer directions are designed to demonstrate the generalizability of our model in diverse transfer groups for real-

robot task. The task proceeds through several stages: moving down, grasping, lifting up, slip detection and compensation at the top position, releasing, and returning to home. The force controller of the robot hand is active only during grasping and slip detection stage. We use three objects with different sizes and surface conditions from the YCB dataset: strawberry (rough), banana (medium smooth) and egg (smooth). External forces are applied by a human operator at the top position to induce slip. The robot grasps objects by using the mean normal force (F_z) from both sensors when it reaches 1N. The robot detects slip if changes of shear force (F_x or F_y) in TacTip or uSkin exceed 0.2N, while the robot compensates the slip by narrowing the gripper width by using the weighted sum of changes of shear force from both sensors (see Methods). As shown in **Fig. 7B** and Supplementary Video 7, the robot completes all stages successfully by coordinating the multi-sensor forces. By using this multi-sensor strategy, the mean normal force demonstrates smoother profiles compared with single sensor, which is beneficial for stabilizing the grasping process due to sudden changes of force in one sensor. If one sensor indicates a significant change of force while the other remains stable, the controller can quickly determine the presence of slip, such as the first slip is detected by TacTip but not by uSkin in **Fig. 7B**. On the other hand, when slip is detected, the weighted sum of the gripper’s width response can achieve an adaptive response by combining changes of shear force from both sensors. We also provide more demonstrations for this task with four objects (Supplementary Figure 27C) in Supplementary Video 6 to show the force control by using single sensor. Above tasks successfully demonstrate the practical applications of our GenForce model in robot manipulation, particularly a new paradigm for coordinating heterogeneous sensors on robot hands with transferable force sensing.”

(Line 582-598) “

Robot grasping and slip compensation

We use a Franka FR3 robot arm with a Franka hand (maximum opening 0.08 m), an Intel RealSense camera, fifteen daily objects (nine for grasping and six for slip compensation), and three tactile sensors. For robot grasping in the first task, we implement a proportional control loop that modulates the gripper width based on normal force feedback from one sensor. This high-level controller runs at 10 Hz with a gain of $K_p = 0.0004$. During execution, the gripper closes at a slow speed of 0.005 m/s from an object-dependent initial width (ranging from 0.025 m for grapes to 0.075 m for oranges). The target normal force is adapted to the object’s fragility and is set at 0.6 N, 0.8 N, or 1.2 N (e.g., lower forces for fragile items like chips and grapes, and higher forces for rigid items like a meat box). Specifically, the same grasping controller is used in the second task, but it utilizes the mean normal force from both sensors as feedback. For robot slip detection and compensation, the robot only detects slip at the top position with a controller runs at 20 Hz for 10 seconds, keeping a 0.5-second window with three historical force measurements. If the change of shear force from any sensors over the window exceeds $\tau = 0.2$ N, we tighten the grasp width Δw by considering both sensors and command the changes of gripper width at a rate of 0.015 m/s:

$$\Delta w = \frac{\Delta w_{base}}{|N|} \sum_{i=1}^N \frac{\delta F_i}{\tau}$$

where $\Delta w_{base} = 1$ mm is the base compensation width, N is the number of sensors currently detecting slip (i.e., where $\delta F_i > \tau$), δF_i is change of shear force of the i -th sensor.”

Fig. 4 | Robot grasping and slip compensation with transferable force sensing. (A) Multi-sensor force coordination with transferred force models. (i) A Franka Hand equips left finger with a TacTip sensor while right finger with a uSkin sensor. (ii) A sequence showing the robot grasping a strawberry with force control, slip detection and slip compensation. **(B)** Real-time measurements of forces, slip-detection status, gripper width, and robot status during the robot grasping task shown in A-ii. See more examples in Supplementary Video 4-7.

Supplementary Figure 27. Transferable force sensing in real-robot tasks. (A) Objects used for robot grasping task. **(B)** Daily objects grasping with transferable force sensing and control, using GelSight (A-II) and uSkin (three-axis). **(C)** Objects and setup used for robot slip detection and compensation with single sensor. **(D)** Objects and setup used for robot slip detection and compensation with multiple sensors.

5) The dynamic-contact results in Supplementary Figs. 18–24 provide helpful qualitative insight, but the human-operated trajectories are not repeatable and cannot serve as quantitative demonstrations of dynamic performance. Clarifying that these results are qualitative and avoiding strong claims based solely on them would make the interpretation more consistent with the presented evidence.

Response: We appreciate the reviewer’s insightful comment regarding the dynamic-contact results. The human-operated trajectories were intended to mimic the unstructured contact events commonly seen in robot manipulation tasks, serving as a complement to the structured trajectories presented in Supplementary Video 2. However, we agree that due to the inherent lack of repeatability in human trajectories, the results in Supplementary Figures 18–24 should be interpreted as qualitative demonstrations of robustness rather than quantitative results. In accordance with the reviewer’s suggestion, **we have revised the manuscript accordingly and have moderated the associated claims.**

Main modifications to the revised manuscript can be found as follows.

(Line 287-296) “We also qualitatively evaluate our model in real-time across six transfer groups under more dynamic conditions (see Fig. 6E and Supplementary Video 3). Tactile sensors are mounted on an ATI Nano17 F/T sensor, and we apply random forces using four daily objects not seen in our dataset with different shapes and materials, including screwdriver, glue stick, plastic pizza, and a LEGO block (Supplementary Figure 19). A human operator performs five unstructured, dynamic contact events commonly seen in robot manipulation tasks, including press, rub, roll, push and pull, as well as continuous combinations of these on sensor surfaces. The transfer groups cover both homogeneous and heterogeneous translations described earlier. As shown in Supplementary Video 3 and Supplementary Figure 20-24, all test groups exhibit good generalizability to new objects and low-force-error responses when compared with a commercial high-accuracy F/T sensor.”

6) Thank you for your detailed comparison with [30]. Some of the points you raise, particularly the force-prediction capability, are very useful to understand the novelty of your method. However, several other entries in Supplementary Table 1 are presented only conceptually, since no experimental evidence is provided to support them. It would strengthen the manuscript to adjust the wording so that the comparison reflects only what is directly demonstrated. More generally, I am not convinced that the table is necessary; a clear and focused supporting explanation in the text could convey the same information without requiring a separate table.

Response: We thank the reviewer for the constructive suggestion regarding the necessity of Supplementary Table 1. We agree that a focused textual explanation is more appropriate than a table containing conceptual comparisons. **In accordance with your suggestion, we have removed Supplementary Table 1 and provided a concise comparison in the main text.** This revision specifically highlights the novelty of our method regarding unified tactile representation and force-prediction capabilities, ensuring that all claims are strictly grounded in the experimental evidence provided.

Main modifications to the revised manuscript and figures can be found as follows.

(Line 83-86) “Some efforts to directly transform tactile signals across sensors³⁰⁻³³ similarly fall short on one-to-one translation, as they overlook the importance of a unified tactile representation among sensors and lack of generalizability to diverse tactile sensors with force-prediction capability (see ~~Supplementary Table 1~~.)”

7) Please provide a clear definition of the “source-only” baseline, as it is referenced multiple times. Readers who are not already familiar with this term may have difficulty understanding the comparisons without an explicit description.

Response: We thank the reviewer for highlighting the need for a clear definition of the “source-only” baseline. We agree that an explicit description is essential for readers to fully understand the comparative results. In response to the reviewer’s suggestion, we have added a precise definition to the position where it first appears in our revised manuscript.

Main modifications to the revised manuscript can be found as follows.

(Line 141-142) “A common approach is to directly apply trained force prediction models from existing sensors to other sensors³⁹, i.e., the source-only method.”

I hope these suggestions are helpful to improve the final version of the paper.

Response: We sincerely appreciate the reviewer’s professional feedback, which has been very helpful in strengthening the rigor of our manuscript. The reviewer’s comments have greatly improved the quality and clarity of our work. We hope that the revised version, together with our detailed responses, addresses all of the reviewer’s concerns. Thank you!